# AGENT-CONTROLLER REPRESENTATIONS: PRINCIPLED OFFLINE RL WITH RICH EXOGENOUS INFORMATION

## ABSTRACT

Learning to control an agent from data collected offline in a rich pixel-based visual observation space is vital for real-world applications of reinforcement learning (RL). A major challenge in this setting is the presence of input information that is hard to model and irrelevant to controlling the agent. This problem has been approached by the theoretical RL community through the lens of *exogenous information*, i.e, any control-irrelevant information contained in observations. For example, a robot navigating in busy streets needs to ignore irrelevant information, such as other people walking in the background, textures of objects, or birds in the sky. In this paper, we focus on the setting with visually detailed exogenous information, and introduce new offline RL benchmarks offering the ability to study this problem. We find that contemporary representation learning techniques can fail on datasets where the noise is a complex and time dependent process, which is prevalent in practical applications. To address these, we propose to use multi-step inverse models, which have seen a great deal of interest in the RL theory community, to learn Agent-Controller Representations for Offline-RL (ACRO). Despite being simple and requiring no reward, we show theoretically and empirically that the representation created by this objective greatly outperforms baselines.

## 1 INTRODUCTION

Effective real-world applications of reinforcement learning or sequential decision-making must cope with exogenous information in sensory data. For example, visual datasets of a robot or car navigating in busy city streets might contain information such as advertisement billboards, birds in the sky or other people crossing the road walks. Parts of the observation (such as birds in the sky) are irrelevant for controlling the agent, while other parts (such as people crossing along the navigation route) are extremely relevant. How can we effectively learn a representation of the world which extracts just the information relevant for controlling the agent while ignoring irrelevant information?

Real world tasks are often more easily solved with fixed offline datasets since operating from offline data enables thorough testing before deployment which can ensure safety, reliability, and quality in the deployed policy (Lange et al., 2012; Ebert et al., 2018; Kumar et al., 2019; Jaques et al., 2019; Levine et al., 2020). The Offline-RL setting also eliminates the need to address exploration and planning which comes into play during data collection.[1] Although approaches from representation learning have been studied in the online-RL case, yielding improvements, exogenous information has proved to be empirically challenging. A benchmark for learning from offline pixel-based data (Lu et al., 2022a) formalizes this challenge empirically. Combining these challenges, is it possible to learn distraction-invariant representations with rich observations in offline RL?

Approaches for discovering small tabular-MDPs ($\leq$500 discrete latent states) or linear control problems invariant to exogenous information have been introduced (Dieterich et al., 2018; Efroni et al., 2021; 2022b;a; Lamb et al., 2022) before. However, the planning and exploration techniques in these algorithms are difficult to scale. A key insight that Efroni et al. (2021); Lamb et al. (2022) uncovered is the usefulness of multi-step action prediction for learning exogenous-invariant representation.

---

[1] This elimination however can make offline RL more difficult if the wrong data is collected.

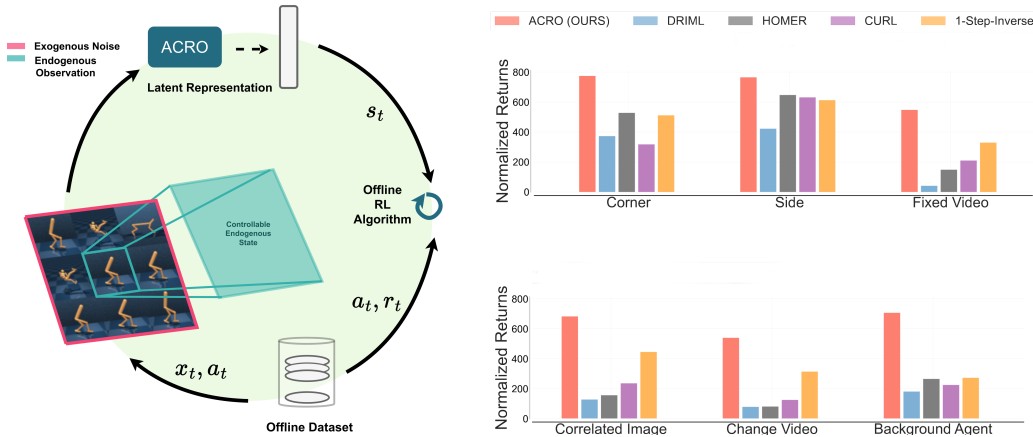

Figure 1: **Left: Representation Learning for Visual Offline RL in Presence of Exogenous Information**. We propose ACRO, that recovers the controller latent representations from visual data which includes uncontrollable irrelevant information, such as observations of other agents acting in the same environment. **Right: Results Summary**. ACRO learns to ignore the observations of task irrelevant agents, while baselines tend to capture such exogenous information. We use different offline datasets with varying levels of exogenous information (Section 5) and find that baseline methods consistently under-perform w.r.t. ACRO, as is supported by our theoretical analysis.

Following these, we propose to learn *Agent-Controller Representations for Offline-RL (ACRO)* using multi-step inverse models, which predict actions given current and future observations as in Figure 2. ACRO avoids the problem of learning distractors, because they are not predictive of the agent's actions. This property even holds for temporally-correlated exogenous information. At the same time, multi-step inverse models capture all the information that is sufficient for controlling the agent (Efroni et al., 2021; Lamb et al., 2022), which we refer to as the agent-controller representation. ACRO is learned in an entirely reward-free fashion. Our first contribution is to show that ACRO outperforms all current baselines on datasets from policies of varying quality and stochasticity. Figure 1 gives an illustration of ACRO, with a summary of our experimental findings.

A second core contribution of this work is to develop and release several new benchmarks for offline-RL designed to have especially challenging exogenous information. In particular, we focus on *diverse temporally-correlated* exogenous information, with datasets where (1) every episode has a different video playing in the background, (2) the same STL-10 image is placed to the side or corner of the observation throughout the episode, and (3) the observation consists of views of nine independent agents but the actions only control one of them (see Fig. 1). Task (3) is particularly challenging because which agent is controllable must be learned from data.

Finally, we also introduce a new theoretical analysis (Section 3) which explores the connection between exogenous noise in the learned representation and the success of Offline-RL. In particular, we show that Bellman completeness is achieved from the agent-controller representation of ACRO while representations which include exogenous noise may not verify it. Bellman completeness has been previously shown to be a sufficient condition for the convergence of offline RL methods based on Bellman error minimzation (Munos, 2003; Munos & Szepesvári, 2008; Antos et al., 2008).

## 2 ACRO: AGENT-CONTROLLER REPRESENTATIONS FOR OFFLINE-RL

### 2.1 PRELIMINARIES

We consider a Markov Decision Process (MDP) setting for modeling systems with both relevant and irrelevant components (also referred as exogenous block MDP in Efroni et al. (2021)). This MDP consists of a set of observations, $\mathcal{X}$; a set of latent states, $\mathcal{Z}$; a set of actions, $\mathcal{A}$; a transition distribution, $T(z' \mid z, a)$; an emission distribution $q(x \mid z)$; a reward function $R : \mathcal{X} \times \mathcal{A} \to \mathbb{R}$; and a start state distribution $\mu_0(z)$. We also assume that the support of the emission distributions of any two latent states are disjoint. The latent state is decoupled into two parts $z = (s, e)$ where $s \in \mathcal{S}$ is the agent-controller state and $e \in \mathcal{E}$ is the exogenous state. For $z, z' \in \mathcal{Z}, a \in \mathcal{A}$ the transition function is decoupled as $T(z' \mid z, a) = T(s' \mid s, a)T_e(e' \mid e)$, and the reward only depends on

$(s, a)$. These definitions imply that there exist mappings $\phi_\star : \mathcal{X} \to \mathcal{S}$ and $\phi_{\star,e} : \mathcal{X} \to \mathcal{E}$ from observations to the corresponding controller and exogenous and uncontrollable latent states. The agent interacts with the environment, generating a latent state, observation and action sequence, $(z_1, x_1, a_1, z_2, x_2, a_2, \cdots, )$ where $z_1 \sim \mu(\cdot)$ and $x_t \sim q(\cdot \mid z_t)$. The agent does not observe the latent states $(z_1, z_2, \cdots)$, instead receiving only the observations $(x_1, x_2, \cdots)$. The agent chooses actions using a policy distribution $\pi(a \mid x)$. A policy is an *exo-free policy* if it is not a function of the exogenous noise. Formally, for any $x_1$ and $x_2$, if $\phi_\star(x_1) = \phi_\star(x_2)$, then $\pi(\cdot \mid x_1) = \pi(\cdot \mid x_2)$.

## 2.2 PROPOSED METHOD

We consider learning representations from an offline dataset $\mathcal{D} = (\mathcal{X}, \mathcal{A})$ consisting of sequences of N observations $\mathcal{X} = (x_1, x_2, x_3, ..., x_N)$ and the corresponding actions $\mathcal{A} = (a_1, a_2, a_3, ..., a_N)$. We are in the rich-observation setting, *i.e.*, observation $x_t \in \mathbb{R}^m$ is sufficient to decode $z_t$. Our focus is on pre-training an encoder $\phi : \mathbb{R}^m \to \mathbb{R}^d$ on $\mathcal{D}$ such that the frozen representation $s_t = \phi(x_t)$ is suitable for offline policy optimization.

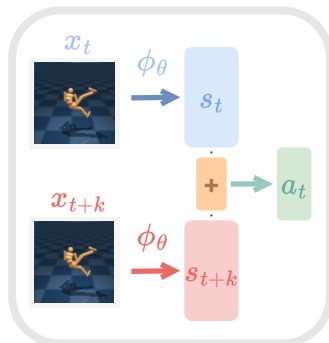

To learn representations that discard exogenous information, we leverage prior works from the theoretical RL community and train a multi-step inverse action prediction model, which captures long-range dependencies thanks to its conditioning on distant future observations. This leads to the ACRO objective, namely to predict the action conditioning on $\phi(x_t)$ and $\phi(x_{t+k})$. Note that even though we are conditioning on the future observation, we only predict the first action instead of the sequence of actions up to the k-th timestep, as the former is easier to learn.

Figure 2: **ACRO** is a multi-step inverse model that predicts the first action conditioned on the current state and the future state. **+** denotes concatenation.

Our proposed method, which we call *Agent-Controller Representations for Offline-RL* (ACRO), optimizes the following objective based on a multi-step inverse model:

$$\phi_\star \in \arg\max_{\phi \in \Phi} \mathbb{E}_{t \sim U(0,N)} \mathbb{E}_{k \sim U(0,K)} \log\left(\mathbb{P}(a_t \mid \phi(x_t), \phi(x_{t+k}))\right). \tag{1}$$

This approach is motivated by two desiderata: (i) ignoring exogenous information and (ii) capturing the latent state that is necessary for control. The following invariance lemma (Lemma 1, see Appendix A for proof) states that optimal action predictor models can be obtained without dependence on exogenous noise, when the data-collection policy is assumed not to depend on it either.

**Lemma 1** (Invariance Lemma: Multi-step inverse model is invariant to exogenous information, Lamb et al. (2022)). *For any exo-free policy $\pi : \mathcal{X} \to \mathcal{A}$, for all $a_t \in \mathcal{A}$, and $(x_t, x_{t+k}) \in$ supp $\mathbb{P}_\pi(X_t, X_{t+k})$:*

$$\mathbb{P}_\pi(a_t \mid x_t, x_{t+k}) = \mathbb{P}_\pi(a_t \mid \phi_\star(x_t), \phi_\star(x_{t+k})) \tag{2}$$

At the same time, prior works have shown that single step inverse models of action prediction can fail to capture the full controller latent states (Efroni et al., 2021; Lamb et al., 2022; Hutter & Hansen, 2022). One type of counter-example for single-step inverse models stems from a failure to capture long-range dependencies. For example, in an empty gridworld, a pair of positions which are two or more spaces apart can be mapped to the same representation without increasing the loss of a one-step inverse model. Another simple counter-example involves a problem where the last action the agent took is recorded in the observation, in which case the encoder can simply retrieve that action directly while ignoring all other information (although recording all recent actions in the observation is an issue for multi-step inverse models). The use of multi-step inverse models resolves both of these counter-examples and is able to learn the full agent-controller state (Lamb et al., 2022).

We emphasize here that even though inverse models of action prediction have appeared in past literature (as discussed in related works), they are often proposed for the purposes of exploration and reward bonus. In contrast, we propose to learn the multi-step inverse model to explicitly uncover a

representation that contains only the controller, endogenous part of the state. Recently, Lamb et al. (2022) proposed a multi-step inverse model where the learnt representation $\phi(\cdot)$ is regularized, so that $\phi(\cdot)$ discards irrelevant details from observations $x$. This was accomplished by using vector-quantization on the encoder's output, forcing discrete latent states to be learnt for constructing a tabular MDP for latent recovery. In contrast, ACRO learns the continuous endogenous latent state, without a bottleneck, and the learnt pre-trained representation $\phi(\cdot)$ is later used for policy optimization in offline RL. More details of our algorithm are discussed in Appendix I.

## 3 BENEFITS OF EXOGENOUS INVARIANT REPRESENTATION IN OFFLINE RL

Due to its importance to practical applications, the offline RL setting has been extensively studied by the theoretical community. The majority of provable value-based offline RL algorithms follow a Bellman error minimization approach (Munos, 2003; Munos & Szepesvári, 2008; Antos et al., 2008), in line with the techniques used in practice. The common representational assumptions needed to derive these results are: (A1) the function class contains the optimal Q function (realizability), (A2) the data distribution is sufficiently diverse (concentrability), and (A3) Bellman completeness (Munos & Szepesvári, 2008). This last condition is the most subtle one, it states that the function class can properly represent the Bellman backup of any function it contains.

**Definition 2** (Bellman Completeness). *We say that a function class $\mathcal{F}$ is Bellman complete if it is closed under the Bellman operator. That is, for any $f \in \mathcal{F}$ it holds that $\mathcal{T}f \in \mathcal{F}$, where $(\mathcal{T}f)(x, a) \equiv R(x, a) + \mathbb{E}_{x' \sim T(x'|x,a)}[\max_{a'} f(x', a')]$ for all $(x, a) \in \mathcal{X} \times \mathcal{A}$.*

Chen & Jiang (2019) conjectured that (A1) and (A2) alone are not sufficient for sample efficient offline RL, and, recently, Foster et al. (2021) established a lower bound proving this claim. Thus, the representational requirements needed for offline RL are more intricate than in supervised learning.

With these observations in mind, we highlight a key advantage of the agent-controller representation $\phi_\star$ relatively to other representations in the offline RL setting. Namely, we show one can construct a Bellman complete function class on top of $\phi_\star$, while some representations that include exogenous information provably violate Bellman completeness. To formalize these claims, we denote by $\mathcal{Q}_\mathcal{S} = \{(s, a) \mapsto [0, 1] : (s, a) \in \mathcal{S} \times \mathcal{A}\}$ the set of Q-functions defined over $\mathcal{S}$, and for a given representation $\phi$, we let $\mathcal{F}(\phi) = \{(s, a) \mapsto Q(\phi(s), a) : Q \in \mathcal{Q}_\mathcal{S}, (s, a) \in \mathcal{S} \times \mathcal{A}\}$ denote the set of Q-functions defined on top of $\phi$. The following proposition states that the Agent-Controller representation leads to a Bellman complete function class (all proofs in Appendix D.1/ Appendix D.2).

**Proposition 3** (ACRO Representation is Bellman Complete). *$\mathcal{F}(\phi_\star)$ is Bellman complete.*

Next, we show that there exists a representation strictly more expressive than ACRO (*i.e.*, one that includes exogenous information) which, surprisingly, violates the Bellman completeness property.

**Proposition 4** (Exogenous Information May Violate Bellman Completeness). *There exists $\phi$ which is a refinement[2] of $\phi_\star$ such that $\mathcal{F}(\phi)$ is not Bellman complete.*

This proposition implies that exogenous information being included in the representation may break the Bellman completeness assumption, which is a requirement for establishing the convergence of offline RL algorithms based on Bellman error minimization. From this perspective, additional information in the representation may deteriorate the performance of offline RL. Conversely, a coarser representation may trivially violate the realizability assumption A1: such a representation may merge states on which the optimal Q-function differs, preventing it from being realized.

Together, these observations motivate the experimental pipeline used this work: learn the agent-controller representation by optimizing Equation 20, then perform offline RL on top of it. In doing so, we obtain a representation that is sufficient for optimal performance, and yet filters the exogenous information which can (i) be impossible to exactly model, and (ii) hurt the offline RL performance.

## 4 RELATED WORK

In Table 1, we list prior works and whether they verify various properties, in particular invariance to exogenous information. An extended discussion on related works is provided in Appendix E.

---

[2]Let $\mathcal{X}$ be a finite set of elements. Given a partition $P$ of $\mathcal{X}$ let its induced equivalence relation be denoted by $\sim_P$. A partition $P_1$ is finer than $P_2$ if for any $x_1, x_2 \in \mathcal{X}$ such that $x_1 \sim_{P_1} x_2$ it also holds that $x_1 \sim_{P_2} x_2$.

Table 1: **Overview of Properties** of prior works on representation learning in RL, in particular their robustness to exogenous information. The comparison to ACRO aims to be as generous as possible to the baselines. ✗ is used to indicate a known counterexample for a given property.

| Algorithms | TD3 (DrQ) | CURL | DRIML | DBC | AE | 1-Step Inverse | Behavior Cloning | BYOL Explore | **ACRO** (Ours) |
|---|---|---|---|---|---|---|---|---|---|
| Time-Ind. Exo-Invariant | ✓ | ✗ | ✓ | ✓ | ✗ | ✓ | ✓ | ✓ | ✓ |
| Reward Free | ✗ | ✓ | ✓ | ✗ | ✓ | ✓ | ✓ | ✓ | ✓ |
| Exogenous Invariant | ✗ | ✗ | ✗ | ✓ | ✗ | ✓ | ✓ | ? | ✓ |
| Non-Expert Policy | ✓ | ✓ | ✓ | ✓ | ✓ | ✓ | ✗ | ✓ | ✓ |
| Agent-Controller Rep. | ✓ | ✗ | ✓ | ✗ | ✓ | ✗ | ✓ | ✗ | ✓ |

**Inverse Dynamics Models**. One-Step Inverse Models predict the action taken conditioned on the previous and resulting observations. This is invariant to exogenous noise but fails to capture the agent-controller latent state (Efroni et al., 2021), as previously discussed in Section 2.2. This can result from inability to capture long-range dependencies (Lamb et al., 2022) or could result from trivial prediction of actions using a dashboard displaying the last action taken, such as the brakelight which turns on after the break is applied on a car (De Haan et al., 2019). Behavior Cloning predicts actions given current state and may also condition on future returns. This is invariant to exogenous noise but can struggle with non-expert policies and generally fails to learn agent-controller latent state. Inverse models predicting sequences of actions, like GLAMOR (Paster et al., 2020) considers an online setting where they learn an action sequence as a sequential multi-step inverse model and rollout via random shooting and re-scoring, using both the inverse-model accuracies and an action-prior distribution. On the other hand, we learn a representation fully offline with a multi-step inverse model and then do policy optimization over the learnt representation, given fixed dataset.

**Contrastive Methods**. CURL (Augmentation Contrastive, Laskin et al. (2020)) learns a representation which is invariant to a class of data augmentations while being different across random example pairs. Depending on what augmentations and datasets are used, the learnt representations would generally learn exogenous noise and also fail to capture agent-controller latent states (which could be removed by some augmentations). HOMER and DRIML (Time Contrastive, Misra et al. (2020), Mazoure et al. (2020)) learns representations which can discriminate between adjacent observations in a rollout and pairs of random observations. This has been proven to not be invariant to exogenous information and neither can capture the agent-controller latent state (Efroni et al., 2021).

**Predictive Models**. Autoencoders learn to reconstruct an observation through a representation bottleneck. Generative modeling approaches usually capture all information in the input space which includes both exogenous noise and the agent-controller latent state (Hafner et al., 2019). Wang et al. (2022a;b) showed that a generative model of transition in the observation space can decompose the space into agent-controller state and exogenous information. While this does, in principle, eventually achieve an Agent-Controller representation, it comes at the cost of learning the exogenous representation and its dynamics before discarding the information. BYOL-EXPLORE (Guo et al., 2022) achieved impressive empirical results in online exploration by predicting future representations based on past representations and actions. While this approach can ignore exogenous information, there is no guarantee that it will do so, nor that it will learn the full agent-controller state.

**Reward-Dependent Methods**. DRQV2 (Kostrikov et al., 2020) learns a value function from offline tuples of observations, rewards, and actions. This could feasibly ignore exogenous noise given a suitable data-collection policy, but will not generally learn the full agent-controller latent state due to a heavy dependence on the reward structure. DBC (Bisimulation, Zhang et al. (2020)) learns representations which have similar values under a learned value function. In general, bisimulation is an overly restrictive state abstraction that fails to transfer to different tasks.

**RL with Exogenous Information**. Several prior works study the RL with exogenous information problem. In Dietterich et al. (2018); Efroni et al. (2022a;b) the authors consider specific representational assumptions on the underlying model, such as linear dynamics or factorized representation of the exogenous information in observations. Our work focuses on the rich observation setting where the representation itself should be learned. Efroni et al. (2021) proposes a deterministic path planning algorithm for being invariant to exogenous noise. Unlike their approach which requires interaction with the environment using a tabular-MDP, ACRO is a purely offline algorithm. Lastly,

Table 2: **EASY-EXO**. Comparison of different representation methods on the standard v-d4rl benchmark, without additional exogenous information. ACRO consistently outperforms baseline methods in visual offline data. Performance plots in Appendix Figure 10. 10 seeds and std. dev. reported.

| ENVIRONMENT | DATASET | ACRO | DRIML | HOMER | DRQv2 | CURL | 1-STEP INVERSE |
|---|---|---|---|---|---|---|---|
| CHEETAH-RUN | Expert | $451.0 \pm 3.9$ | $330.2 \pm 2.9$ | $227.8 \pm 1.6$ | $256.9 \pm 2.2$ | $213.0 \pm 0.6$ | $239.9 \pm 0.4$ |
| | Medium-Expert | $466.0 \pm 3.2$ | $399.2 \pm 2.5$ | $390.7 \pm 1.3$ | $388.1 \pm 3.5$ | $328.4 \pm 2.1$ | $299.3 \pm 0.6$ |
| | Medium | $528.7 + 0.8$ | $508.5 \pm 0.7$ | $518.1 \pm 0.4$ | $488.3 \pm 0.5$ | $377.0 \pm 0.8$ | $400.3 \pm 0.4$ |
| | Medium-Replay | $416.9 \pm 0.9$ | $233.3 \pm 1.2$ | $333.2 \pm 1.2$ | $381.5 \pm 1.6$ | $279.4 \pm 1.8$ | $272.3 \pm .6$ |
| WALKER-WALK | Expert | $924.5 \pm 2.2$ | $485.10 \pm 4.9$ | $670.55 \pm 4.1$ | $888.6 \pm 6.0$ | $800.36 \pm 2.5$ | $831.5 \pm 3.4$ |
| | Medium-Expert | $914.6 \pm 1.8$ | $438.3 \pm 3.3$ | $774.5 \pm 2.5$ | $906.6 \pm 0.9$ | $724.6 \pm 4.5$ | $651.8 \pm 4.0$ |
| | Medium | $486.7 \pm 0.2$ | $469.4 \pm 0.5$ | $485.1 \pm 0.7$ | $425.6 \pm 1.6$ | $429.0 \pm 2.0$ | $389.4 \pm 1.1$ |
| | Medium-Replay | $277.8 \pm 0.5$ | $204.3 \pm 3.4$ | $318.9 \pm 4.0$ | $308.5 \pm 1.5$ | $234.8 \pm 2.4$ | $146.7 \pm 0.7$ |
| HUMANOID-WALK | Expert | $79.9 \pm 1.1$ | $17.5 \pm 0.1$ | $21.6 \pm 0.4$ | $34.1 \pm 0.3$ | $28.5 \pm 0.2$ | $25.4 \pm 0.1$ |
| | Medium-Expert | $142.4 \pm 1.2$ | $26.8 \pm 0.2$ | $31.8 \pm 0.1$ | $70.8 \pm 0.5$ | $63.2 \pm 0.9$ | $56.3 \pm 0.5$ |
| | Medium | $103.8 \pm 1.8$ | $35.1 \pm 0.3$ | $53.8 \pm 0.4$ | $96.4 \pm 0.9$ | $40.6 \pm 0.4$ | $46.7 \pm 0.1$ |
| | Medium-Replay | $197.8 \pm 0.5$ | $92.6 \pm 0.3$ | $102.7 \pm 0.6$ | $121.0 \pm 0.4$ | $77.8 \pm 0.8$ | $100.7 \pm 1.1$ |
| AVERAGE | | 415.8 | 270.0 | 327.4 | 363.9 | 299.7 | 288.4 |

the work of Lamb et al. (2022) suggests an endogenous latent state recovery algorithm through the use of a discretization bottleneck to construct a small tabular-MDP. In contrast, ACRO recovers the continuous counterpart of the endogenous latent state space directly, without the need to construct a tabular-MDP. Moreover, here we focus on reward optimization, and not only latent state discovery.

## 5 EXPERIMENTS: OFFLINE RL WITH EXOGENOUS INFORMATION

This section provides extensive analysis of representation learning from visual offline data under rich exogenous information (Figure 3). Our experiments aim to understand the effect of exogenous information and if ACRO can truly learn the agent controller state and thus improve performance in visual offline RL. To this end, we evaluate ACRO against several state of the art representation learning baselines across two axes of added exogenous information: *Temporal Correlation* and *Diversity*, hence characterizing the level of difficulty systematically. We find that under exogenous information in offline RL, the performance of several state of the art representation learning objectives can degrade dramatically.

Two particular challenges in the datasets we explore are the temporal correlation and diversity in the exogenous noise. *Temporal Correlation:* Exogenous noise which lacks temporal correlation (time-independent noise) is relatively easy to filter out in the representation, especially in tasks where the agent-controller latent state has

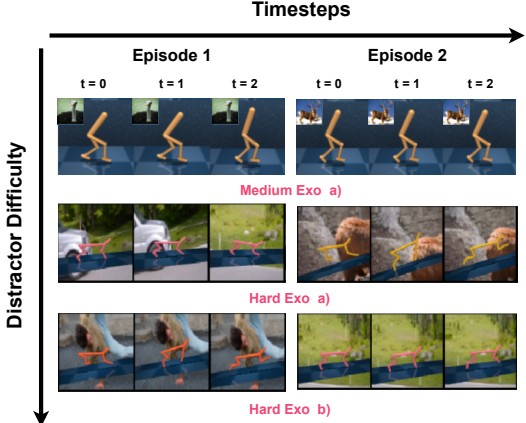

Figure 3: **Examples** of Different Categories of Exogenous Information. Further details of different exogenous information in offline datasets, with visual examples, are provided in Appendix F.1.

strong temporal correlation. *Diversity:* Similarly for the other axis, if exogenous noise is more diverse, it has a greater impact on the complexity of the subsequently learned representation. For example, if there are only two possible distracting background images, in the worst case the cardinality of a discrete representation is only doubled. On the other hand if there are thousands of possible distracting background images, then the effect on the complexity of representation would be far greater. We primarily categorize our novel visual offline datasets into *three categories* (Figure 3 in appendix provides observations under different exogenous distractors):

- **EASY-EXO**. Exogenous noise with low-diversity and no time correlation. **a)** Visual offline datasets from v-d4rl benchmark (Lu et al., 2022b) without any background distractors; **b)** Distractor setting (Lu et al., 2022a) with a single fixed exogenous image in the background.

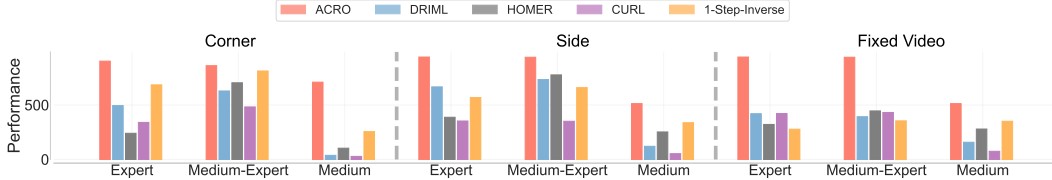

Figure 4: **MEDIUM-EXO Results**. Performance comparison of ACRO with several other baselines, with varying levels of exogenous information settings, either from STL10 dataset (Coates et al., 2011) or fixed video distractors in background during offline data collection.

- **MEDIUM-EXO**. Exogenous noise with either low-diversity or simple time-correlation. **a)** Exogenous image placed in the corner of agent observations, changes per episode; **b)** Exogenous image placed on the side of agent observations, changes per episode; **c)** A single fixed exogenous video playing in the background.

- **HARD-EXO**. Exogenous noise with both high-diversity and rich temporal correlation. **a)** Exogenous image in the background which changes per episode; **b)** Exogenous video in the background which changes per episode; **c)** Exogenous observations of nine agents placed in a grid, but the actions only control one of the agents (see Figure 1).

**Experiment Setup**. We provide details of each EXOGENOUS DATASETS in Appendix F.1, along with descriptions for the data collection process in Appendix G. Following Fu et al. (2020); Lu et al. (2022b), we release these datasets for future use by the RL community. All experiments involve pre-training the representation, and then freezing it for use in an offline RL algorithm. We use TD3 + BC as the downstream RL algorithm, along with data augmentations (Kostrikov et al., 2020), a combination which has been shown to be a reasonable baseline for visual offline RL (Lu et al., 2022b). Experiment setup and implementation details are discussed in Appendix I.

**Baselines**. We compare *five* other baselines, which are standard for learning representations of visual data. The baselines we consider are: (i) two temporal contrastive learning methods, DRIML (Mazoure et al., 2020) and HOMER (Misra et al., 2020); (ii) a data augmentation method, DRQ Kostrikov et al. (2020), and a spatial contrastive approach, CURL Laskin et al. (2020); and (iii) inverse dynamics model learning, *i.e.*, 1-step inverse action prediction (Pathak et al., 2017). We do not consider baselines such as SPR (Schwarzer et al., 2020) and SGI (Schwarzer et al., 2021) which work well on the ALE Atari100K benchmark but not on continuous control benchmarks (Tomar et al., 2021). We also include

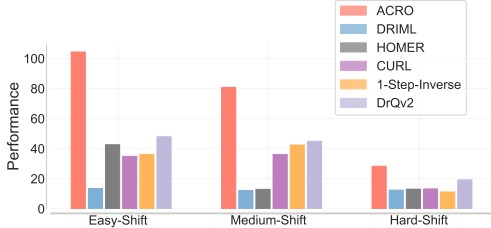

Figure 5: Normalized results across two domains from the v-d4rl distractor suite with varying levels of data shift severity. The easy, medium and hard categories are within the v-d4rl distractor suite of varying shift severity.

preliminary Atari results in Appendix Section H where the representations are pre-trained using ACRO and used over a Decision Transformer architecture (Chen et al., 2021b).

## 5.1 EASY-EXOGENOUS INFORMATION OFFLINE DATASETS

Table 2 summarizes results from the v-d4rl benchmark with visual offline data (Lu et al., 2022a). We label this as EASY-EXO since the dataset only contains a blank background without any additional exogenous noise being added. We find that ACRO learns a good agent-controller latent representation from pixel data with no apparent noise in observations, and can lead to effective performance improvements through pre-training representations. Extending results of EASY-EXO with static uncorrelated image background distractors from the v-d4rl benchmark, we see that the performance significantly decreases for all methods, while ACRO can strongly outperform all baselines, with the smallest drop in performance. Figure 5 shows normalized results across two different datasets and domains from v-d4rl. The distractors in this case belong to varying degree of shifts in the data distribution, according to (Lu et al., 2022a).

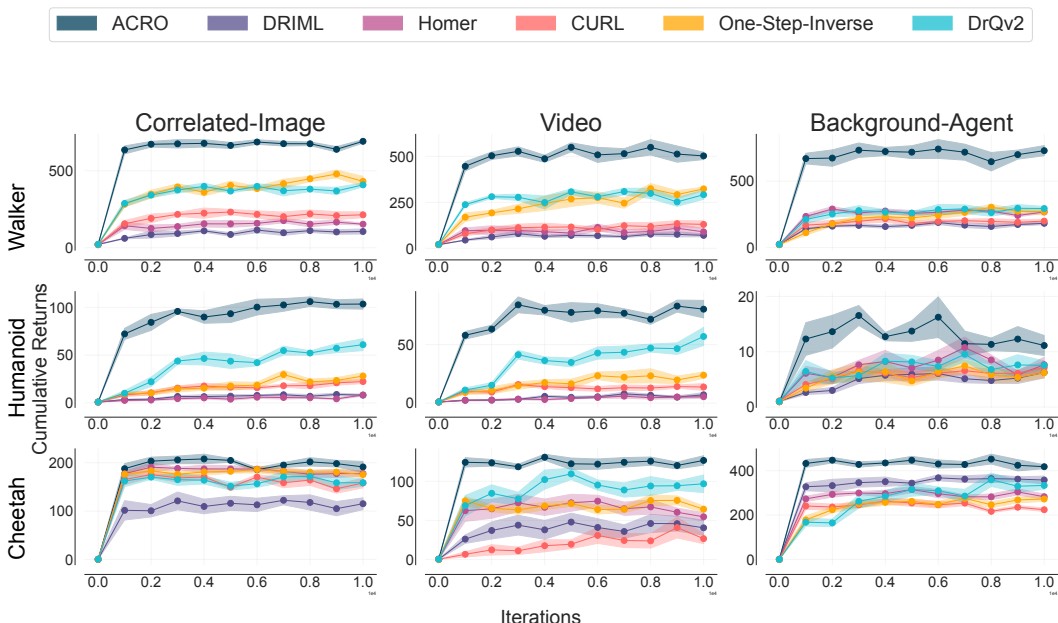

Figure 6: **HARD-EXO Results** Normalized performance across three datasets: medium, expert, and medium-expert. **First Column**. Time correlated exogenous distractor in background; **Second Column**. Video distractors that changes per episode in background; **Third Column**. Multiple background agent observations as distractors are placed in a grid of agent observation space

## 5.2 MEDIUM-EXOGENOUS INFORMATION OFFLINE DATASETS

Figure 4 shows normalized results across three domains (cheetah-run, walker-walk, humanoid-walk) for the MEDIUM-EXO setting. Among these, the fixed background video is the hardest task. As expected, most methods underperform on data collected from a medium policy, compared to medium-expert and expert policies. However, ACRO consistently outperforms all methods across all datasets and distractor settings. Note the high variability in performance of baselines when changing the type of exogenous information (from corner, to side, to fixed video). Conversely, with the exception of Corner + Medium policy, ACRO performs similarly for all three settings. This suggests that baseline methods learn representations that are affected by the exogenous information, while ACRO remains rather impervious to it.

## 5.3 HARD-EXOGENOUS INFORMATION OFFLINE DATASETS

With correlated exogenous noise in the form of either images or video, we observe that baseline representation objectives can be remarkably broken. Figure 6 shows normalized performance comparisons across different types of datasets (expert, medium-expert, medium) for three different types of HARD-EXO settings. Comparatively, ACRO can be more robust to the hard exogenous distractors, even though as the HARD-EXO types increase in difficulty, the maximum performance reached by all methods can degrade. Among the three HARD-EXO settings, changing video distractors in background during data collection seems to be the hardest, leading to performance drops for most methods. This suggests there is a strong correlation issue between the representation and the video pixels, which breaks when the episode changes, hence leading to the worst scores across the three settings. However, ACRO remains comparatively robust and outperforms all baselines across all the HARD-EXO settings.

## 5.4 METHOD ABLATIONS AND ANALYSIS

We compare ACRO with three of its variations, 1) when $k = 1$, i.e. a standard one-step inverse model; 2) with $\mathbf{x}_{t+k}$ not provided as input, i.e. simply training the representation with a behavior cloning loss; and 3) when $m(k)$ the timestep embedding, is additionally provided as input. Ablations are shown over three different policies: random, medium-replay and expert in Table 3.

For both random and medium-replay policies, $k = 1$ leads to similar results than when $k$ is randomly chosen from 1 to 15. ACRO performs much better under an expert policy. We conjecture that the benefits of larger $k$ can only be realized when the policy is of a high enough quality to preserve information over long time horizons. Additionally training a behavior cloning loss performs similarly to ACRO for the medium-replay and expert datasets. However, when the actions come from a random policy, ACRO performs much better while the behavior cloning ablation collapses completely. This result is analyzed theoretically in Appendix B, which shows

Table 3: **Ablations** for different policies. The highlighted cells indicate where each variant fails to match ACRO's performance, hence showing that each component of ACRO is essential for consistently good performance. 5 seeds and std. dev. reported.

| ENVIRONMENT | RANDOM | MEDIUM-REPLAY | EXPERT |
|---|---|---|---|
| ACRO | $82.9 \pm 5.5$ | $228.8 \pm 50.1$ | $525.8 \pm 89.0$ |
| K=1 | $94.7 \pm 7.9$ | $241.0 \pm 9.9$ | $187.5 \pm 33.8$ |
| ONLY $x_t$ | $0.5 \pm 0.1$ | $229.4 \pm 64.7$ | $496.8 \pm 100.2$ |
| WITH $k$ | $43.1 \pm 49.5$ | $251.8 \pm 15.3$ | $302.2 \pm 29.1$ |

that ACRO is equivalent to behavior cloning under a deterministic and fixed expert policy, but should be much better otherwise. Adding a $k$ embedding generally degrades performance, although the effect is inconsistent. These results suggest that ACRO is a more well rounded and robust objective than other variants.

**Visualizing Reconstructions**. Having learnt a representation, we can train a decoder over it to minimize the reconstruction loss given the original observation. Such reconstructions would therefore measure how much information in the original observation is preserved in the representation, and thus act as a metric for evaluating the quality of representations. We compare such reconstructions in Figure 7 for the cheetah domain where the exogenous noise comes from a video playing in the background. Notably, ACRO is able to remove most background information while keeping the relevant body pose information intact. On the other hand, DRIML performs contrastive comparisons between states in a given trajectory and is not able to remove exogenous information quite as well. DRQ is able to remove exogenous noise but is unable to learn the controller state.

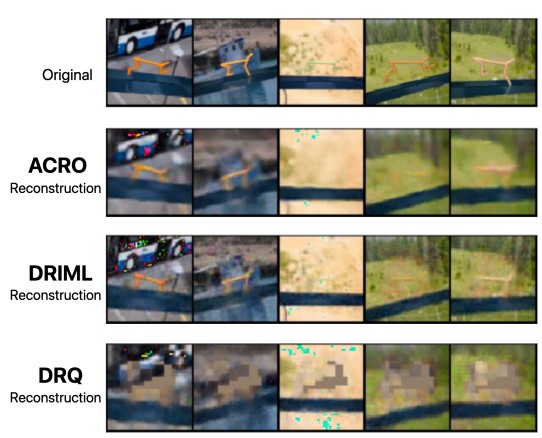

Figure 7: **Reconstructions** from a decoder with a static background image per episode: **Top-Bottom**: Original, ACRO, DRIML, DRQ.

## 6 DISCUSSION

In this work, we introduced offline RL datasets with varying difficulties of exogenous information in the observations. Our results show that existing representation learning methods can significantly drop in performance for certain types of exogenous noise. We presented ACRO, a pre-training objective for offline RL based on a multi-step inverse prediction model, and showed it is far more robust to exogenous information, both theoretically and empirically. We hope this work will drive future interests in offline RL under different definitions of exogenous information.

**Limitations and Future Work**. Since ACRO does not require reward information for learning representations, it is natural to wonder if data from multiple datasets (e.g. combining random and medium-replay) or different domains (e.g. a transfer learning setting) can be used to train a stronger representation than when using a single dataset. Additionally, under varying transition dynamics across tasks, a model-based counterpart of ACRO would be interesting to study. For the domains and tasks considered in this work, it would be interesting to quantify how accurately ACRO recovers the underlying relevant or endogenous latent states. We also studied variations of ACRO through ablation studies, and believe that with carefully designed information bottlenecks, ability of ACRO to further recover relevant latent states for empirical improvement can be significantly improved.

## 7 REPRODUCIBILITY STATEMENT

The experiment details and use of different exogenous offline datasets are discussed in section 5. We provide further architecture and algorithm details in appendix I. For implementation, we use the open source codebase from the visual D4RL benchmark (Lu et al., 2022a) and implement ACRO and other baseline representation objectives using the same structure and pipeline from the codebase. All the representation learning objectives use the same encoder architecture and optimization specifications. In the main draft, we provide normalized performance plots for comparison, where we average across different types of datasets (expert, medium, medium-expert); and we provide individual performance plots for comparison in the appendix. We also provide code for our implementation along with supplementary materials. The use of different exogenous offline datasets hopefully introduces new benchmarks to be considered in the offline RL community, and we plan to release these benchmark datasets for future use. The proof for propositions and lemmas are included in details in the appendix.

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

# Appendix

## A    INVARIANCE LEMMA PROOF

For any $k > 0$, consider $x, x' \in \mathcal{X}$ such that $x$ and $x'$ are separated by k steps. Both proofs first use bayes theorem, then apply the factorized transition dynamics, and then eliminate terms shared in the numerator and denominator. This proof is essentially the same as lemmas found in Lamb et al. (2022); Efroni et al. (2021), but is presented here for clarity.

**Lemma 1** states that the multi-step inverse model is invariant to exogenous noise. For any exo-free policy $\pi : \mathcal{X} \to \mathcal{A}$, for all $a_t \in \mathcal{A}$, and $(x_t, x_{t+k}) \in \texttt{supp}\, \mathbb{P}_\pi(X_t, X_{t+k})$:

$$\mathbb{P}_\pi(a_t \mid x_t, x_{t+k}) = \mathbb{P}_\pi(a_t \mid \phi_\star(x_t), \phi_\star(x_{t+k})) \tag{3}$$

*Proof.*

$$
\begin{aligned}
&\mathbb{P}_{\pi,\mu}(a \mid x', x) \\
&\overset{(a)}{=} \frac{\mathbb{P}_{\pi,\mu}(x' \mid x, a)\mathbb{P}_{\pi,\mu}(a \mid x)}{\sum_{a'} \mathbb{P}_{\pi,\mu}(x' \mid x, a')} \\
&\overset{(b)}{=} \frac{\mathbb{P}_{\pi,\mu}(x' \mid x, a)\pi(a \mid \phi_\star(x)))}{\sum_{a'} \mathbb{P}_{\pi,\mu}(x' \mid x, a')\pi(a' \mid \phi_\star(x))} \\
&\overset{(c)}{=} \frac{q(x' \mid \phi_\star(x'), \phi_{\star,e}(x'))\mathbb{P}_{\pi,\mu}(\phi_\star(x') \mid \phi_\star(x), a)\mathbb{P}_{\pi,\mu}(\phi_{\star,e}(x') \mid \phi_{\star,e}(x))\pi(a \mid \phi_\star(x))}{\sum_{a'} q(x' \mid \phi_\star(x'), \phi_{\star,e}(x'))\mathbb{P}_{\pi,\mu}(\phi_\star(x') \mid \phi_\star(x), a')\mathbb{P}_{\pi,\mu}(\phi_{\star,e}(x') \mid \phi_{\star,e}(x))\pi(a' \mid \phi_\star(x))} \\
&= \frac{\mathbb{P}_{\pi,\mu}(\phi_\star(x') \mid \phi_\star(x), a)\pi(a \mid \phi_\star(x))}{\sum_{a'} \mathbb{P}_{\pi,\mu}(\phi_\star(x') \mid \phi_\star(x), a')\pi(a' \mid \phi_\star(x))}.
\end{aligned}
$$

$\square$

Relation $(a)$ holds by Bayes' theorem. Relation $(b)$ holds by the assumption that $\pi$ is uniformly random (in the first proof) or exo-free (in the second proof). Relation $(c)$ holds by the factorization property. Thus, $\mathbb{P}_{\pi,\mu}(a \mid x', x) = \mathbb{P}_{\pi,\mu}(a \mid \phi_\star(x'), \phi_\star(x))$, and is constant upon changing the observation while fixing the agent controller state.

## B    CONNECTION BETWEEN ACRO AND BEHAVIOR CLONING

In the special case where all data is collected under a fixed deterministic, exogenous-free policy, ACRO and behavior cloning become equivalent. This case can still be non-trivial, if the start state of the episode is stochastic or if the environment dynamics are stochastic.

**Lemma 5.** *Under fixed, deterministic, and exo-free policy $\hat{\pi} : \mathcal{X} \to \mathcal{A}$, multi-step inverse model is equivalent to behavior cloning. For all $a_t \in \mathcal{A}$, and $(x_t, x_{t+k})$ such that $\mathbb{P}_{\hat{\pi}}(X_t = x_t, X_{t+k} = x_{t+k}) > 0$ we have:*

$$\mathbb{P}_{\hat{\pi}}(a_t \mid \phi_\star(x_t), \phi_\star(x_{t+k})) = \mathbb{P}_{\hat{\pi}}(a_t \mid \phi_\star(x_t)) \tag{4}$$

The proof of this claim is simply that behavior cloning is already able to predict actions perfectly in this special case, so there can be no benefit to conditioning on future observations.

*Proof.* By the assumption of the deterministic exo-free policy, we have that $\mathbb{P}_{\hat{\pi}}(a_t = \hat{a}(\phi_\star(x_t)) \mid \phi_\star(x_t)) = 1$ where $\hat{a} : \mathcal{S} \to A$ is a function mapping the latent state to the action.

Using bayes theorem we write:

$$\mathbb{P}_{\hat{\pi}}(a_t \mid \phi_\star(x_t), \phi_\star(x_{t+k})) = \frac{\mathbb{P}_{\hat{\pi}}(\phi_\star(x_{t+k} \mid \phi_\star(x_t), a_t)\mathbb{P}_{\hat{\pi}}(a_t \mid \phi_\star(x_t))}{\sum_{a''} \mathbb{P}_{\hat{\pi}}(\phi_\star(x_{t+k} \mid \phi_\star(x_t), a'')\mathbb{P}_{\hat{\pi}}(a'' \mid \phi_\star(x_t))} \tag{5}$$

For any examples in the dataset and for all $a' \in A$:

Case 1: $a' = \hat{a}(\phi_\star(x_t))$. It holds that

$$\mathbb{P}_{\hat{\pi}}(a_t \mid \phi_\star(x_t), \phi_\star(x_{t+k})) = \frac{\mathbb{P}_{\hat{\pi}}(\phi_\star(x_{t+k} \mid \phi_\star(x_t), a_t)\mathbb{P}_{\hat{\pi}}(a_t \mid \phi_\star(x_t))}{\sum_{a''} \mathbb{P}_{\hat{\pi}}(\phi_\star(x_{t+k} \mid \phi_\star(x_t), a'')\mathbb{P}_{\hat{\pi}}(a'' \mid \phi_\star(x_t))}$$

$$\mathbb{P}_{\hat{\pi}}(a_t \mid \phi_\star(x_t), \phi_\star(x_{t+k})) = \frac{\mathbb{P}_{\hat{\pi}}(\phi_\star(x_{t+k} \mid \phi_\star(x_t), a_t = a')}{\mathbb{P}_{\hat{\pi}}(\phi_\star(x_{t+k} \mid \phi_\star(x_t), a_t = a')}$$

$$\mathbb{P}_{\hat{\pi}}(a_t \mid \phi_\star(x_t), \phi_\star(x_{t+k})) = 1.$$

On the other hand, it holds that $\mathbb{P}_{\hat{\pi}}(a_t = a' \mid \phi_\star(x_t)) = 1$ since $a' = \hat{a}(\phi_\star(x_t))$. Hence, for this case, the claim holds true.

Case 2: $a' \neq \hat{a}(\phi_\star(x_t))$. It holds that

$$\mathbb{P}_{\hat{\pi}}(a_t \mid \phi_\star(x_t), \phi_\star(x_{t+k})) = \frac{\mathbb{P}_{\hat{\pi}}(\phi_\star(x_{t+k} \mid \phi_\star(x_t), a_t)\mathbb{P}_{\hat{\pi}}(a_t \mid \phi_\star(x_t))}{\sum_{a''} \mathbb{P}_{\hat{\pi}}(\phi_\star(x_{t+k} \mid \phi_\star(x_t), a'')\mathbb{P}_{\hat{\pi}}(a'' \mid \phi_\star(x_t))}$$

$$\mathbb{P}_{\hat{\pi}}(a_t \mid \phi_\star(x_t), \phi_\star(x_{t+k})) = \frac{0}{\mathbb{P}_{\hat{\pi}}(\phi_\star(x_{t+k} \mid \phi_\star(x_t), a_t = \hat{a}(\phi_\star(x_t)))}$$

$$\mathbb{P}_{\hat{\pi}}(a_t \mid \phi_\star(x_t), \phi_\star(x_{t+k})) = 0.$$

On the other hand, it holds that $\mathbb{P}_{\hat{\pi}}(a_t = a' \mid \phi_\star(x_t)) = 0$ since $a' \neq \hat{a}(\phi_\star(x_t))$. Hence, for this case the claim also holds true. This concludes the proof since the two distributions are equal in both cases. $\square$

## C  PREDICTING THE FIRST ACTION VS. PREDICTING ACTION SEQUENCES

In ACRO we only predict the first action from $x_t$ to $x_{t+k}$ rather than predicting the entire action sequence. In an environment with deterministic dynamics, we will prove that these two approaches are asymptotically equivalent. For the proof we will also make a stronger assumption that the dynamics are deterministic in the learned latent space, i.e. for the learned encoder $\phi$, there exists a function $f$ such that: $\phi(x_j) = f(\phi(x_t), a_{t:j})$. This assumption will hold for the optimal $\phi$, and it is also likely to be empirically true since $\phi$ is a high-dimensional continuous latent state, thus no two points are likely to have exactly the same representation. In a stochastic environment, the two approaches are different, but we will provide a counter-example to make the case against predicting action sequences.

The ACRO objective optimizes the following

$$\phi_\star \in \underset{\phi \in \Phi}{\arg\max} \; \underset{t \sim U(0,N)}{\mathbb{E}} \; \underset{k \sim U(0,K)}{\mathbb{E}} \; \log\left(\mathbb{P}(a_t \mid \phi(x_t), \phi(x_{t+k}))\right). \tag{6}$$

The $k$ step action sequence prediction approach optimizes:

$$\phi_\star \in \underset{\phi \in \Phi}{\arg\max} \; \underset{t \sim U(0,N)}{\mathbb{E}} \; \underset{k \sim U(0,K)}{\mathbb{E}} \; \log\left(\mathbb{P}(a_t, \ldots a_{t+k} \mid \phi(x_t), \phi(x_{t+k}))\right). \tag{7}$$

### C.1  DETERMINISTIC DYNAMICS

$$\mathbb{P}(a_{t:t+k} \mid \phi(x_t), \phi(x_{t+k})) = \prod_{j=t}^{t+k} \mathbb{P}(a_j \mid \phi(x_t), \phi(x_{t+k}), a_{t:j}) \tag{8}$$

By the assumption of deterministic dynamics in the latent space:

$$\mathbb{P}(a_{t:t+k} \mid \phi(x_t), \phi(x_{t+k})) = \prod_{j=t}^{t+k} \mathbb{P}(a_j \mid \phi(x_t), a_{t:j}, \phi(x_j), \phi(x_{t+k})) \tag{9}$$

After applying the markov assumption as we have assumed an MDP:

$$\mathbb{P}(a_{t:t+k} \mid \phi(x_t), \phi(x_{t+k})) = \prod_{j=t}^{t+k} \mathbb{P}(a_j \mid \phi(x_j), \phi(x_{t+k})) \tag{10}$$

Now we can put this back into the k-step action sequence prediction problem:

$$\phi_\star \in \arg\max_{\phi \in \Phi} \mathbb{E}_{t \sim U(0,N)} \mathbb{E}_{k \sim U(0,K)} \log \left( \prod_{j=t}^{t+k} \mathbb{P}(a_j \mid \phi(x_j), \phi(x_{t+k})) \right) \tag{11}$$

$$\phi_\star \in \arg\max_{\phi \in \Phi} \mathbb{E}_{t \sim U(0,N)} \mathbb{E}_{k \sim U(0,K)} \sum_{j=t}^{t+k} \log \left( \mathbb{P}(a_j \mid \phi(x_j), \phi(x_{t+k})) \right) \tag{12}$$

$$\phi_\star \in \arg\max_{\phi \in \Phi} \mathbb{E}_{t \sim U(0,N)} \mathbb{E}_{k \sim U(0,K)} \mathbb{E}_{j \sim U(t,t+k)} \log \left( \mathbb{P}(a_j \mid \phi(x_j), \phi(x_{t+k})) \right) \tag{13}$$

When $N \gg k$, the distributions of $t$ and $j$ converge, and we can write:

$$\phi_\star \in \arg\max_{\phi \in \Phi} \mathbb{E}_{t \sim U(0,N)} \mathbb{E}_{k \sim U(0,K)} \log \left( \mathbb{P}(a_t \mid \phi(x_t), \phi(x_{t+k})) \right) \tag{14}$$

which we can see is the same as the first-action prediction objective that ACRO optimizes.

## C.2 STOCHASTIC DYNAMICS

If the environment has stochastic dynamics, predicting future actions to reach a goal state without conditioning on the preceding observations is very difficult. This is because what action needs to be taken may depend on what actually happened in the environment. For example, if I'm playing a video game, and there is a small chance that the game pauses for one minute, the actions will need to depend on whether the pause occurred. We can construct a stochastic environment in which every action following the first action is completely unpredictable. We can imagine an environment where there is some information in the observation space which is set randomly on every step and indicates how the agent's controls are randomly permuted for that step. In principle, the first-action predictor can easily use this information to adapt what actions it predicts, and can obtain its original accuracy given sufficient model capacity. On the other hand, the action-sequence predictor ($\mathbb{P}(a_t, \ldots a_{t+k} \mid \phi(x_t), \phi(x_{t+k}))$) will only be able to predict the first action well, and can have no better than uniformly random accuracy at predicting the remaining actions. This is because only $\phi(x_t)$ contains the information about what has happened in the environment which is necessary for control, the history of past actions do not contain the necessary information. In this example, predicting the sequence of actions makes the task much noisier, while providing no additional signal for the model.

## D BENEFITS OF EXOGENOUS INVARIANT REPRESENTATION IN OFFLINE RL

### D.1 PROOF OF PROPOSITION 3.

We need to show that for any $f \in \mathcal{F}(\phi_\star)$ and $x, a$, it holds that
$$R(x,a) + \mathbb{E}_{x' \sim T(\cdot \mid x,a)}[\max_{a'} f(x', a')] \tag{15}$$
is contained in $\mathcal{F}(\phi_\star)$. Since the reward is a function of the agent controller representation only, and since $f \in \mathcal{F}(\phi_\star)$, equation 15 can be written as:
$$R(x,a) + \mathbb{E}_{x' \sim T(\cdot \mid x,a)}[\max_{a'} f(x', a')]$$
$$= R(\phi_\star(x), a) + \mathbb{E}_{x' \sim T(\cdot \mid x,a)}[\max_{a'} f(\phi_\star(x'), a')]$$
$$= R(\phi_\star(x), a) + \mathbb{E}_{x' \sim T(\cdot \mid \phi_\star(x),a)}[\max_{a'} f(\phi_\star(x'), a')]. \tag{16}$$

The first relation holds since $f \in \mathcal{F}(\phi_\star)$ and by the assumption on the reward function (that it is a function of the endogenous states). The second relation holds by

$$\mathbb{E}_{x' \sim T(\cdot|x,a)}[\max_{a'} f(\phi_\star(x'), a')]$$

$$\overset{(a)}{=} \sum_{s',e'} \sum_{x' \in \mathrm{supp}q(x'|s',e')} q(x' \mid \phi_\star(x'), \phi_{\star,e}(x'))T(s' \mid \phi_\star(x), a)T_e(e' \mid \phi_{\star,e}(x))f(s', a')$$

$$\overset{(b)}{=} \sum_{s'} T(s' \mid \phi_\star(x), a)f(s', a') \sum_{e'} T_e(e' \mid \phi_{\star,e}(x))$$

$$\overset{(c)}{=} \sum_{s'} T(s' \mid \phi_\star(x), a)f(s', a'),$$

where $(a)$ holds by the Ex-BMDP transition model assumption,

$$T(x' \mid x, a) = q(x' \mid \phi_\star(x'), \phi_{\star,e}(x'))T(\phi_\star(x') \mid \phi_\star(x), a)T_e(\phi_{\star,e}(x') \mid \phi_{\star,e}(x)),$$

$(b)$ and $(c)$ hold by marginalizing over $x'$ and $e'$. This establishes equation 16 and the proposition: the function $R(\phi_\star(x), a) + \mathbb{E}_{x' \sim T(\cdot|\phi_\star(x),a)}[\max_{a'} f(\phi_\star(x'), a')]$ is contained within $\mathcal{F}(\phi_\star)$ since it only depends on $\phi_\star$.

## D.2 PROOF OF PROPOSITION 4.

Consider an Ex-BMDP with one action $a$ where the agent controller representation is trivial and has a single fixed state (the agent has no ability to affect the dynamics). We will establish a counterexample by constructing a tabular-MDP. Because tabular-MDP is a special case of a more general MDP with continuous states, this will also establish a counterexample for the more general non-tabular setting considered in the paper.

Let the observations and dynamics be given has follows. The observation is a 2-dimensional vector $x = (x(1), x(2))$ where $x(1), x(2) \in \{0, 1\}$. The dynamics is deterministic and its time evoluation is given as follows:

$$x_{t+1}(1) = x_t(1) \oplus x_t(2)$$
$$x_{t+1}(2) = x_t(2),$$

where $\oplus$ is the XOR operation. In this case, the transition model is given by $T(x' \mid x, a) = T(x' \mid x)$ and $\phi_\star = \{s_0\}$ where $s_0$ is a single state; since the observations are not controllable the controller representation maps all observations to a unique state. Further, assume that the reward function is $0$ for all observations.

Assume that $\phi(x) = (x_1)$, i.e., the representation ignores the second feature $x_2$. This representation is more refined than $\phi_\star$ since the latter maps all observations into the same state. Consider the tabular Q function class on top of this representation $\mathcal{Q}_{N=2}$, and consider $Q \in \mathcal{Q}_{N=2}$ given as follows

$$Q(x_1 = 1) = 1$$
$$Q(x_1 = 0) = 0.$$

We now show that $\mathcal{T}Q$ is not contained in $\mathcal{Q}_{N=2}$. According to the construction of the transition model, it holds that

$$(TQ)(x_1 = 1, x_2 = 1) = 0$$
$$(TQ)(x_1 = 0, x_2 = 1) = 1$$
$$(TQ)(x_1 = 1, x_2 = 0) = 1$$
$$(TQ)(x_1 = 0, x_2 = 0) = 0.$$

This function cannot be represented by a function from $\mathcal{Q}_{N=2}$; we cannot represent $(TQ)$ since it is not a mapping of the form $x_1 \to \mathbb{R}$ by the fact that, e.g.,

$$(TQ)(x_1 = 1, x_2 = 1) \neq (TQ)(x_1 = 1, x_2 = 0).$$

Meaning, it depends on the value of $x_2$.

# E  EXTENDED RELATED WORK

**Representation learning in Offline RL**. Representation learning offers an exciting avenue to address the demands of learning compact feature for state by incorporating the auxiliary task of the state feature within the learning task. Empirical studies on representation learning in Offline RL have been first addressed by **?**, which evaluate the ability of a broad set of representation learning objectives in the offline dataset and propose Attentive Contrastive Learning (ACL) to improve downstream policy performance. After that, Chen et al. (2021a) investigate whether the auxiliary representation learning objectives that broadly used in NLP or CV domains can help for imitation across different Offline RL tasks. Lu et al. (2022b) further explores the existing challenges for visual observation input with the Offline RL dataset, meanwhile providing simple modifications on several state-of-the-art Offline RL algorithms to establish a competitive baseline. Another branch of representation learning in Offline RL is theoretical side. Uehara et al. (2021) studies the representation learning in low-rank MDPs with Offline settings and proposes an algorithm that leverages pessimism to learn under a partial coverage condition, Nachum & Yang (2021) develops a representation objective that provably accelerate the sample-efficiency of downstream Offline RL tasks, Ghosh & Bellemare (2020) theoretically shows that the stability of the policy is tightly connected with the geometry of the transition matrix, which can provide stability conditions for algorithms that learn features from the transition matrix of a policy and rewards.

**Offline RL**. The predominant approach to train offline RL agent is regularizing the learned policy to be close to the behavior policy of the offline dataset. This can be implemented by generating the actions that similar to the dataset and restricting the output of the learned policy close to the generated actions (Fujimoto et al., 2019), penalizing the distance between the learned policy and the behavior of the dataset (Kumar et al., 2019; Zhang et al., 2021b), or introducing a pessimism term to regularize the Q function for avoiding high Q value of the out-of-distribution actions (Kumar et al., 2020; Buckman et al., 2021). Some approaches utilize BC as a reference for policy optimization with the baseline methods (Fujimoto & Gu, 2021a; Laroche et al., 2019; Nadjahi et al., 2019; Simão et al., 2020; Rajeswaran et al., 2018). Some other approaches improve the performance by measuring the uncertainty of the model's prediction (Yu et al., 2020; Kidambi et al., 2020; An et al., 2021).

## F    EXOGENOUS INFORMATION DATASETS

In this section, we provide a detailed summary of the different types of exogenous information based datasets as demonstrated in Figure 3. In Figure 8 we show samples from all of the datasets we explored. We further provide details for how each dataset is collected in Appendix G.

### F.1    DATASET DETAILS

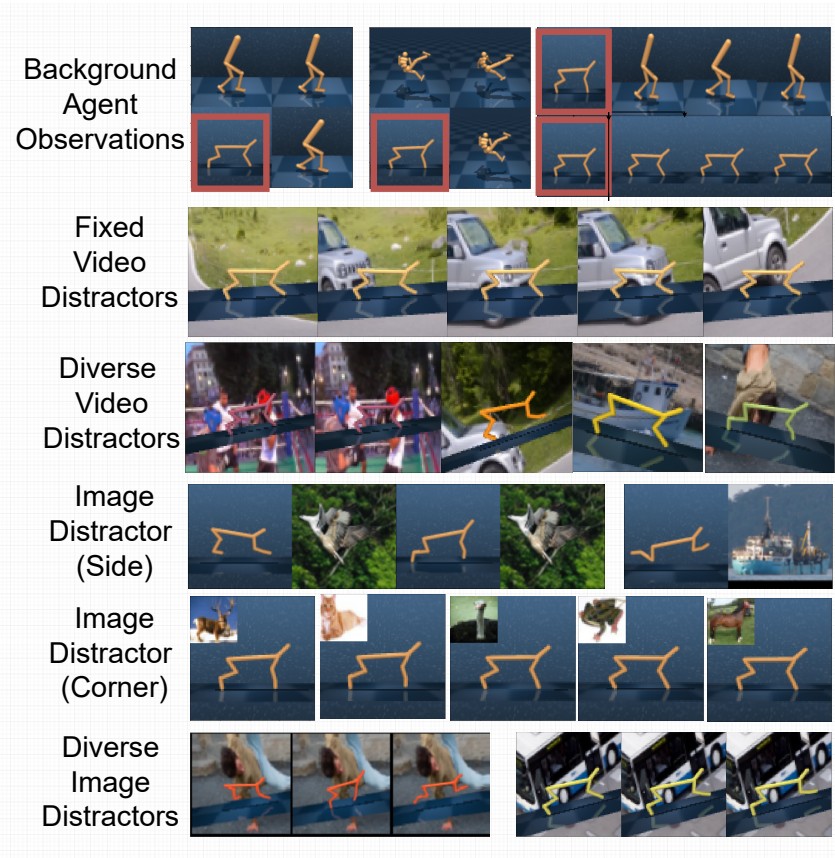

Figure 8: **Summary of Different Exogenous Information Offline Datasets**. In this work, we collected new datasets under different exogenous observations for offline RL. **Top to Bottom**: (a) **Background Agent Observations as Exogenous Information** (first row) showing endogenous controller agent (in red) and other exogenous agent observations taking random actions from other environments. (b) **Fixed and Changing Video Distractors**  (second and third row respectively) where background video distractor changes per episode during offline data collection. (c) **Uncorrelated Image Distractors** (fourth and fifth row) showing that exogenous distactors can either be on the background, on the side or in the corner of the agent's observation space. (d) **Correlated Background Image Distractors** (sixth row) where background image distractor remains fixed per episode during data collection, and only changes per episode, to introduce time correlated exogenous image distractors. We find as the type of exogenous distractor becomes harder, from uncorrelated to correlated exogenous noise, the ability for baselines to learn good policies significantly breaks, as seen from the performance evaluations; whereas ACRO can still be robust to the exogenous information.

**Easy-Exogenous Information (EASY-EXO)**. A visual RL offline benchmark has been recently proposed in (Lu et al., 2022a), where the authors provided pixel-based offline datasets collected using varying degrees of a soft actor-critic (SAC) policy. Furthermore, (Lu et al., 2022a) proposed a suite of distractor based datasets with different levels of severity in distractor shift, ranging from easy-shift, medium-shift to hard-shift. In the EASY-EXO setting, we first consider pixel-based offline

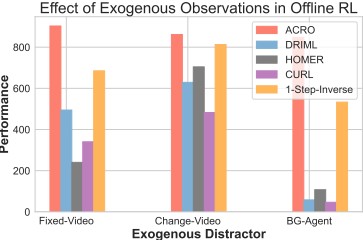 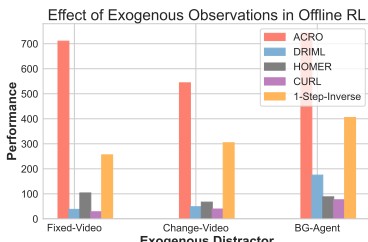

Figure 9: **Summary of Experiment Results on Walker Domain with Expert Dataset:** We vary the type of exogenous distractors present in offline datasets, and evaluate the ability of ACRO for policy learning from provably robust exogenous-free representations, while baseline methods can be prone to the exogenous information present in datasets. From easiest distractor (uncorrelated static images placed in corner or on the side), to corrrelated background exogenous images, and then to fixed or changing video distractors playing in the background, to finally the hardest exogenous information of other random agent information in the agent observation space; we show that ACRO can consistently learn good policies for downstream control tasks, while the ability of baselines to ignore exogenous information dramatically degrades, as we move to hard exogenous information settings. Appendix F.5 provides further ablation studies on the different HARD-EXO offline settings, and performance difference for different domains and datasets.

data without and with visual distractors, as shown in Table 2. Experimental results in Table 2 show that even without any exogenous noise, ACRO learns controller latent state representations more accurately than the different state-of-the-art baselines, such that by efficiently decoupling the endogenous state from the exogenous states, policy learning during the offline RL algorithm can lead to significantly better evaluation performance compared to several other baselines.

**Medium-Exogenous Information (MEDIUM-EXO).** We then consider three different types of medium exogenous information that might appear in visual offline data. To that end, we consider exogenous uncorrelated images from STL-10 image dataset (Coates et al., 2011) that appear on the corner or the side of the agent observation, and the goal of ACRO is to filter out the exogenous information while recovering only the controller part of the agent state. We consider *three different types of* MEDIUM-EXO *information:*,

- The exogenous image from STL10 dataset appears in the corner of the agent observation. This does not change the observation size of the agent; and we simply add the exogenous image in one corner, which is fixed during an entire episode during the offline data collection. Figure 12 summarizes the result with different exogenous images placed in the corner. ACRO consistently outperforms several other baselines for a range of different datasets, since it can suitably filter out the exogenous part of the agent state.

- A slightly more difficult setting where now the STL10 exogenous image appears on the side of the agent observation space. This augments the agent observation space from $84 \times 84 \times 3$ to $84 \times 84 \times 2 \times 3$ since we consider downsampled STL10 images. Figure 13 summarizes this result comparing ACRO with the baseline representation objectives.

- Finally we consider the distractor setting that has appeared in prior works in online RL (Zhang et al., 2021a) with fixed video distractors playing in the background of agent observation space. For this setting, we specifically re-collect the dataset following the procedure in (Lu et al., 2022a) where the SAC data collecting agent also sees a fixed video distractor playing in the background. Figure 14 summarizes the result and shows performance plots where ACRO can significantly outperform all the baselines across all different types of exogenous datasets.

**Hard Exogenous Information (HARD-EXO).** We finally consider three sets of different hard exogenous information settings, and find that these HARD-EXO can remarkably make it difficult for existing state of the art representation objectives to learnt underlying agent controller states. This setting provides evidence that under suitably constructed exogenous information, which appear highly time correlated during the offline data collection, the baseline methods can fail to capture underlying controller latent representation of states. In contrast, the objective we consider in ACRO, along

with the theoretical guarantees for learning a suitable encoder to recover the endogenous states, shows that policy optimization based on the endogenous controller latent states can lead to efficient learning in these control tasks. We consider *three different types of* HARD-EXO *information:*

- We first consider time correlated static images appearing in the background of the agent observation. For this setting, during data collection, the agent sees a fixed image in the background for an entire episode, and it changes per episode of data collection. This time correlated exogenous information ensures that the baselines can remarkably get distracted, while ACRO can still be robust to the static image background. Figure 15 summarizes the results and shows that several existing representation baselines can fail due to time correlated static image distractors.

- We then consider an even more difficult HARD-EXO setting where now the video distractors playing in background also changes per episode during data collection. This is a novel setting with video distractors in RL, since we explicitly consider diverse set of background videos which also changes per episode of data collection. Similar to the above, Figure 16 summarizing the results with diverse video distractors in background per episode, shows that this setting can also break the baseline representation learners to recover the controller latent states, while performance of ACRO remains robust to it, since the ACRO objective learns encoder to recover the endogenous controller latent states accurately.

- Finally, we consider the most challenging HARD-EXO where now in addition to the environment observation, the agent additionally sees other random action agent observations. Here, the goal of the agent is to learn representations to identify the *controllable* environment, while other random-action observations are *uncontrollable* or exogenous to the agent. This is quite a difficult task since the agent we are tring to control also sees observations from the same domain, of other agents playing with random actions. The controller agent observation now consists of other agents placed in a $3 \times 3$ grid. Figure 18 summarizes the experiment results showing that ACRO significantly outperforms all baseline representation learners.

### F.2 EASY-EXO: PIXEL-BASED OFFLINE RL FROM V-D4RL BENCHMARKS

**Visual Offline Control (V-D4RL) without Distractors**. We first verify the effectiveness of learning representations with ACRO without any additional exogenous distractors, and compare with several baselines for learning representations. Figure 10 provides detailed performance curves for Table 2.

**V-D4RL with Varying Severity of Distractor Data Shift**. We then consider the distractor setting in v-d4rl benchmark (Lu et al., 2022a) with varying levels of distractor difficulty. Here, the exogenous noise is based on background static image distractors inducing a distribution shift in the dataset, depending on the level of difficulty from *easy*, to *medium* to *hard* distractors. As shown in Figure 11, we consider two different domains and find that with varying difficulty levels, ACRO can consistently outperform several state of the art baselines, learning directly from pixel data.

### F.3 MEDIUM-EXO: STL10 EXOGENOUS IMAGES OR FIXED VIDEO DISTRACTORS

We extend our experimental results with different types of exogenous image distractors in the observation space of the agent. Detailed description of the dataset collection process is provided in Appendix G.

**Exogenous Image Distractors Placed on the Corner or Side of Agent Observation**. We consider two different settings where the agent environment observation is augmented with STL-10 image Coates et al. (2011) distractors, either placed in the corner or on the side of the environment observation. Here, we consider adding uncorrelated exogenous images where for each pre-training of representations update, the environment observation has image distractors. When placed on the side, it extends the observation space of the agent. Figure 12 and Figure 13 shows results with exogenous images placed in the corner or on the side of the agent observations respectively.

**Fixed Video Distractor**. We first consider a setting where the exogenous distractor in the background is fixed with a single type of video distraction. Figure 14 shows results with fixed video distractor showing that across several datasets, ACRO can consistently outperform baselines.

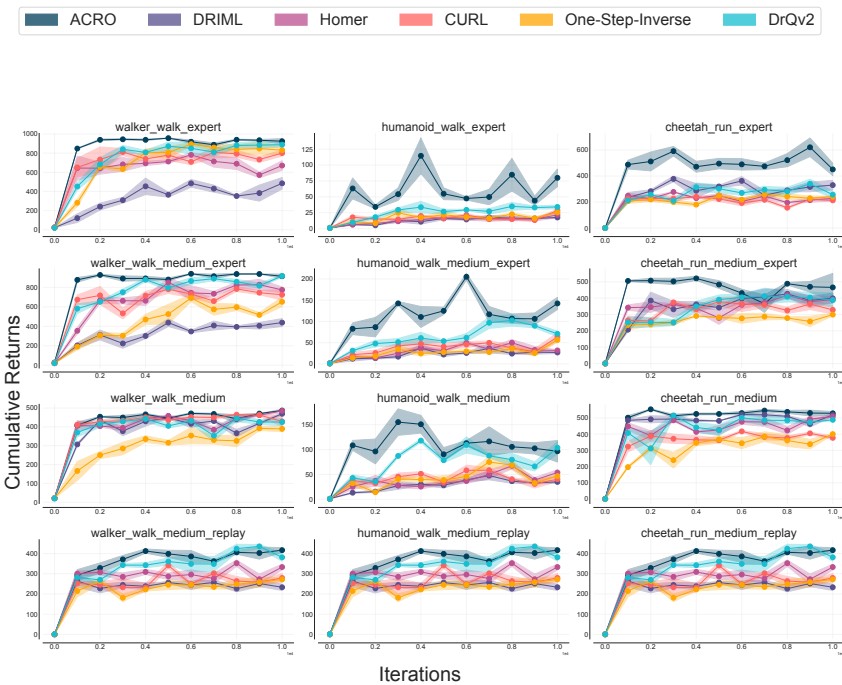

Figure 10: **EASY-EXO-No Distractors Full Results**. Experiments over 6 random seeds. For these experiments, we use the visual offline benchmark from (Lu et al., 2022a) and compare ACRO with several state of the art representation learning objectives. We find that across all tasks, ACRO either outperforms or equally performs as well as the best performing baseline method.

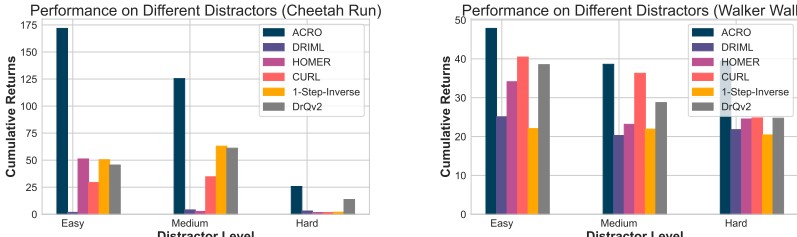

Figure 11: **EASY-EXO-Image Distractors Full Results**. Comparison of ACRO with baselines using the distractor suite of data shift severity from v-d4rl (Lu et al., 2022a) benchmark. We compare results with the two domains and datasets that were released in the v-d4rl benchmark distractor suite.

### F.4 HARD-EXO: TIME CORRELATED AND MOST DIVERSE EXOGENOUS DISTRACTORS

**Static Background Image that Changes Per Episode**. We further experiment with time correlated exogenous distractors in the background, where during every episode of data collection, we provide a background image to the data collecting policy. We find that in presence of exogenous background distractors, ACRO can still be robust to the exogenous noise, while the existing baselines learning representations are more likely to fail, as shown in Figure 15.

**Exogenous Video Distractors that Changes Per Episode**. We then consider a more difficult setting where the type of video distractor playing in background changes during every episode. Further details on the data collection with *fixed* and *changing* video distractor in background is provided in the appendix. Figure 16 shows results with changing video distractor, where ACRO can consistently outperform baselines representation learning methods.

**Multi-Environment Agent Observations as Exogenous Information**. We then consider a setting where the observation space of the agent is augmented with other random agents moving in the

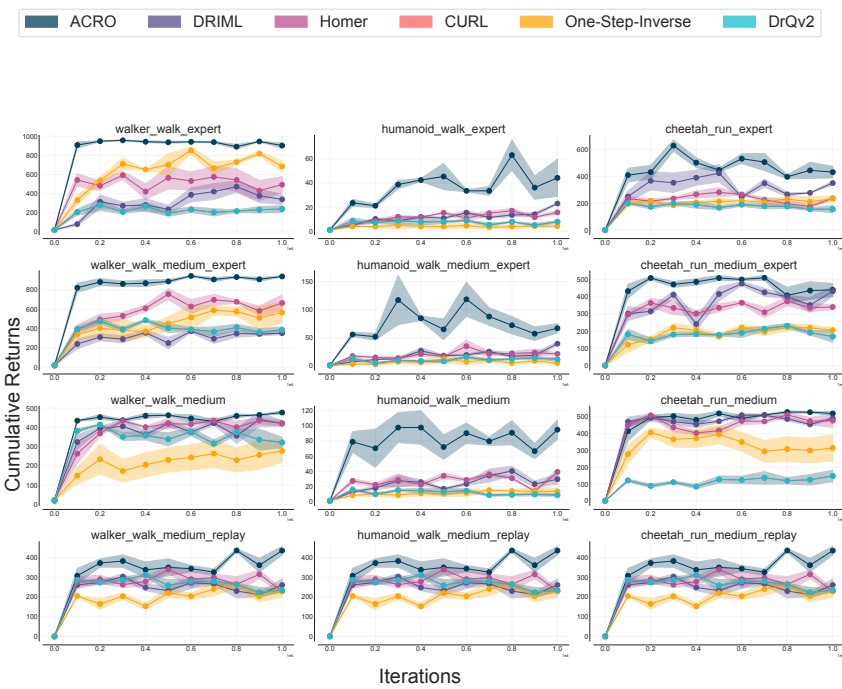

Figure 12: **MEDIUM-EXO-Corner Full Results**. Performance comparison of ACRO with several other baselines.

environment. For this experiment, we take the observations from the environment the agent wants to control, while the other observations of other agents come from a random-policy dataset, either from the same domain, or from different domains. We do this since the random observations from other agents can be treated as exogenous information to the representation encoder; our experiment results show that ACRO can ignore the exogenous information from other agents, leading to significant performance improvements from the offline RL algorithm based on the exogenous observation dataset. Figure 18 summarizes the results.

### F.5 ABLATION STUDIES - HARD-EXO OFFLINE RL

Figure 17 provides a summary of comparison between different datasets in the HARD-EXO noise setting.

## G DATA COLLECTION FOR OFFLINE RL WITH EXOGENOUS INFORMATION

**EASY-EXO Datasets**. For the EASY-EXO setting with exogenous information, we consider uncorrelated visual distractors in the background of the observation space. For this setting, we extensively use the datasets released from the v-d4rl benchmark (Lu et al., 2022a) for offline RL. We note that the data shift severity in v-d4rl benchmark are only limited to two different domains and two data distributions (medium-expert and random). We experiment with both these datasets, and additionally consider the setting with no uncorrelated static images in the background.

**MEDIUM-EXO Datasets**. For the MEDIUM-EXO datasets, we collect new offline datasets using a SAC policy, following the same data collection procedure as in the literature from d4rl benchmark (Fu et al., 2020). The main difference is that when collecting new datasets with the SAC policy, we consider variations of different exogenous noise types in the dataset. As discussed earlier, for the MEDIUM-EXO setting, we collect three different exo-types of datasets : **(a)** Exogenous stl10 images placed in the **corner** of the agent observation space. For this setting, during an episode of data collection, at each time step, we sample a new STL-10 image and place in the corner of the agent observations. **(b)** In the **side** exogenous information setting, instead of the corner, we place

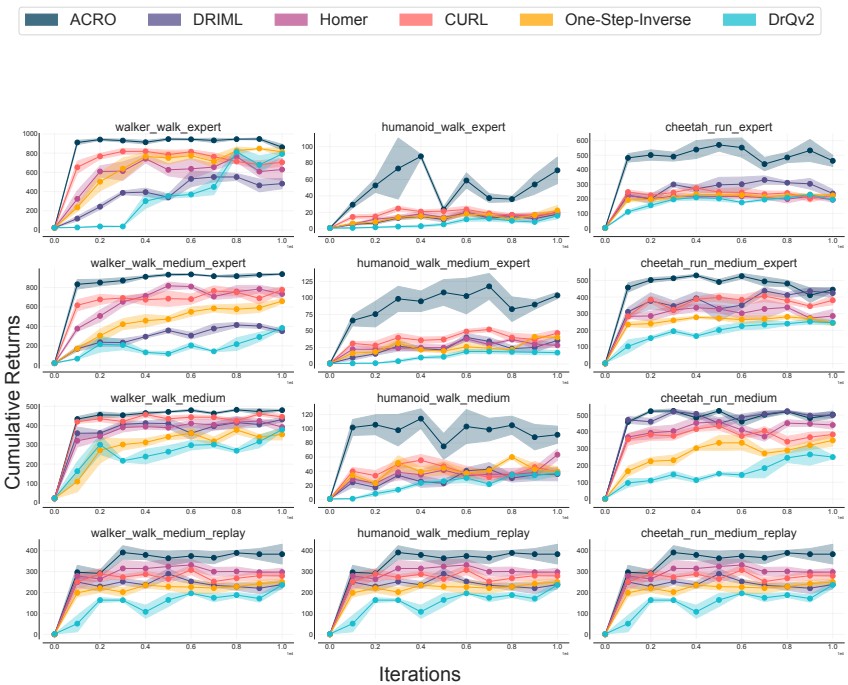

Figure 13: **MEDIUM-EXO-Side Full Results** ACRO can be quite robust to the exogenous images when the exogenous images appear to be similar to the agent in the environment. For example, consider the Cheetah-Run environment with a dog run image on the side, which can be quite distracting to the baseline methods.

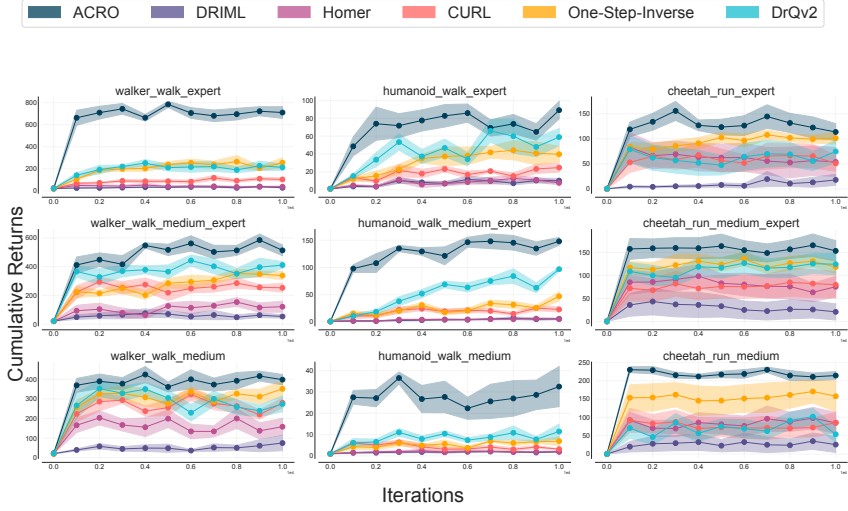

Figure 14: **MEDIUM-EXO-Fixed Video Full Results**. ACRO can outperform baselines with fixed exogenous distractors in background, which is time-correlated in nature. The fixed video distractor setting have often been studied in online RL literature. In this work, we study fixed video distractors in offline RL, where the distractors were present during the offline data collection. .

the exogenous image on the side of the environment observation. This augments the entire agent observaton space. A major difference with (a), however is that, in this setting we consider time correlated exogenous images where each step of SAC policy during an episode sees the same exogenous image, which only changes per episode of data collection. We consider this to be a harder setting compared to changing the exo image at each time-step, since this induced time correlation

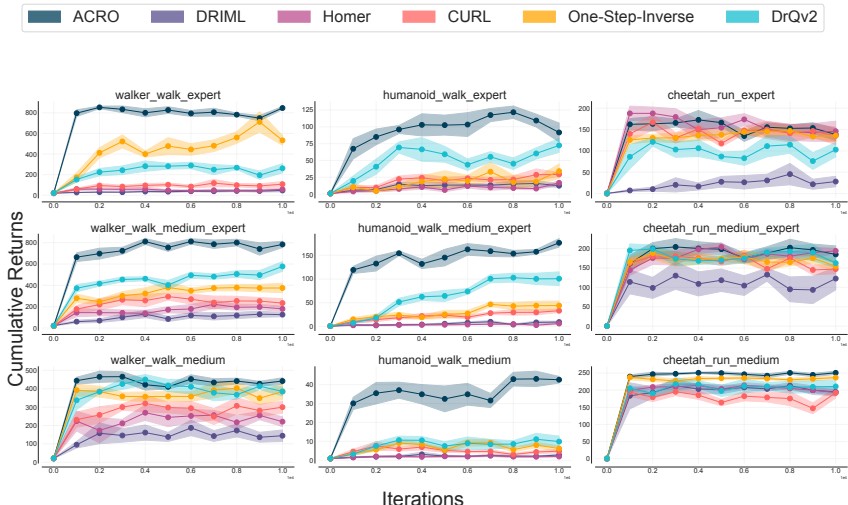

Figure 15: **HARD-EXO-Static Image Full Results**. We find that ACRO can significantly outperform baselines in presence of correlated exogenous static images playing in the background

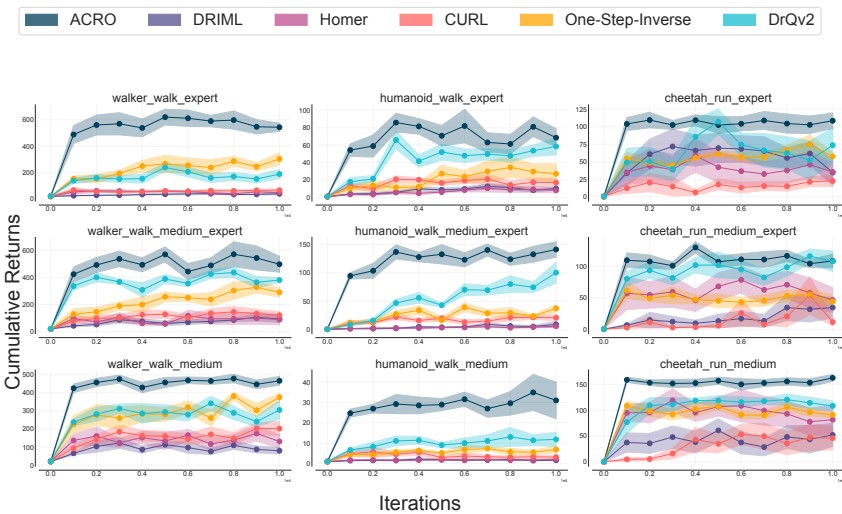

Figure 16: **HARD-EXO-Video Full Results** Across all datasets, ACRO can consistently outperform baseline methods for all types of datasets. When learning representations for offline policy optimization, the ability of ACRO to ignore exogenous information makes it outperform baseine representation objectives in almost all cases. The changing and time correlated video distractors are often hard for baseline methods to ignore, leading to significant performance drops depending on the offline data distribution

can make it harder for the representation objectives to be completely robust to the side exogenous information. **(c)** Finally, we conisder a **fixed video** distractor setting, which has been extensively studied in the online control benchmark (Tassa et al., 2018). Prior works have experimented in the online setting with fixed video distractors. We use the same procedure and fixed video distractor as here, except we use a SAC policy for data collection to be used in the offline setting. All these categories, cumulatively are denoted as MEDIUM-EXO in this work. We release datasets and detailed experiment details for all these settings for MEDIUM-EXO based offline RL datasets.

**HARD-EXO Datasets**. We consider this to be the hardest of the exogenous distractor setting. For this setting, we introduce several new offline benchmarks with different types of exogenous information. **(a) Time Correlated Exogenous Image in the Background** This is a hard distractor type

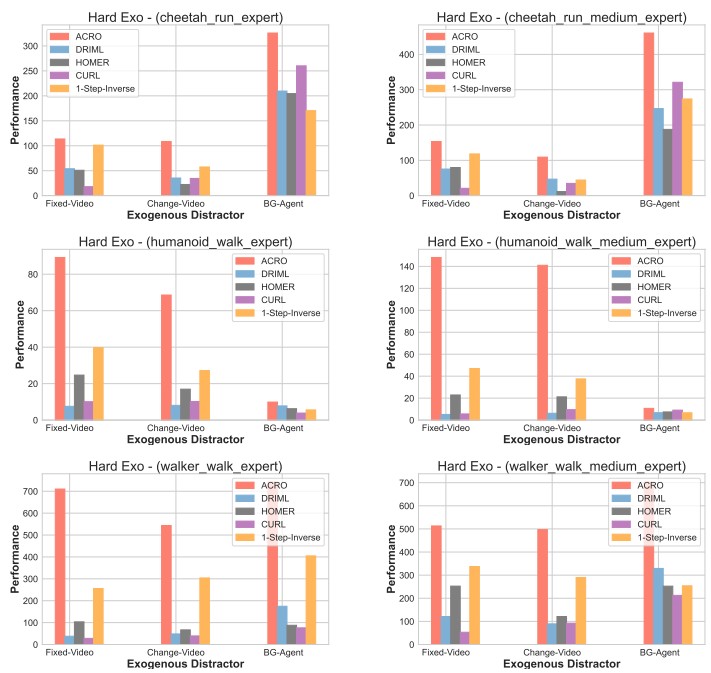

Figure 17: **Summary and Performance Difference between ACRO and baselines in the HARD-EXO offline RL setting**

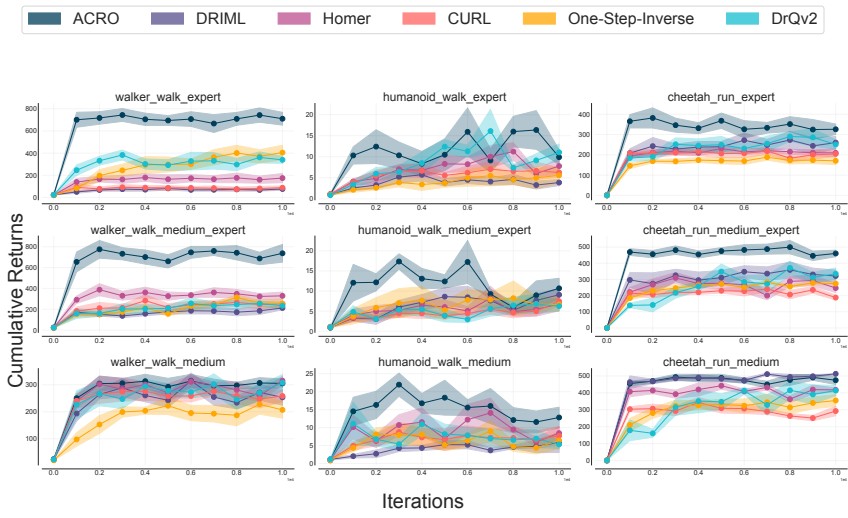

Figure 18: **HARD-EXO-Background Agent Full Results**. Random agent observations placed on the grid, of the entire agent observation space.

where most existing representation learning baselines can fail. For this, during the data collection with SAC agent, we place a static background image from the STL10 dataset in the background. This image remains fixed for all timesteps within an episode, and we only sample a new exogenous background image at every episode. This makes the exogenous noise to be highly time correlated, where baseline representations are likely to capture the background image in addition to learning an embedding of the environment. **(b) Changing Video Distractors** In this setting, like (a), at every new episode we change the *type* of video distractor that we use. During an episode for different timesteps, the agent already sees correlated noise from the video frames playing in the background. However, since the type of video sequence that plays in background changes, this makes the offline datasets even more difficult to learn from. This setting is inspired by a real world application, where

for example, the agent perceiving the world, can see background data from different data distributions (e.g background moving cars compared to people walking in the background). **(c)** Finally, we consider another HARD-EXO dataset, where now during every timestep of data collection, the agent sees other agents playing randomly. Here, we place several other agents, taking random actions, in a grid, and the goal of the agent is to recover only the controllable agent, while being able to ignore the other uncontrollable agents taking random actions, within its observation space. For this setting, the other uncontrollable agents can either be from the same domain (e.g using a Humanoid walker agent for both the endogenous, controllable part and the exogenous, uncontrollable part), or a different setting where the exogenous agents can be from other domains (e.g using a Humanoid agent for the controllable part, while the uncontrollable agents are from a Cheetah agent). We demonstrate both these types of observations either same-exogenous or different-exogenous in Figure 21 and Figure 20 respectively. We also release these datasets as an additional contribution to this work, and hope that future works in offline RL will use these datasets as benchmarks, for learning robust representations.

## H ADDITIONAL EXPERIMENT RESULTS: ATARI MEDIUM-EXO

We also consider the setting of Atari. We build on the setup introduced in decision transformer (Chen et al., 2021b). While decision transformer focuses on framing the reinforcement learning problem as a sequence modeling problem, we mainly focus on learning representations which can learn to ignore exogenous noise. We use the same 4 games used in Chen et al. (2021b) - Pong, Qbert, Breakout, and Seaquest. We consider the MEDIUM-EXO setting where a different image is used in each episode and concatenated to the side of each observation. We add

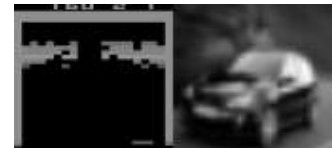

Figure 19: **Example** of an observation from Breakout with a CIFAR10 image on the side.

randomly sampled images from the CIFAR10 dataset (Krizhevsky & Hinton, 2009) as exogenous noise. Figure 19 shows an example observation from breakout with exogenous noise.

Decision Transformer uses the DQN-Replay dataset (Agarwal et al., 2020) for training. The model is trained using a sequence modeling objective to predict the next action given the past states, actions, and returns-to-go $\hat{R}_c = \sum_{c'=c}^{C} r_c$, where $c$ denotes the timesteps. This results in the following trajectory representation: $\tau = \left( \hat{R}_1, s_1, a_1, \hat{R}_2, s_2, a_2, \hat{R}_3, s_3, a_3, \dots \right)$, where $a_c$ denotes the actions and $s_c$ denotes the states. At test time, the start state $s_1$ and desired return $\hat{R}_1$ is fed into the model and it autoregressively generates the rest of the trajectory.

Decision Transformer uses a convolutional encoder to encode the observations. We first pretrain this encoder using the proposed ACRO objective. We use the 1-step inverse objective and DRIML as our baselines. After pretraining, we train the decision transformer using the sequence modeling objective

Table 4: **Atari (MEDIUM-EXO)**. Here we compare ACRO to One-Step-Inverse model and DRIML on various games from the atari benchmark with exo-noise. We can see that ACRO consistently outperforms both the baselines. Results averaged across 5 seeds.

| Game | 1-Step Inv. | DRIML | ACRO |
|---|---|---|---|
| Breakout | $3.8_{\pm 0.4}$ | $1.0_{\pm 0.0}$ | $20.6_{\pm 3.2}$ |
| Pong | $8.6_{\pm 3.2}$ | $-20.0_{\pm 0.0}$ | $11.8_{\pm 3.37}$ |
| Qbert | $536.2_{\pm 233.75}$ | $277.8_{\pm 46.24}$ | $657.4_{\pm 271.52}$ |
| Seaquest | $274.0_{\pm 29.61}$ | $94.4_{\pm 4.63}$ | $972.4_{\pm 136.09}$ |

keeping the encoder fixed. We present results in Table 4. We can see that ACRO outperforms both the baselines in all games further showing the effectiveness of the proposed approach.

**Hyperparameter Details**. We keep most of the hyperparameter details same as used in Chen et al. (2021b). They use episodes of fixed length during training - also referred to as the *context length*. We use a context length of 30 for Seaquest and Breakout and 50 for Pong and Qbert. Similar to Chen et al. (2021b), we consider one observation to be a stack of 4 atari frames. To implement the ACRO objective, we sample 8 different values for $k$ and calculate the objective for each value of $k$, obtaining the final loss by taking the sum across all the sampled values of $k$. We do not feed the embedding for $k$ in the MLP that predicts the action while computing the ACRO objective.

## I  EXPERIMENT SETUP AND DETAILS

We describe our experiment setup in details. We use the visual d4rl (v-d4rl) benchmark Lu et al. (2022a) and additionally add exogenous noise to visual datasets, as described earlier. For all our experiments, comparing ACRO with other baseline representation objectives, we pre-train the representations for $100K$ pre-training steps. Given pixel based visual offline data, we use a simple CNN+MLP architecture for encoding obeservations and predicting the ACRO actions. We also use cropping-based data augmentation as in DrQv2 while pre-training the representations for all methods. Specifically, the ACRO encoder uses 4-layers of convolutions, each with a kernel size of 3 and 32 channels. The original observation is of $84 \times 84 \times 9$, corresponding to a 3 channel-observation and a frame stacking of 3. The final encoder layer is an MLP which maps the convolutional output to a representation dimension of 256, giving the output $\phi(x)$. This is followed by a 2-layer MLP (hidden dim-256) that is used to predict the action given a 512 input corresponding to a concatenated $s_t$ and $s_{t+k}$ representations. For ACRO, we sample $k$ from 1 to 15 uniformly. We use ReLU non-linearity and ADAM for optimization all throughout.

For our experiments, we build off from the open source code base accompanying the v-d4rl benchmark (Lu et al., 2022a). We implement the pre-trained representation objectives in a model-free setting, where for the baseline offline RL algorithm we use **TD3 + BC** (Fujimoto & Gu, 2021b) since it achieves state of the art performance in raw state based offline RL benchmarks (Fu et al., 2020) and is a reasonably performing algorithm for visual offline RL as well Lu et al. (2022a). The policy improvement objective for the baseline **TD3 + BC** algorithm is thus:

$$\pi = \arg\max_{\pi} \mathbb{E}_{(\mathbf{s}_t, \mathbf{a}_t) \sim \mathcal{D}_{\text{env}}} \left[ \lambda Q(\mathbf{s}_t, \pi(\mathbf{s}_t)) - \left( \pi(\mathbf{s}_t) - \mathbf{a}_t \right)^2 \right] \tag{17}$$

where the critic $Q(s, \pi(s))$ is evaluated by a TD loss, and we use re-paramterized gradients through the critic for policy improvement step. For pixel based visual observations, recent work Lu et al. (2022a) proposed the **DrQ + BC** algorithm, which is essentially the TD3 + BC algorithm, except it additionally applies the data augmentations on pixel based inputs. In DrQv2, data augmentation is applied only to the images sampled from the replay buffer, and not during the sample collection procedure. Given the pixel based control tasks, where the images are $84 \times 84$, DrQ pads each side by 4 pixels (repeating boundary pixels) and then selects a random $84 \times 84$ crop, yielding the original image, shifted by 4 pixels. This procedure is repeated every time an image is sampled from the replay buffer; and makes DrQ data augmentation quite effective based on the random shifts alone, without the need for any additional auxiliary losses. More concretely, denoting policy as $\pi_\theta$, the policy network is trained with the following loss :

$$L_\theta(\mathcal{D}) = -\mathbb{E}_{x_t \sim \mathcal{D}} \left[ Q_\psi(\mathbf{z}_t, \tilde{a}_t) \right] \tag{18}$$

where $\mathbf{z}_t = f_\epsilon(\text{aug}(x_t))$ is the encoded augmented visual observation, $\tilde{a}_t = \pi_\theta(\mathbf{z}_t) + \epsilon$ (action with clipped noise to smooth targets, $\epsilon \sim \text{clip}(\mathcal{N}(0, \sigma^2), -c, c)$). Overall loss for policy improvement :

$$\mathcal{L}_\theta(\mathcal{D}) = -\mathbb{E}_{x_t, \mathbf{a}_t \sim \mathcal{D}} \left[ \lambda Q_\psi(\mathbf{z}_t, \mathbf{a}_t) - (\pi_\theta(\mathbf{z}_t) - \mathbf{a}_t)^2 \right] \tag{19}$$

where DrQ passes the gradients of the critic to learn the encoder, and there are no separate or explicit representation losses other than the critic estimation, for training the encoder in DrQ. For all our experiments, in the EASY-EXO setting, we use datasets provided from v-d4rl benchmark (Lu et al., 2022a). For the MEDIUM-EXO and HARD-EXO settings, we follow the procedure from (Fu et al., 2020; Lu et al., 2022a) to collect new datasets using a soft actor-critic (SAC) policy, under different variations of exogenous information that we considered in this work.

## J  BACKGROUND AGENT CONSECUTIVE FRAMES VISUALIZATION

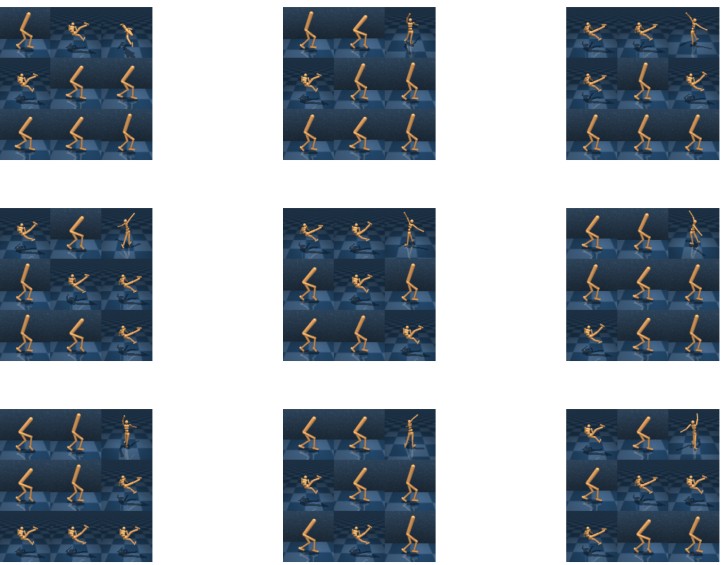

Figure 20: Consecutive timesteps from three different episodes (each row); where the controllable agent and the background agents are from different domains

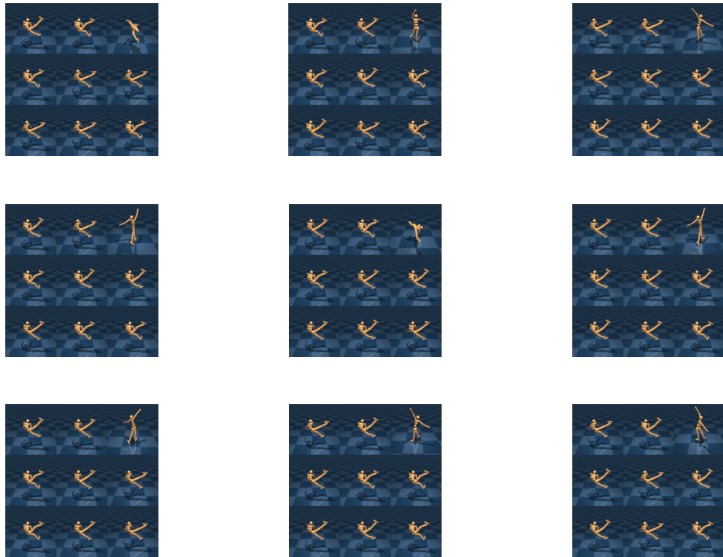

Figure 21: Consecutive timesteps from three different episodes (each row); where the controllable agent and the background agents are from same domains

## K    VISUALIZING RECONSTRUCTIONS FROM THE DECODER

We show additional reconstructions for ACRO, DrQ, behavior cloning, and random features on different types of exogenous noise in Figure 22, Figure 23, and Figure 24.

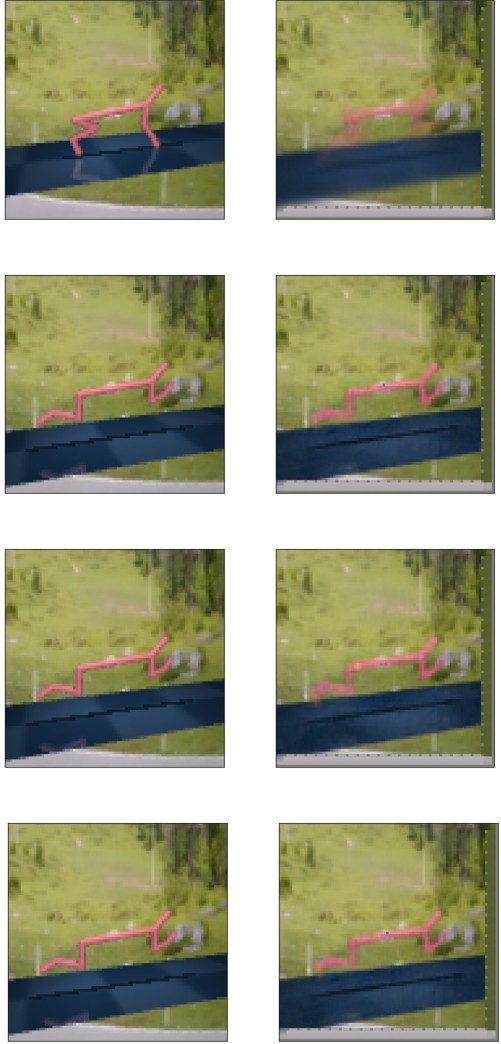

Figure 22: **Reconstructions for Fixed Background Distractor** from a decoder learnt of over two kinds of representations: **Top-Bottom**: Random features, DrQ, Behavior cloning, and ACRO. **Left Column**: Original observation, **Right column**: Reconstruction.

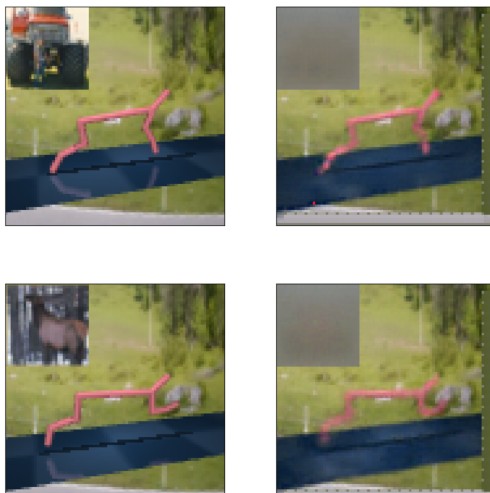

Figure 23: **Reconstructions with CIFAR images in background** from a decoder learnt of two kinds of representations: **Top-Bottom**: ACRO, DrQ. **Left Column**: Original observation, **Right column**: Reconstruction.

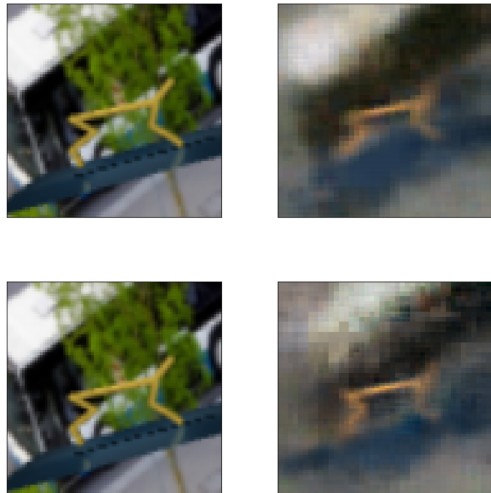

Figure 24: **Reconstructions with a distractor video** playing in the background from a decoder learnt of two kinds of representations: **Top-Bottom**: BC, ACRO. **Left Column**: Original observation, **Right column**: Reconstruction.

# Author Rebuttal

We thank all the reviewers for helpful and detailed feedback, which we have already taken advantage of to improve the paper. We thank the reviewers for praising the strong experimental results, new theory that this paper provides, along with an important contribution to the community by providing new offline RL datasets containing varying levels of exogenous information in the datasets. We plan to release these datasets, along with our implementation, learning robust representations in offline RL, which we hope will be significant to the RL community.

**Note** : After the rebuttal period, we will merge the responses and figures from the sections below into the main draft. Currently, we include these as separate sections to address the reviewer comments explicitly.

## L  COMMON RESPONSE TO ALL REVIEWERS

We first provide a common response to all reviewers, since some of this feedback is useful for overall improvement and understanding of the paper.

### L.1  COMPARISON WITH AC-STATE ( LAMB ET AL. (2022)) AND ACRO, NOVELTY OF USING INVERSE MODELS

We would like to thank the reviewers for highlighting the need to produce comparisons between our proposed method and the AC-State algorithm (Lamb et al., 2022). The reviewers rightly pointed out, as we already addressed in the paper, the similarity with the AC-State objective from Lamb et al. (2022) and ACRO. We would like to highlight here the main differences between AC-State and ACRO and what makes ACRO critical for learning robust representations that can be used for any RL algorithm. The AC-State objective uses a multi-step inverse model with an additional conditioning on the k-th time-step $\mathbb{E}_{t \sim U(0,N)} \mathbb{E}_{k \sim U(0,K)} \log \left( \mathbb{P}(a_t \mid \phi(x_t), \phi(x_{t+k}), k) \right)$, unlike ACRO objective as in equation 1. We would also point out that the theoretical support for multi-step inverse models for representation learning largely comes from Efroni et al. (2021), an ICLR2022 paper which predates the AC-State paper.

The work of Lamb et al. (2022) primarily has a different purpose and scope, where the goal is to recover latents with perfect accuracy, for then using it for exact planning in a tabular MDP, an approach which only scales to very small discrete systems. Most importantly, Lamb et al. (2022) learns 100 discrete latent states using controlled exploration, by the use of a discretization bottleneck Van Den Oord et al. (2017) and primarily addresses recovering discrete ground truth latent states. Lamb et al. (2022) considers only environments with well under 100 discrete endogenous states. In comparison, ACRO considers offline RL with general continuous hidden states, which is much more general. Lamb et al. (2022) does not address how the learnt representation can be used in any RL algorithm, and the controlled exploration is mostly required to recover the ground truth states. The discrete bottleneck along with the requirement for exploration (Lamb et al., 2022) are not generally applicable for any deep RL algorithm to be used off-the-shelf with learnt representations. Lamb et al. (2022) considered moderate state space sizes and used planning on top of the latents. In contrast, we consider larger scale problems and train an RL algorithm on top of the representation. Hence, unlike prior works, we study larger scale problems in offline deep RL.

In contrast, ACRO addresses how to effectively use the multi-step inverse objective in situations where there are infinite continuous latent states and uncontrolled exploration. ACRO shows the capability of learning robust representations in presence of exogenous noise, that can be used for learning continuous representations, and does not require any use of a discretization bottleneck. While the objective remains the same, the data and representations here differ in critical ways, greatly broadening potential applications. Together these changes provide results in regimes the AC-state paper does not address. In ACRO, the focus shifts from doing exact planning (DP or Dijkstra) in a small tabular-MDP to extracting a representation that enables learning a policy via Offline-RL with a learned value function.

In figure 25 and 26, we empirically compare ACRO and a version of ACRO with a discretized bottleneck representation, an extension of Lamb et al. (2022) that can be used in an offline RL

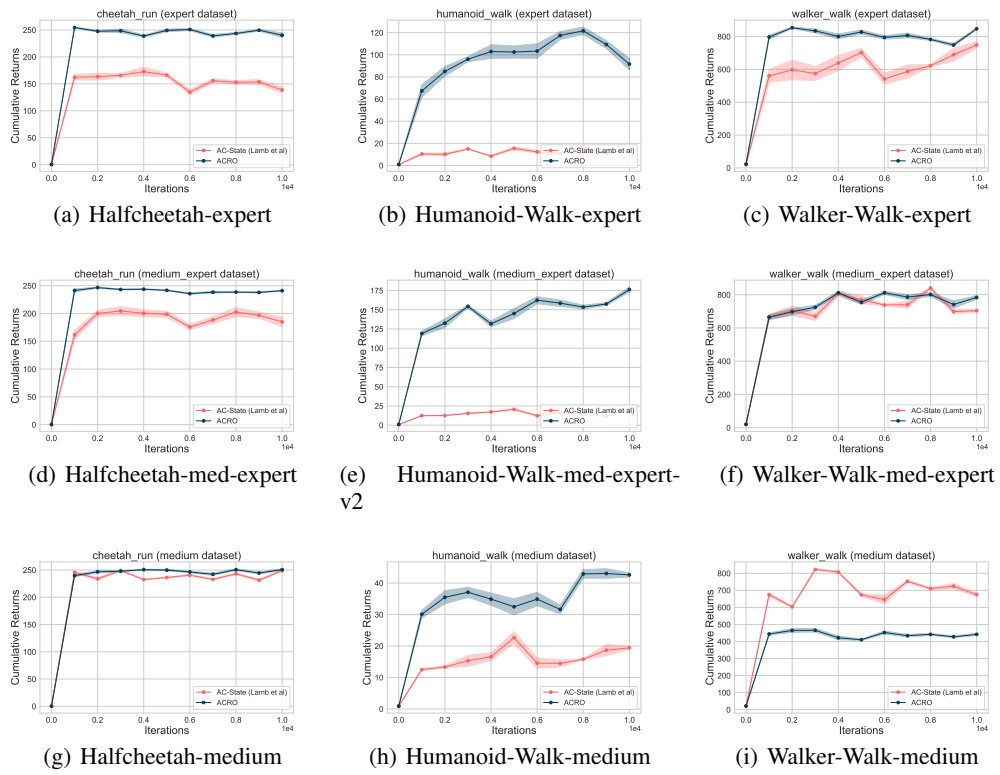

Figure 25: Comprisons between ACRO and AC-State Lamb et al. (2022) in **time-correlated** HARD-EXO offline datasets of different data distributions (expert, medium-expert and medium)

algorithm. We compare both these methods under two of the hardest HARD-EXO offline datasets of varying data distributions. The primary difference is the use of the small discrete representation required by AC-State. In most cases, ACRO works significantly better in the offline RL task with HARD-EXO noise.

**Theoretical Claims :** We see our work as a bridge between earlier theoretical work on multi-step inverse models and the Offline-RL literature. In terms of the novelty of theory, it is true that the proof in Appendix A is the same as these papers, which we already stated very clearly in our paper: "This proof is essentially the same as lemmas found Lamb et al. (2022), Efroni et al. (2021), but is presented here for clarity.". The proofs in Appendix B and Appendix C are novel and specifically related to conditions for Offline-RL to succeed. To better explain the differences, we added a table with these prior works, and we added clearer explanations of novelty to the main text.

We established new results for the setting of RL with exogenous information. Namely, that including some exogenous information as part of the representation may violate Bellman completeness, i.e., the ability to optimally represent any possible Q function that should be learned during the training process. Additionally, we showed that the agent controller representation is sufficient to represent the optimal Q function and induces a Bellman complete class. We first showcase the challenge of the offline RL problem with exogenous noise, and then show the advantage of the agent controller representation.

**Novelty of Using Inverse Models :** We highlight here that even though inverse dynamics models have been used previously in the literature, they are primarily used for either exploration solely (Pathak et al., 2017) or for representation learning in an online task that requires exploration (Misra et al., 2020; Efroni et al., 2021; Lamb et al., 2022). In contrast, in this work, we first use inverse dynamics models in exogenous block MDPs to show the capability of learning robust representations that can then be integrated to an offline RL algorithm. As such, ACRO does not require any

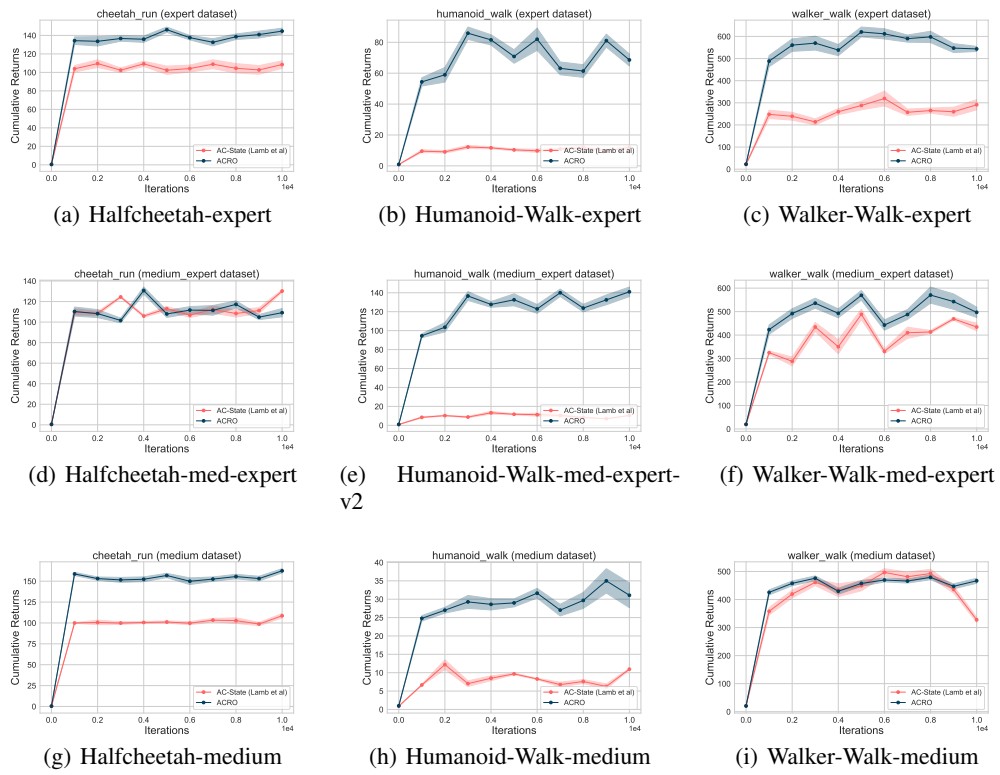

Figure 26: Comprisons between ACRO and AC-State Lamb et al. (2022) in **changing video** HARD-EXO offline datasets of different data distributions (expert, medium-expert and medium)

additional exploration and can work with any data distribution in the offline datasets. We already show results comparing ACRO to one-step inverse models (Pathak et al., 2017).

## L.2 SIGNIFICANCE OF MULTI-STEP ACTION PREDICTION VS. SINGLE ACTION PREDICTION

We thank the reviewers for asking insightful questions about how the ACRO algorithm compares if we use multi-action prediction up to the k-th timestep, in comparison to only predicting a single next step action. Since ACRO already conditions on $\phi(x_{t+k})$ for k-th step in the future, it is natural to ask how the performance varies if we predict multiple future actions compared to a single action. Concretely, the ACRO objective optimizes the following

$$\phi_\star \in \arg\max_{\phi \in \Phi} \mathop{\mathbb{E}}_{t \sim U(0,N)} \mathop{\mathbb{E}}_{k \sim U(0,K)} \log\left(\mathbb{P}(a_t \mid \phi(x_t), \phi(x_{t+k}))\right). \quad (20)$$

whereas, we could instead predict the $k$ step action sequence:

$$\phi_\star \in \arg\max_{\phi \in \Phi} \mathop{\mathbb{E}}_{t \sim U(0,N)} \mathop{\mathbb{E}}_{k \sim U(0,K)} \log\left(\mathbb{P}(a_t, \ldots a_{t+k} \mid \phi(x_t), \phi(x_{t+k}))\right). \quad (21)$$

where equation 21 is implemented using an LSTM that outputs the $k$ actions. One reason to prefer predicting just the first action is that it is a simpler model and is computationally cheaper (as it requires just a classifier over actions and not an autoregressive model over sequences like the LSTM). Intuitively, we also felt that predicting the first action to reach a goal would be sufficient, because ultimately every action along the trajectory is still predicted, but conditioned on the observation prior to the action being taken. In an environment with stochastic dynamics, we see this as being better in principle, because the best action to take at a given step is dependent on what has happened in the environment. In a deterministic environment, both approaches are valid in principle.

Nonetheless, we agree that it is important to also answer this question experimentally. Figures 27 and 28 show comparison between ACRO and a variation of ACRO predicting multiple actions in

the future. We use the same training setup, where all the representations are pre-trained in presence of HARD-EXO noise in observations.

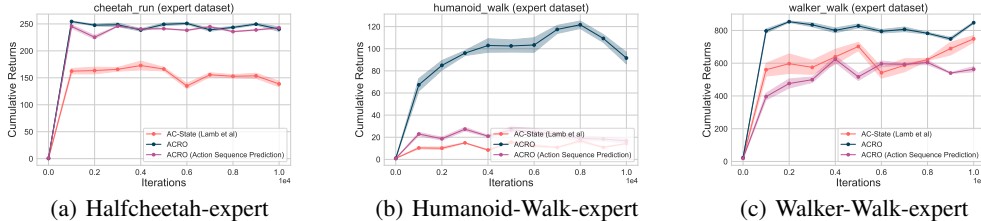

(a) Halfcheetah-expert      (b) Humanoid-Walk-expert      (c) Walker-Walk-expert

Figure 27: Comparions between ACRO, ACRO with multiple action predicton (ie, predicting an action sequence) and also with AC-State (Lamb et al., 2022) in **time-correlated** HARD-EXO offline datasets

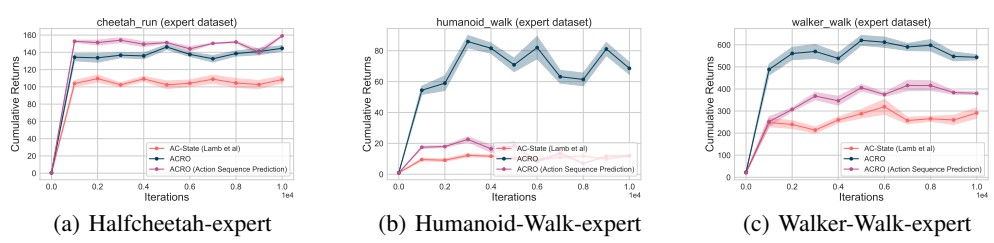

(a) Halfcheetah-expert      (b) Humanoid-Walk-expert      (c) Walker-Walk-expert

Figure 28: Comparions between ACRO, ACRO with multiple action predicton (ie, predicting an action sequence) and also with AC-State (Lamb et al., 2022) in **changing video** HARD-EXO offline datasets

### L.3 DOES ACRO ALSO REMOVE TASK RELEVANT INFORMATION?

Various reviewers expressed concerns about the agent-controller representation ignoring information that is vital to the task (*pedestrians* for reviewer GMBs and *objects / blocks in a robot hand manipulation task* for reviewers GMBs and yx3D). However, the notion of what is in the agent-controller representation is actually more extensive than might be imagined at first glance. Anything that influences the actions taken by the policy needs to be captured by the representation in order to predict the first action it took from a pair of representations. Going back to the examples above, in the case of pedestrians, the policy used to collect the data (hopefully) avoided the pedestrians to reach its goal. In order to properly predict this avoidance behavior (compared to presumably moving in a straight line in the absence of pedestrians), the representation needs to encode the presence of these pedestrians. Similarly, for the example of a robot hand grabbing a block, let us consider a pair of states $(s_t, s_{t'})$, where $s_t$ corresponds to the state before the object was grasped and $s_{t'}$ after. Then, the agent-controller representation needs to include information that pertains to the block's position and orientation, otherwise, it would be impossible to predict the actions governing the robot's motion (presumably in the direction of the object), or how its joints are adjusting to grab the object. Note that this would not be required for a simple one-step inverse model (which justifies the use of a multi-step one) as the robot hand's joint positions in successive states are sufficient to infer what action was taken.

Additionally, in ACRO we also consider expert policies as a source of offline training data. In the case of an expert policy, the knowledge about what parts of the world matter will be captured even by a behavioral cloning policy. ACRO inherits this strength of behavioral cloning, while dramatically outperforming it when the data comes from a random policy.

### L.4 EXPLANATION OF NORMALIZED RESULT FIGURES

We apologize that the results presented in our paper, given we had too many results to present, might have been confusing for reviewers to interpret. In offline RL, since our experiments are across several domains (cheetah, walker, humanoid), and for each domain, we have different dataset distributions (expert, medium-expert, medium) as generally present in offline benchmarks (Lu et al., 2022a; Fu et al., 2020), we *averaged* or *normalized* results to present the significance of ACRO across the experiments. Given the experiments, whether we are considering MEDIUM-EXO or HARD-EXO, we either average the results across domains or datasets. All individual experiments are run for 5 different seeds across the board. We emphasize that the normalization or averaging is done so we could present all our experimental results, explaining significance of ACRO in a precise way, across the board. All individual results showing improvements that ACRO achieves are included in the appendix.

Figure 4 for example, presents results where averaging is being made across domains. We further include individual results for each domain in the appendix. We show that ACRO achieves improvements across all domains in general, in addition to improvements across all dataset types, as shown in figure 4.

Figure 6 shows results where the normalization is being made across the dataset types, instead of domains, as in figure 4. Figure 6 shows results for the 3 domains (cheetah, humanoid, walker) for different HARD-EXO types, and averaged across expert, medium-expert and medium datasets. We show that ACRO can achieve significantly better performance compared to 5 other baselines considered in this work.

### L.5 IMPACT IN OFFLINE RL AND SIGNIFICANCE OF THE WORK

We would like to emphasize and repeat here, to the reviewers, the significance of our work and the impact it can have in the community. Firstly, we propose an approach for learning robust representations, where we considered *many diverse set* of exogenous noise that can be present in the observations. We emphasize that we are the first to study, both empirically and theoretically, presenting a practically feasible algorithm, capable of working under different exogenous noises. We propose an algorithm based on the multi-step inverse dynamics objective, a variation of which has already been studied theoretically (Efroni et al., 2021; Lamb et al., 2022), and show that when pre-trained under exogenous noises in observations, it can lead to significant performance during fine-tuning. Additionally, we are *releasing all exogenous noise based offline datasets* such that it can be used later in the community. We believe, for practical scalability and significance of RL in the real world, offline RL can play a significant role, and demonstrating ability to learn from such datasets, that are likely to appear in practice, is an important step in the future.

## M  RESPONSE TO INDIVIDUAL REVIEWERS

### M.1  RESPONSE TO REVIEWER GMBs

#### M.1.1  EXOGENOUS INFORMATION AND DISTRACTOR SETTINGS FROM PRIOR WORKS

***Question*** : *Similarity of D4RL with some exogenous images or videos incorporated as noise in observations*

***Response :***  We would like to emphasize, as briefly mentioned in L.5 that this work studies pixel-based offline RL with exogenous noises in observations. In comparison, Fu et al. (2020) provides benchmark datasets for raw state-based offline RL, whereas recently Lu et al. (2022a) provided pixel-based offline RL benchmarks. We extended from Lu et al. (2022a) to additionally add exogenous noises in observations, with different levels of difficulty : EASY-EXO is what we call the existing Lu et al. (2022a) benchmark; we then extend and re-collect datasets using the same pipeline as Lu et al. (2022a); Fu et al. (2020), but now additionally add MEDIUM-EXO or HARD-EXO exogenous noise during data collection with SAC policy. For example, during MEDIUM-EXO, we place random STL10 images in corner or side, or place fixed video in background. The fixed video background distractor has been studied in online RL before (Zhang et al., 2020), and we provide an extension of it to offline RL. In the HARD-EXO setting, we consider novel datasets and distractors, where

now we have either changing video distractors in background, or have multiple agent observations that the agent sees, in addition to the controllable environment observation. Please see figure 8 for samples of what the agent exogenous observation looks like. Such datasets and distractors have not been studied in the literature before, and we provide a well capable representation learning approach to robustly learn representations under such difficult distractors.

### M.1.2 CONCERN ON REPRESENTATION CONTAINING ONLY TASK RELEVANT INFORMATION.

***Feedback :*** *Weaknesses: The major concern is about the limitation of the representation only containing control-relevant information. How about there are other moving entities in the observations, which are not controllable by the agent but are still very important for the success of the policy? For example, a robot navigates busy streets. Pedestrians are not controllable by the robot, but the robot should consider the status of the pedestrian to avoid collisions. Will the proposed representation learning approaches fully ignore pedestrians and make errors in the decision? Another example is robotics manipulation tasks. The robot hand should manipulate the objects, so the object position and orientations are important information for the control policy. But the proposed method may tend to ignore the object information. This is problematic and limits the scope of the proposed method*

***Response :*** Thank you for the comments and feedback, which are playing a major role in improving the impact of the paper. Please see section L.3 where we provide a generic response to all reviewers.

### M.1.3 IMPORTANCE OF BELLMAN COMPLETENESS

***Feedback :*** *It will be great to explain the importance of the Bellman completeness more clearly and intuitively*

**Response and Comment on Bellman Completeness** : As we elaborated on in the paper, the importance of the Bellman completness assumption was recently demonstrated in the theoretical RL community. Intuitively, this assumption says the following: for any function $f(x, a)$ in the function class it is possible to represent its Bellman backup $r(x, a) + E[\max_{a'} f(x', a') \mid x, a]$. That is, it is possible to optimally learn the Bellman backup at each point of the training process. We will highlight this fact better in the text. Bellamn completeness overall means the ability to represent the optimal solution that mininimizes the bellman error of any function. For example, if the expected bellman error is small then completeness holds, however, vice versa does not necessarily hold. That is, the bellman error might be large because of stochasticisty and not because completeness doesn't hold.

### M.1.4 ADDITIONAL EXPERIMENT RESULTS ON D4RL

***Feedback :*** *The quality is good with the technically solid method and impressive experiment results. However, the experiments are only conducted on locomotion tasks. It will be great to show results in D4RL kitchen tasks, especially because these tasks require object manipulation, and the proposed representation approach may fail in this scenario. I'm curious whether the proposed method can be modified or extended to the manipulation tasks.*

Thank you for the feedback on the impressive experimental results that ACRO achieves. We agree that all our experiments are on the locomotion tasks, based on the available offline benchmark (Lu et al., 2022a) and our newly proposed benchmark datasets. We definitely agree that it would be interesting to show whether ACRO kind of objectives can also perform well in object manipulation based tasks.

For the D4RL kitchen task, we implemented ACRO on top of the existing CQL and TD3 + BC baselines for testing. However, unfortunately we find that, as also reported in past works, that none of the baselines actually work well on the kitchen tasks. For ACRO to work well, we would still require a well performing offline RL algorithm that can solve the kitchen tasks based on the learnt representations, and we found that the existing baselines are not significantly well performing on this task.

However, we also ran ACRO on the D4RL benchmark (Fu et al., 2020). We would like to first mention that D4RL is only a raw state based benchmark and not pixel based observations. As such,

even though past works have shown results for learning representations in D4RL, learning in raw state based environments does not play a significant role for learning representations. Moreover, exogenous noise in observations are primarily for rich observation MDPs with pixels. Despite that, we implemented ACRO on top of the TD3 + Baseline Fujimoto & Gu (2021b) which has been shown to work well on raw state based D4RL benchmark. In figure 29 we show results on two different domains with two different datasets, where on top of the existing baseline, we learn representations with ACRO. Experimental results show that we can still achieve some marginal improvements over baselines by learning with ACRO. We would like to mention here that even though there are marginal improvements, this result may not be significant in the context of our work, since there is no exogenous noise present here, and ACRO is primarily proposed for pixel based RL tasks, to demonstrate ability of learning robust representations by ignoring the irrelevant information.

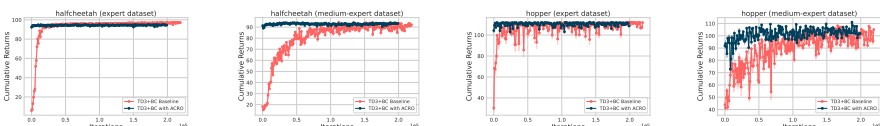

Figure 29: Experimental results on the D4RL benchmark comparing ACRO with the baseline TD3 + BC method. We mention here that previously we do not do experiments with D4RL since the environments are primarily raw state based, instead of pixel based observations. In this work, we primarily propose ACRO for learning robust representations that can ignore exogenous information from pixel observations. We include this result since it is asked by the reviewer on why we do not do experiments with D4RL. We include this preliminary result, and demonstrate that ACRO can work well in raw state D4RL too, even though that is not the primary contribution of our work.

### M.1.5 NOVELTY OF WORK AND RELEASE OF OFFLINE DATASETS

***Feedback :*** *The novelty is okay but not very great, because the multi-step inverse model has been studied for a well in the RL area, and this work is just to adapt it from standard RL to offline RL.*

**Response :** Thank you for your feedback on this, and on raising the question on the novelty of the work. We provided several generic responses in section L clarifying how this work is different from what appeared in the past literature. Most importantly, please see section L.5 where we highlight how this work can be significant to the community. We understand the issue that inverse dynamics models have appeared in the past literature. We emphasize the context of which such objectives have been used (mostly for online RL and exploration), whereas we are primarily focused on learning in presence of exogenous noise (of varying difficulty) and found an instantiation of multi-step inverse models that be very significant in learning from such datasets in offline RL. Furthermore, theoretical works have studied variations of the ACRO objective theoretically (Efroni et al., 2021; Lamb et al., 2022), while we provide a practical algorithm based on pre-training representations that can be used off the shelf in any existing deep RL based offline RL pipeline. Our primary focus is to provide such findings, along with the novel datasets, which we believe would be significant for stepping towards practical scalability of RL in the real world.

### M.2 RESPONSE TO REVIEWER M6QW

### M.2.1 KEY ASSUMPTIONS AND CLARIFICATION

***Feedback :*** *2) there is a lack of in-depth discussion on the assumption and limitation of the current method, including requirement for offline dataset. I may missed some information in the paper and appendix. I would consider increase the ratings if the authors can provide convincing analysis/results for the above points.*

Thank you for the feedback, and apologies if the key assumptions made in the work were not clear from the main text. We repeat here some of the key yet minor assumptions :

1. We assume that the reward function is free of any exogenous noise; ie, even if observations consist of exogenous noise, the rewards depend on the raw endogenous states, r(s,a).

2. We clarify that a purely random policy can be free of any exogenous noise in observations, since the action selection of a random policy does not depend on the observations. Any other non-random polciy would depend on the exgeonous noises $\pi(a|x)$ where $x$ consists of both exogenous and endogenous part of the state space. However, when we pre-train representations, and then use it in an RL algorithm, we are indeed learning a policy $\pi(a|z)$ and depending on the representation being learnt, the resulting optimal policy would depend on the extent to which $z$ depends on the exogenous noise. We show that the latents recovered from ACRO are robust and filters out exogenous noise compared to other baselines considered.

3. There are no assumptions being made on the requirement of the dataset. The offline datasets are collected with SAC policy in presence of exogenous noise, and can have different data distributions depending on the data collecting policy (expert, medium-expert or medium datasets). We emphasize and repeat here that the offline datasets consist of all pixel-based observations.

4. We assume we are in an exogenous block MDP setting Efroni et al. (2021); Du et al. (2019) that has been studied in the theoretical RL community before. In this work, we provide a practical algorithm that can be scaled easily to existing deep RL algorithm pipeline, where learning representations play a key role. The most important contribution of the work is that ACRO can be generally applicable in settings with different types of exogenous information, under block MDP assumptions Du et al. (2019)

### M.2.2 COMPARISONS WITH PRIOR WORK LAMB ET AL. (2022)

***Question :*** *Novelty of the work compared to Lamb et al. (2022) Following bullet 1, I am wondering whether the authors can comment more on the theoretical comparison with Lamb et al. (2022). Without the bottleneck constraint, will the learned representation be guaranteed to discard exogenous information? If not, the claims of Section 3 do not hold for the proposed algorithm, right?*

Thank you for the question about comparisons with prior work Lamb et al. (2022). We empirically compare ACRO with the AC-State objective Lamb et al. (2022) along with details on how ACRO is different compared to previous works (also mentioned in related works). Please see the generic response to all reviewers about this in section L.1, since we believe clarifying these differences would be helpful for all reviewers.

### M.2.3 DATA COVERAGE AND ROLE OF EXPLORATION IN OFFLINE RL

***Question :*** *The algorithm adopts a process of offline representation pretraining and policy fine-tuning with frozen representation. However, such representation learning may highly depend on coverage of the offline dataset. I hope there could be more discussion on the requirement of data/exploration, both in theory and in experiments.*

**Response :** Role of data coverage in offline representation learning: we agree that lack of coverage would degrade the algorithm. This, however, is a well known fact in the offline RL setting. The fact that the concentrability coefficient is finite implies there is a good dataset coverage. We do agree that studying the theoretical aspect of this question is needed. In this work, we mostly focused on this problem from its empirical side and established the failure of existing approaches, as well as offering a fix (that works under the dataset coverage assumption.

Further, the online RL problem with exogenous noise is also of interest. As of now, there is no algorithm with provable guarantees for the general RL problem with exogenous noise (there are some solutions under different sets of assumptions as we elaborated on in the related work section). Our work does not tackle this challenging problem, but study the offline aspects of its; a prevalent problem from a practical perspective.

**Experiments :** We varied the amount of data coverage in the offline datasets, by taking the of times, the data collecting policy takes random actions. We vary the % of random actions from 10% to 50% where we assume that more randomness in actions taken by an expert policy means higher state space coverage. We follow the same experiment setup as before, and now show how the performance of ACRO (and two other baselines) varies as we have lower to higher coverage in the datasets, as in figure 30.

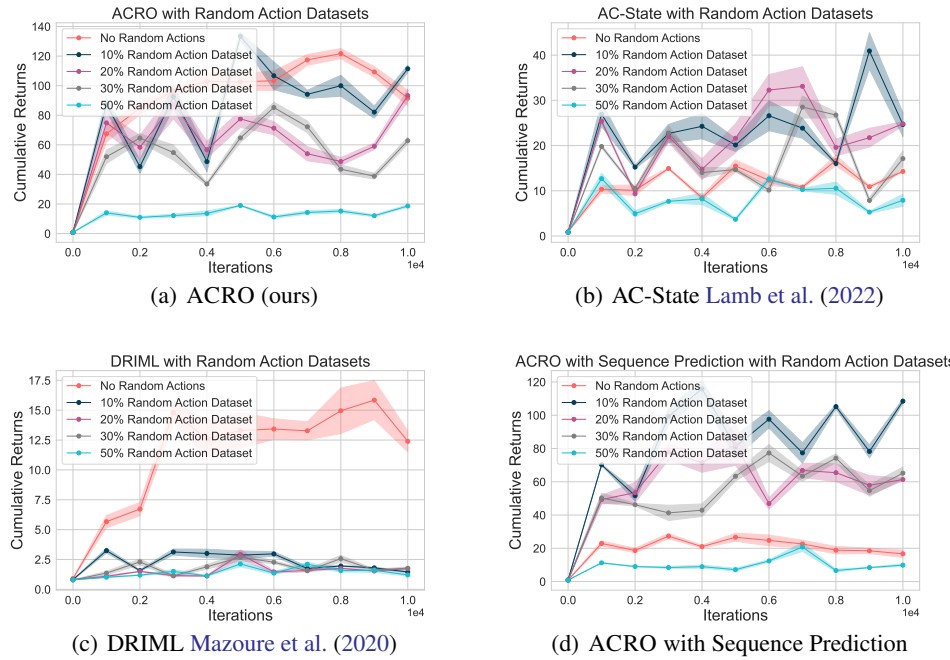

Figure 30: Experimental results comparing different representation learning methods as we have varying amounts of coverage on the datasets. We make the assumption that adding more random actions to the datasets means the dataset has a higher coverage of state and action space. We show that as the % of random actions on datasets increases, the performance of each method degrates, especially the ones like DRIML that are independent of action prediction. In contrast, all other methods that rely on action prediction, suffer less depending on the amount of random actions that may exist in the datasets. This shows a potential for representation objectives based on action prediction that the performance of these methods degrades least even if the dataset quality is poor (where in our case, higher coverage due to more random actions means the dataset is degrading from an expert dataset to more of a random dataset).

### M.2.4    ACRO IN ONLINE RL FOR REPRESENTATION LEARNING AND EXPLORATION

***Feedback :*** *The representation is only tested with offline RL finetuning, while it is not clear whether the learned representation is sufficient for learning optimal policies in an online manner. Intuitively, the latter would be harder and thus more interesting (the representation should be not only good for modeling optimal behaviors, but also good for exploration). If the claims hold that the learned representation is controllable, it should be able to learn a policy by interaction. Combined with bullet 3 above, I am worried that if the offline dataset does not have good coverage, downstream online policy learning will be hard.*

Thank you for the comment and providing suggestions on how we can extend the paper. We would like to note here that the problem of considering the online setting, and extending beyond our current focus on offline RL is that the online setting is dependent on exploration too (since representation learning in online setting is inherently coupled with the amount of exploration) and this is something, while interesting, is perhaps beyond the scope of our current work. In future, we would definitely consider extending ACRO to the online setting, which might additionally require a mechanism for exploration while ignoring exogenous noise. In our current work, we are mostly studying robustness to exogenous information given datasets, and we study the exploration question there through assumptions on the role of coverage in the datasets (as you noted in the previous question). As pointed out rightly, in offline RL this is tested based on data coverage in the data distributions, and our response and results above is an attempt towards addressing this with ACRO.

## M.3 Response to Reviewer yx3d

### M.3.1 Inverse Dynamics in RL

**Feedback :** *It would have been nice to include in the introduction some indication of how this work innovates over existing representation learning methods since inverse dynamics models have been heavily used in RL and model-based RL.*

**Response :** We would like to thank you for bring this question, and highlight that inverse dynamics models have appeared in the literature in the past Pathak et al. (2017); Efroni et al. (2021); Lamb et al. (2022). Please see how response on this in section L.1 where we compare how the ACRO is different compared to past works in terms of novelty, and significance in offline RL. We emphasize that even though objective-wise ACRO is a variation of past works, the context to which we study this in presence of exogenous noise, and to show the capability of learning generic robust representations that can be used in any RL algorithm, can be of practical importance. In addition to discussions around this in generic responses in L, we would be happy to answer further questions on novelty and significance of ACRO.

### M.3.2 POMDP and Exogenous Block MDP

**Question :** *The description of an MDP actually describes a PO-MDP, with an exogenous block context. In addition, the typical formulation of the agent carries $\pi(a \mid x_{1...t})$, or the history of past observations. Policies conditioned on the current observation require the "rich observation" assumption, which should be stated in 2.1, not 2.2.*

The decision process we consider here is an MDP in the sense that the observations are Markovian; the observation at time step $x_t$ does not depend on the history conditioning on $x_{t-1}$ and $a_{t-1}$. We will highlight that in section 2.2 to avoid confusions and miss-understandings in the future. Indeed, in this work we do not study the problem of representation learning in POMDPs (in the presence of high-dimensional observation data). This problem is of great interest and challenge.

### M.3.3 About Multi-Step Inverse Dynamics Objective

**Question :** *The work claims that the multi-step objective allows for more diverse representations because it forces the representation to encode long-range dependencies. However, only the first action in the sequence is added, so it is entirely unintuitive as to why the representation would have to be different.*

What is the specific reference on the claim that ACRO allows for "more diverse representations"? We talk about our newly proposed datasets having diverse exogenous distractors, but I'm not aware of us claiming that the representations are more "diverse".

**Feedback :** *In the example of the action being stored in the observation, this would still be the case even if t+k is given.*

Suppose that $\tilde{x}_t = (x_t, a_{t-1})$, so the observation stores an original observation along with the most recent action. Then consider the 1-step inverse model: $\mathbb{P}(a_t | \tilde{x}_t, \tilde{x}_{t+1}) = \mathbb{P}(a_t | x_t, a_{t-1}, x_{t+1}, a_t)$. The optimal solution can thus completely ignore $x$. Note that you can't use $\tilde{x}_t = (x_t, a_t)$ as the observation for a counter-example because it violates causality. You are right that recording many recent actions will cause issues for multi-step inverse models, so we updated the text to reflect this.

**Feedback :** *"Furthermore, this representation introduces its own issues, since the problem itself is now ill-defined: suppose that there are multiple action sequences between $s_t$ and $s_{t+1}$. In this case, the inverse dynamics model would have to output an ambiguous probability which would be dependent on the frequency one path was taken over another in the dataset. It is not clear what representation this would encode."*

You are right that there are multiple action sequences for travelling from one state to another. Even in the case of a simple empty gridworld, there are two equally valid actions for going from the top-left to the bottom-right in as few steps as possible. We can either go down, or go right on the first step. As another example, if the agent starts in the top-left corner, going left or going up will both keep the agent in the same position. Thus the NLL of the optimal multi-step inverse model will indeed not be zero for most environments. Can you explain further why you believe this is a problem?

### M.3.4 TASK RELEVANT INFORMATION

***Question :*** *A clear weakness of this work is that even though it equates exogenous with irrelevant, it is actually likely to remove information that could be task-vital. This is because the inverse dynamics model only needs to capture sufficient information to predict the first action between a pair of states, and can lose any information once it can make that prediction. Suppose that we are in an object manipulation domain where the agent uses an arm to grasp a block. The representation has no reason to capture the block because only the arm is necessary to predict the action. None of the experiments appear to capture these kinds of relationships*

**Response :** We would like to thank you for such insightful comments and feedback, on whether ACRO objective can ignore task relevant information. If this is true, then we agree this would definitely be a key weakness of our work. We provided a generic response to this in section L.3 since this would also provide better insights to other reviews. Please refer to section L.3.

### M.3.5 COMPARISONS TO PRIOR WORK LAMB ET AL. (2022)

***Question :*** *This work repeatedly cites Guaranteed Discovery of Controllable Latent States with Multi-Step Inverse Models (Lamb et al 2022), which appears to approach the same problem with the same solution. If this work is by the authors, then this is disingenuous since the authors' claims are being supported by citations of the author's own work. Furthermore, the claims being supported, such as the resolving of single-step issues by using a multi-step predictor, appear to be made without strong support.*

**Response :** The question about comparisons to prior work Lamb et al. (2022) have been brought up by other reviewers too. We provided a generic response, along with additional experimental results, and detailed comparison between ACRO and Lamb et al. (2022) in section L.1. We emphasize that there are few major differences between the two works, and how Lamb et al. (2022) can be practically limited as it only recovers a small underlying tabular MDP for planning and does not show any significance in a practical deep RL setting. In contrast, ACRO pre-trains robust representations that can then be used in any existing deep RL algorithm, which makes the approach more generic, and not requiring any specific discrete bottlenecks as in Lamb et al. (2022).

### M.3.6 PREVIOUS WORKS WITH DISTRACTORS IN RL ZHANG ET AL. (2020)

***Question :*** *The proposed offline RL datasets are the same (at least in conceit) as those described by Learning invariant representations for reinforcement learning without reconstruction (Zhang et. al. 2020)*

**Response :** We would like to emphasize how the exogenous distractors considered in this work is quite different than what appeared in past literature. In section L.5 we provided a generic response to all reviewers about the significance and impact this work can have, towards practical scalability of RL in real world. Prior works in the online setting, as in Zhang et al. (2020) have only considered time uncorrelatred distractors whereas we build primarily on the theoretical studies of exogenous RL Efroni et al. (2022b; 2021) to further provide an offline dataset benchmark with exogenous information. We would like to note that we will be releasing these datasets so they can be used as benchmarks, for future research and reproducibility, beyond the novelty of specific exogenous noise that we considered. We also provided a similar response to another reviewer question about this in section M.1.1.

### M.3.7 ASSUMPTIONS ON DATASETS

Please see response in section M.2.1 where we listed the key assumptions and clarified them.

*On the issue of assumptions in datasets :* There are no assumptions being made on the requirement of the dataset. The offline datasets are collected with SAC policy in presence of exogenous noise, and can have different data distributions depending on the data collecting policy (expert, medium-expert or medium datasets). We emphasize and repeat here that the offline datasets consist of all pixel-based observations.

### M.4 RESPONSE TO REVIEWER GC23

#### M.4.1 EMPIRICAL COMPARISON TO PRIOR WORK LAMB ET AL. (2022)

**Question :** *No empirical comparison to the closely related prior work Lamb et al. (2022). I highly encourage the authors to move the majority part of Section 2.2 (Proposed Method) to Section 2.1 (Preliminaries) and only highlight new ideas in Section 2.2...... A comparison to Lamb et al. (2022) should be included, because the paper claims that learning a continuous representation is the main distinction (hence novelty) to Lamb et al. (2022). Without this, the use of continuous representation is not well-justified.*

**Response :** We included a generic response in section L.1 comparing the ACRO objective with Lamb et al. (2022), both empirically and theoretically. We included details on how ACRO differs from prior works, and why an approach based on ACRO can be generally applicable, compared to the limitations in Lamb et al. (2022).

#### M.4.2 EXPERIMENTS WITHOUT ANY EXOGENOUS NOISE; ACRO WORKS BETTER EVEN WITHOUT ANY EXOGENOUS INFORMATION

**Question :** *It would be more comprehensive to include a noise/distractor-free setting to get an idea of how much each method suffers from visual distractors.*

**Response :** We actually included results in table 2 where we do not consider any exogenous noise or distractors at all. All the results provided in 2 are based on the no-distractor setting from the Lu et al. (2022a) pixel-based offline benchmark. We label this as the EASY-EXO setting, with other more difficult exogenous noise datasets as in MEDIUM-EXO and HARD-EXO based results, that are extended, and provided as new benchmarks in this work.

We would like to emphasize here that even though the main focus of the work is on exogenous information in offline datasets, our results as in table 2 also highlights that ACRO can be generally applicable for representation learning. While we focus mostly on exogenous information in the description of our setting, we should have emphasized the scope of the ACRO technique is more broadly applicable in the representation learning literature for RL.

#### M.4.3 NORMALIZED PERFORMANCE FIGURES

**Response :** We apologize that some of the normalized or averaged results that were presented in the paper were not clear. We originally included details of the averaging in the caption of figures. For furhter clarification, and since it would be helpful for other reviewers as well, we provided an explanation of the normalized or averaged results figures in section L.4.

#### M.4.4 EXPLANATION OF RECONSTRUCTION FIGURES

**Feedback :** *Although I appreciate the analysis through reconstruction (Figure 7), it's unclear whether the proposed method is clearly better than DRIML.*

**Response :** The reconstruction for ACRO removes exogenous information more than DRIML. Note that blurring of the controllable part does not necessarily mean that there is less information about the controllable part in the encoder. Specifically for such locomotion tasks, only capturing certain pixels for each joint is actually sufficient to predict the actions accurately. Therefore, in principle the controllable part can be even more noisy while still encoding sufficient information for control. This combined with the fact that ACRO achieves better downstream performance on both with and without distractors case suggests that it is able to remove exogenous information and retain endogenous information better than DRIML. Another case in point here is the DrQ reconstructions, which blur out most of the agent yet achieve similar performance to DRIML.

### M.5 RESPONSE TO REVIEWER moWf

We would like to thank the reviewer for the high score in the paper, and we greatly appreciate the insightful comments and feedback. The other reviewers raised some concerns about the scope of this work too, and we provided a generic response to all reviewers, as in section L. We hope the

responses to other reviewers would also help in fully understanding the significance and impact of our work.

### M.5.1 ASSUMPTIONS AND LIMITATIONS

***Feedback :*** *Limitation on exo-free policy : that the learning problem requires data from an exo-free policy. The experimental results are both extensive and in support of the proposed method. The main drawbacks I see are the requirement for an exo-free policy to learn from and the block assumption, which excludes POMDPs.*

**Response :** Thank you for the feedback, and we apologize that some of the assumptions made in this work were not clear. We list below some of the key assumptions made in this work

1. We assume that the reward function is free of any exogenous noise; ie, even if observations consist of exogenous noise, the rewards depend on the raw endogenous states, r(s,a).

2. We clarify that a purely random policy can be free of any exogenous noise in observations, since the action selection of a random policy does not depend on the observations. Any other non-random polciy would depend on the exgeneous noises $\pi(a|x)$ where $x$ consists of both exogenous and endogenous part of the state space. However, when we pre-train representations, and then use it in an RL algorithm, we are indeed learning a policy $\pi(a|z)$ and depending on the representation being learnt, the resulting optimal policy would depend on the extent to which $z$ depends on the exogenous noise. We show that the latents recovered from ACRO are robust and filters out exogenous noise compared to other baselines considered.

3. There are no assumptions being made on the requirement of the dataset. The offline datasets are collected with SAC policy in presence of exogenous noise, and can have different data distributions depending on the data collecting policy (expert, medium-expert or medium datasets). We emphasize and repeat here that the offline datasets consist of all pixel-based observations.

4. We assume we are in an exogenous block MDP setting Efroni et al. (2021); Du et al. (2019) that has been studied in the theoretical RL community before. In this work, we provide a practical algorithm that can be scaled easily to existing deep RL algorithm pipeline, where learning representations play a key role. The most important contribution of the work is that ACRO can be generally applicable in settings with different types of exogenous information, under block MDP assumptions Du et al. (2019)

### M.5.2 ACRO PERFORMS BETTER EVEN WITHOUT ANY EXOGENOUS INFORMATION

***Question :*** *It is interesting that ACRO outperforms other method even when there are no distractors (if I understand Table 2). This would imply that the representation learned by ACRO is a better representation than alternatives, even when there is no exogenous information. This is interesting, since the theory used to derive ACRO is all based on the existence of exogenous information. Do the authors have some thoughts here?*

**Respone :** We would like to emphasize here that even though the main focus of the work is on exogenous information in offline datasets, our results as in table 2 also highlights that ACRO can be generally applicable for representation learning. While we focus mostly on exogenous information in the description of our setting, we should have emphasized the scope of the ACRO technique is more broadly applicable in the representation learning literature for RL.

Furthermore, since the offline datasets from Lu et al. (2022a) which we label as EASY-EXO are pixel-based data, we want to emphasize that technically there is still some level of exogenous information present here : for example, the background, floor and colors in the control tasks may contain some exogenous info from the observations that are not relevant for solving the task. This is why, in the main text, we refer to the results of 2 as EASY-EXO. However, as the reviewer has rightly pointed - ACRO is more generic as a representation learning approach for offline RL. Even though inverse dynamics models have appeared in the literature in past, in other context such as exploration in online RL, we emphasize that ACRO objective based on a multi-step inverse model can be particularly suited for offline representation learning, which has not been well studied, empirically, in the past literatre.

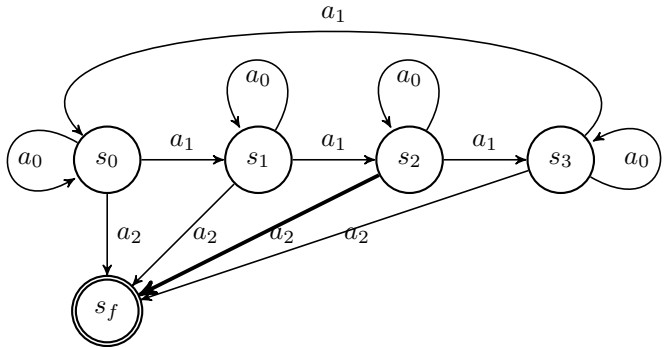

Figure 31: An MDP where one-step models can fail.

### M.5.3 EXPLANATION OF RESULTS

***Question :*** *What is the difference in setup between Table 2 and Figure 5? As far as I understood, Table 2 contains no distractors, while Figure 5 contains the distractors used by Lu et al. [1]. What I don't understand is what "easy", "medium" and "hard" refer to in Figure 5.*

**Response :** Thank you for asking for the clarification. You are correct that 2 contains results without any exogenous distractors. It contains results from using the Lu et al. (2022a) benchmark as it is, without distractors. Figure 5 contains results from the Lu et al. (2022a) benchmark too, except now we have the distractor suite from V-D4RL benchmark. In their datasets, they have 3 different type of distractors, which induces a distribution shift in the dataset, which is referred to asy *easy-shift*, *medium-shift* and *hard-shift*. Figure 5 is therefore what we refer to as the results for EASY-EXO in our setting. In this work, we mainly extend from Lu et al. (2022a) and collect new datasets where we vary the type of exogenous distractors, not considered in Lu et al. (2022a) referred to as MEDIUM-EXO and HARD-EXO settings.

***Question :*** *Normalized result figures + Where are the numbers in Table 3 coming from? Are these averages across the tasks? In which distractor setting?*

**Response :** We provide a clarification for the normalized or averaged results included in main text, in section L.4. We aplogize that even though we mentioned it in the appendix, this was not clear to few reviewers as to what the normalized results mean in this context.

Table 3 contains results of ablation studies with the ACRO objective. For this, we consider only the Cheetah-Run task with different dataset types (random, medium-replay and expert) with the HARD-EXO exogenous distractor. As is described in Section 5.4, we perform ablations with the ACRO objective where we added the timestep embedding, used only a single-step future observation (instead of a varied value of horizon $k$), and removed the conditioning on future observations.

# Additional Results for Rebuttal

We include some additional results in the rebuttal, which were primarily ***not raised by reviewers***, but are still useful for overall understanding and quality of work.

## N    COUNTEREXAMPLE FOR ONE-STEP MODELS

In this section, we build an MDP and a dataset of trajectories in that MDP where a one-step model will fail to allow the learning of the optimal policy. The MDP can be seen in Figure 31, all transitions are deterministic, and yield 0 reward, apart for action $a_2$ in state $s_2$, which gives a reward of 1 (bold arrow in the graph). The final state $s_f$ denotes the termination of the trajectory

We consider the dataset comprised of the following five trajectories, formatted as $\langle s_t, a_t, r_t, s_{t+1} \dots \rangle$:

$$\mathcal{D} = \{ \underbrace{\langle s_0, a_0, 0, s_0, a_0, 0, s_0, a_2, 0, s_f \rangle}_{\tau_1}, \tag{22}$$

$$\underbrace{\langle s_0, a_1, 0, s_1, a_0, 0, s_1, a_2, 0, s_f \rangle}_{\tau_2}, \tag{23}$$

$$\underbrace{\langle s_0, a_1, 0, s_1, a_1, 0, s_2, a_0, 0, s_2, a_2, \mathbf{1}, s_f \rangle}_{\tau_3}, \tag{24}$$

$$\underbrace{\langle s_0, a_1, 0, s_1, a_1, 0, s_2, a_1, 0, s_3, a_0, 0, s_3, a_2, 0, s_f \rangle}_{\tau_4}, \tag{25}$$

$$\underbrace{\langle s_0, a_1, 0, s_1, a_1, 0, s_2, a_1, 0, s_3, a_1, 0, s_0, a_2, 0, s_f \rangle}_{\tau_5} \}. \tag{26}$$

$\tau_1$ loops twice in $s_0$ and then terminates. The other four trajectories reach $s_1$, $s_2$, $s_3$ and $s_0$ by taking $a_1$ a minimal number of times, then loop once (except $\tau_5$), and terminate. We note that this dataset covers the full state and action space.

It is possible to reach a 0 loss with a one-step inverse model $p_1$ built on top of the following representation $\phi$ from $\mathcal{S}$ to $\mathcal{X} = \{x_0, x_1, x_f\}$: $\phi(s_0) = \phi(s_2) = x_0$, $\phi(s_1) = \phi(s_3) = x_1$ and $\phi(s_f) = x_f$:

$$p_1(a|x_0, x_0) = \delta_{a=a_0}, \tag{27}$$
$$p_1(a|x_0, x_1) = \delta_{a=a_1}, \tag{28}$$
$$p_1(a|x_0, x_f) = \delta_{a=a_2}. \tag{29}$$

Now, once projected onto $\mathcal{X}$, the dataset becomes:

$$\mathcal{D} = \{ \underbrace{\langle x_0, a_0, 0, x_0, a_0, 0, x_0, a_2, 0, x_f \rangle}_{\tau_1}, \tag{30}$$

$$\underbrace{\langle x_0, a_1, 0, x_1, a_0, 0, x_1, a_2, 0, x_f \rangle}_{\tau_2}, \tag{31}$$

$$\underbrace{\langle x_0, a_1, 0, x_1, a_1, 0, x_0, a_0, 0, x_0, a_2, \mathbf{1}, x_f \rangle}_{\tau_3}, \tag{32}$$

$$\underbrace{\langle x_0, a_1, 0, x_1, a_1, 0, x_0, a_1, 0, x_1, a_0, 0, x_1, a_2, 0, x_f \rangle}_{\tau_4}, \tag{33}$$

$$\underbrace{\langle x_0, a_1, 0, x_1, a_1, 0, x_0, a_1, 0, x_1, a_1, 0, x_0, a_2, 0, x_f \rangle}_{\tau_5} \}. \tag{34}$$

We see that action $a_2$ in state $x_0$ has an expected reward of $1/3$, and it is the only state-action pair with a non-zero expected reward. Any reasonable offline RL algorithm applied to this dataset will output a policy with a non-zero probability assigned to that action in $x_0$. However, executing that policy, *i.e.,* taking action $a_2$ in state $s_0$ terminates the episode with 0 reward, which is suboptimal.

On the other hand, an optimal two-step model (and by extension an n-step one) will not collapse states $s_0$ and $s_2$ as this would prevent distinguishing the first action taken in $\tau_1$ from the first action taken in $\tau_3$ (corresponding respectively to pairs $(s_0, s_0)$ and $(s_0, s_2)$ after two timesteps). Consequently, an offline RL applied in representation can still learn an optimal policy.

## O    END TO END FINE-TUNING EXPERIMENTS

In all our current experiments, primarily presented in the paper, we only pre-train the representations and then keep it fixed during later fine-tuning of the offline RL algorithm. In this section, we

perform a variation where instead of only pre-training, we now perform fine-tuning or end to end learning where the representations are simultaneously updated along with the RL algorithm. Experimental results in this section, with different exogenous distractor types, shows that even in the end to end learning setting, ACRO can significantly outperform the baselines by learning robust representations.

Due to the questions about relevancy of task information - we also perform additional experiments where in addition to the ACRO objective, we now also predict the rewards using a reward function predictor. We show that even when trying to predict task relevant reward information, the performance of ACRO with and without reward prediction are almost similar. This further verifies that ACRO does not ignore any task relevant information, such as information present in the reward function.

## O.1    EXPERIMENT RESULTS - CORRELATED AND UNCORRELATED IMAGE DISTRACTORS

**Uncorrelated Exogenous Image Distractors :** Figure 12 shows results with exogenous image distractors placed in the corner of the agent observation space. Figure 13 and Figure 33 additionally shows results with STL10 exogenous distractors placed on the side of the agent observations, augmenting the observation space of the agent. We find that ACRO can significantly outperform baselines irrespective of whether we pre-train representations only, as in Figure 13 or do end to end fine tuning of representations with uncorrelated exogenous observations , as shown in Figure 33.

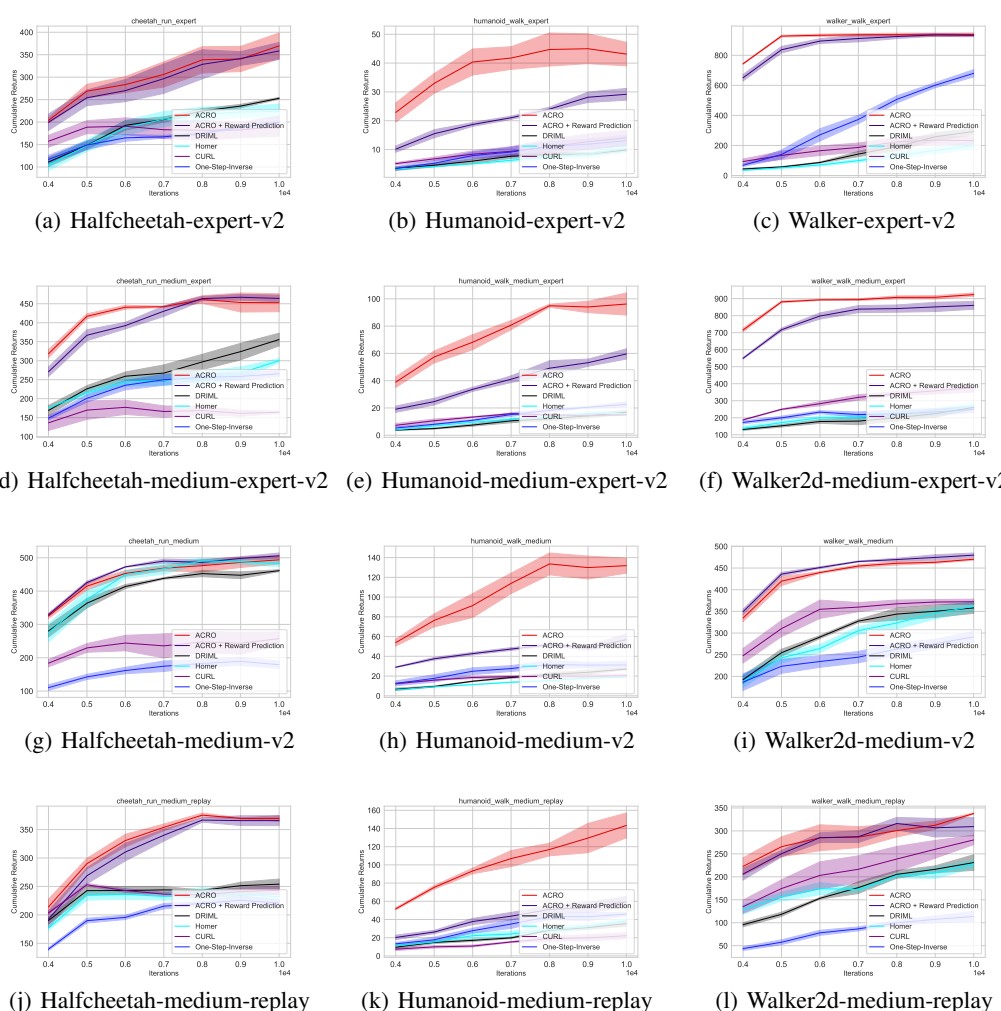

(a) Halfcheetah-expert-v2

(b) Humanoid-expert-v2

(c) Walker-expert-v2

(d) Halfcheetah-medium-expert-v2

(e) Humanoid-medium-expert-v2

(f) Walker2d-medium-expert-v2

(g) Halfcheetah-medium-v2

(h) Humanoid-medium-v2

(i) Walker2d-medium-v2

(j) Halfcheetah-medium-replay

(k) Humanoid-medium-replay

(l) Walker2d-medium-replay

Figure 32: **Performance comparison with uncorrelated exogenous observations on the side; pre-training representations only** In this experiment, we place an exogenous image in the corner of the observation space of the agent; fine-tuning or end to end learning of representations, comparing ACRO, ACRO + **Reward Prediction** with several other baselines in presence of cifar exogenous image distractor.

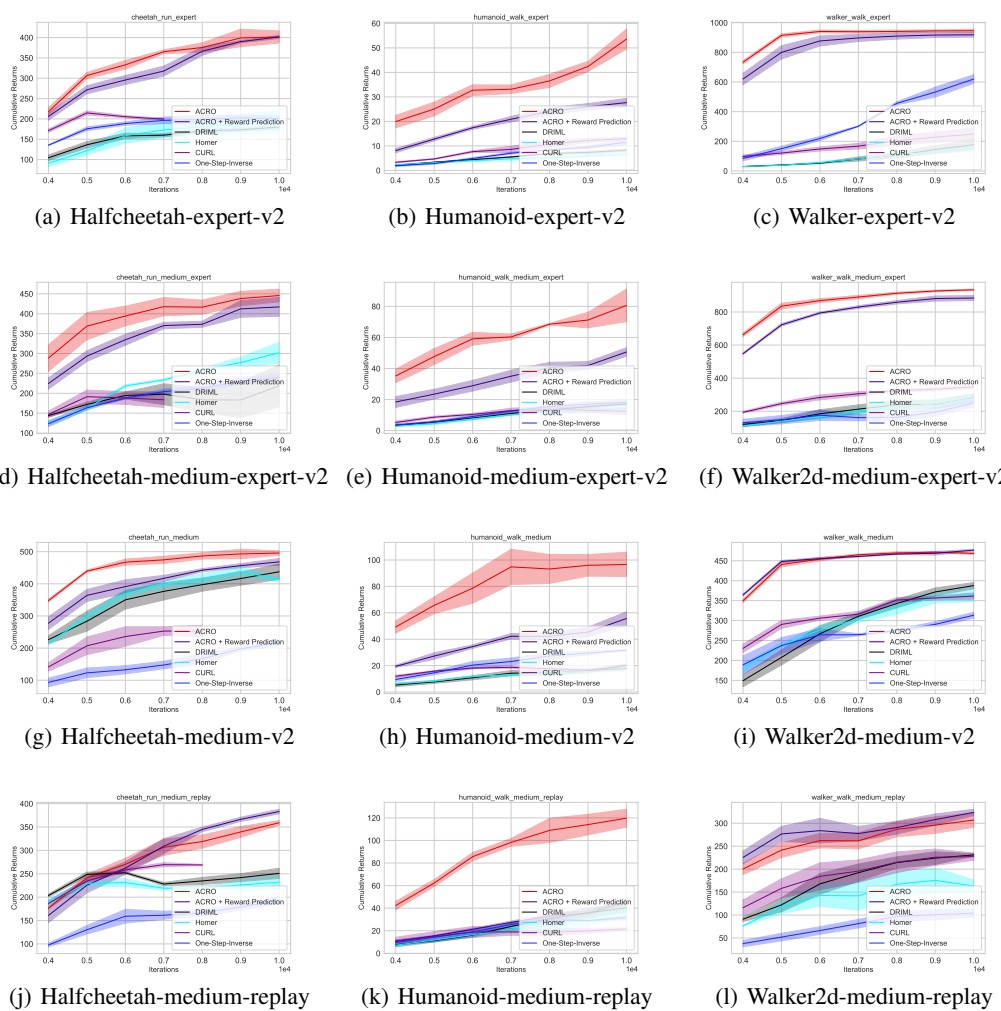

(a) Halfcheetah-expert-v2     (b) Humanoid-expert-v2     (c) Walker-expert-v2

(d) Halfcheetah-medium-expert-v2   (e) Humanoid-medium-expert-v2   (f) Walker2d-medium-expert-v2

(g) Halfcheetah-medium-v2     (h) Humanoid-medium-v2     (i) Walker2d-medium-v2

(j) Halfcheetah-medium-replay     (k) Humanoid-medium-replay     (l) Walker2d-medium-replay

Figure 33: **Performance comparison with uncorrelated exogenous observations on the side; end to end, fine-tuning of representations** Comparisons with baselines for ACRO and ACRO with additional reward prediction

