# OpenReview forum: "Agent-Controller Representations: Principled Offline RL with Rich Exogenous Information"
_ICLR.cc/2023/Conference — Submitted to ICLR 2023_

### Official Review · Reviewer_moWf · 2022-10-21

**Confidence:** 3
**Correctness:** 4
**Technical Novelty And Significance:** 4
**Empirical Novelty And Significance:** 4
**Recommendation:** 8

**Clarity, Quality, Novelty And Reproducibility:**

* The presentation is a bit confusing at first. The description of the setup appears like a POMDP. Eventually it becomes clear that we are in a setup where the observation uniquely identifies the state. Here, I would advise using the term EX-BMDP more prominently (it is mentioned, but appears more like an aside) so that it is clear early on that we are in a special case.
* It is interesting that ACRO outperforms other method even when there are no distractors (if I understand Table 2). This would imply that the representation learned by ACRO is a better representation than alternatives, even when there is no exogenous information. This is interesting, since the theory used to derive ACRO is all based on the existence of exogenous information. Do the authors have some thoughts here?
* What is the difference in setup between Table 2 and Figure 5? As far as I understood, Table 2 contains no distractors, while Figure 5 contains the distractors used by Lu et al. [1]. What I don't understand is what "easy", "medium" and "hard" refer to in Figure 5. Also, the captions reads "Normalized results across two domains [...]": which domains?
* Where are the numbers in Table 3 coming from? Are these averages across the tasks? In which distractor setting?

**Strength And Weaknesses:**

**strengths**

* The paper's about a relevant question (how do we ignore unimportant/distracting information in RL?).

* The paper is rigorous and argues its case convincingly. The approach is motivated by theoretical arguments. The experiments are extensive and show an edge over strong baselines. Ablations empirically motivate the decisions that the authors make.

**weaknesses**

* The block assumption (that the emission distributions of different states are disjoint) is limiting. It amounts to a full-observability assumption.

* Another limiting factor is that the learning problem requires data from an exo-free policy.

**Summary Of The Paper:**

The paper proposes a method for learning representations that are  invariant to task-unrelated information in the offline RL setting.  Theoretical results are presented which show that such representations  allow accurate value function approximation and that using  representations which are not invariant to task-unrelated information  can cause value function approximation to fail. Experiments and ablation studies support these claims.


**Summary Of The Review:**

The paper makes solid experimental and theoretic contributions to the problem of discarding irrelevant information in RL. The experimental results are both extensive and in support of the proposed method. The main drawbacks I see are the requirement for an exo-free policy to learn from and the block assumption, which excludes POMDPs.

---

> ### Author Response · Authors · 2022-11-15
> **Thank you for the high score on the paper**
>
> We would like to thank the reviewer for the high score in the paper, and we greatly appreciate the
> insightful comments and feedback. The other reviewers raised some concerns about the scope of
> this work too, and we provided a generic response to all reviewers, as in section L. We hope the
> responses to other reviewers would also help in fully understanding the significance and impact of
> our work.

---

> ### Author Response · Authors · 2022-11-15
> **ASSUMPTIONS AND LIMITATIONS**
>
> Response : Thank you for the feedback, and we apologize that some of the assumptions made in
> this work were not clear. We list below some of the key assumptions made in this work
>
> 1. We assume that the reward function is free of any exogenous noise; ie, even if observations
> consist of exogenous noise, the rewards depend on the raw endogenous states, r(s,a).
>
> 2. We clarify that a purely random policy can be free of any exogenous noise in observations,
> since the action selection of a random policy does not depend on the observations. Any
> other non-random polciy would depend on the exgeonous noises π(a|x) where x consists
> of both exogenous and endogenous part of the state space. However, when we pre-train representations, and then use it in an RL algorithm, we are indeed learning a policy π(a|z) and
> depending on the representation being learnt, the resulting optimal policy would depend on
> the extent to which z depends on the exogenous noise. We show that the latents recovered from ACRO are robust and filters out exogenous noise compared to other baselines
> considered.
>
> 3. There are no assumptions being made on the requirement of the dataset. The offline datasets
> are collected with SAC policy in presence of exogenous noise, and can have different data
> distributions depending on the data collecting policy (expert, medium-expert or medium
> datasets). We emphasize and repeat here that the offline datasets consist of all pixel-based
> observations.
>
> 4. We assume we are in an exogenous block MDP setting Efroni et al. (2021); Du et al. (2019)
> that has been studied in the theoretical RL community before. In this work, we provide a
> practical algorithm that can be scaled easily to existing deep RL algorithm pipeline, where
> learning representations play a key role. The most important contribution of the work
> is that ACRO can be generally applicable in settings with different types of exogenous
> information, under block MDP assumptions Du et al. (2019)

---

> ### Author Response · Authors · 2022-11-15
> **ACRO PERFORMS BETTER EVEN WITHOUT ANY EXOGENOUS INFORMATION**
>
> We would like to emphasize here that even though the main focus of the work is on
> exogenous information in offline datasets, our results as in table 2 also highlights that ACRO can be
> generally applicable for representation learning. While we focus mostly on exogenous information
> in the description of our setting, we should have emphasized the scope of the ACRO technique is
> more broadly applicable in the representation learning literature for RL.
>
> Furthermore, since the offline datasets from Lu et al. (2022a) which we label as EASY-EXO are pixelbased data, we want to emphasize that technically there is still some level of exogenous information
> present here : for example, the background, floor and colors in the control tasks may contain some
> exogenous info from the observations that are not relevant for solving the task. This is why, in the
> main text, we refer to the results of 2 as EASY-EXO. However, as the reviewer has rightly pointed
> - ACRO is more generic as a representation learning approach for offline RL. Even though inverse
> dynamics models have appeared in the literature in past, in other context such as exploration in online
> RL, we emphasize that ACRO objective based on a multi-step inverse model can be particularly
> suited for offline representation learning, which has not been well studied, empirically, in the past
> literatre.

---

> ### Author Response · Authors · 2022-11-15
> **EXPLANATION OF RESULTS**
>
> Thank you for asking for the clarification. You are correct that 2 contains results
> without any exogenous distractors. It contains results from using the Lu et al. (2022a) benchmark
> as it is, without distractors. Figure 5 contains results from the Lu et al. (2022a) benchmark too,
> except now we have the distractor suite from V-D4RL benchmark. In their datasets, they have 3
> different type of distractors, which induces a distribution shift in the dataset, which is referred to
> as easy-shift, medium-shift and hard-shift. Figure 5 is therefore what we refer to as the results for
> EASY-EXO in our setting. In this work, we mainly extend from Lu et al. (2022a) and collect new
> datasets where we vary the type of exogenous distractors, not considered in Lu et al. (2022a) referred
> to as MEDIUM-EXO and HARD-EXO settings.
>
> We provide a clarification for the normalized or averaged results included in main text,
> in section L.4 of updated draft. We apologize that even though we mentioned it in the appendix, this was not clear to
> few reviewers as to what the normalized results mean in this context.
> Table 3 contains results of ablation studies with the ACRO objective. For this, we consider only
> the Cheetah-Run task with different dataset types (random, medium-replay and expert) with the
> HARD-EXO exogenous distractor. As is described in Section 5.4, we perform ablations with the
> ACRO objective where we added the time step embedding, used only a single-step future observation
> (instead of a varied value of horizon k), and removed the conditioning on future observations.

---

> ### Author Response · Authors · 2022-11-15
> **Summary of Individual Responses to your questions :**
>
> Thank you for the high score of the paper, and for appreciating the significance of our work. Brief summary of our responses to your questions :
>
> 1. We provided clarification of experimental results and normalized/averaged result plots.
>
> 2. We provided brief justification to why ACRO can perform better even without any exogenous noise being present.
>
> 3. List of assumptions and limitations, which are perhaps relevant theoretically, but not empirically for comparing ACRO with other baseline representation learning methods.

---

### Official Review · Reviewer_GC23 · 2022-10-23

**Confidence:** 3
**Correctness:** 2
**Technical Novelty And Significance:** 2
**Empirical Novelty And Significance:** 3
**Recommendation:** 5

**Clarity, Quality, Novelty And Reproducibility:**

**Clarity**

This paper is easy to follow. However, the contribution/novelty in comparison to the relevant prior work was not clearly described. For example, when the paper introduces the main idea, it says "Our proposed method, which we call Agent-Controller Representations for Offline-RL (ACRO), optimizes the following objective based on a multi-step inverse model: [Equation 1]", without referring to [Lamb et al.] which proposed the equivalent objective called Agent-Controller state (AC state). As someone who is less familiar with [Lamb et al.], I had entirely misinterpreted the statement and thought this objective is original until I checked [Lamb et al.]. Similarly, Lemma 1 is in fact from [Lamb et al.], but there is no explicit statement about it. I highly encourage the authors to move the majority part of Section 2.2 (Proposed Method) to Section 2.1 (Preliminaries) and only highlight new ideas in Section 2.2.

**Novelty**

The main objective (multi-step inverse model) is essentially the same as [Lamb et al.] except for a few details such as continuous representation in this paper as opposed to discretized representation in [Lamb et al.]. However, the application of this idea to offline RL setting is novel. In addition, the new visual-temporal disctractors introduced by this paper are new and interesting.

**Quality**

* A comparison to [Lamb et al.] should be included, because the paper claims that learning a continuous representation is the main distinction (hence novelty) to [Lamb et al.]. Without this, the use of continuous representation is not well-justified.

* It would be more comprehensive to include a noise/distractor-free setting to get an idea of how much each method suffers from visual disctractors.

* It is unclear what "normalized performance" means, given that Figure 6 has various ranges from 100 to 500. It would be informative to present performance normalized w.r.t. demonstration performance across the entire main paper (assuming that raw returns are included in the appendix).

* Although I appreciate the analysis through reconstruction (Figure 7), it's unclear whether the proposed method is clearly better than DRIML. It looks like the reconstruction from the proposed method is just blurrier than DRIML not only for the background but also for the controllable part.

**Reproducibility**
The paper provided details of hyperparameters, architectures, datasets, but not the code.

**Strength And Weaknesses:**

[Strength]
* The idea is technically sound.
* The results look good.
* Introduces interesting visual distractors.


[Weakness]
* The technical novelty is not significant, as the main objective is essentially the same as Actor-Controller State from [Lamb et al.].
* The novelty compared to the prior work [Lamb et al.] is not clearly explained.
* No empirical comparison to the closely related prior work [Lamb et al.].
* Some of the empirical results are not convincing.

**Summary Of The Paper:**

This paper proposes to learn a multi-step inverse model [Lamb et al.] to learn a representation that ignores exogenous information (i.e., uncontrollable information) for offline RL. In addition, this paper introduces several temporally-correlated and diverse visual distractors on top of the v-d4rl dataset to investigate the quality of the representation in RL. The results show that the proposed method outperforms several self-supervised learning methods including CURL and DRIML.

**Summary Of The Review:**

Although this paper demonstrates good results on offline RL with exogenous information, the main idea seems to mostly come from the prior work [Lamb et al.], which is not clearly stated in the current manuscript. The paper would benefit from making a clear distinction from the prior work and adding a comparison to the prior work.

---

> ### Author Response · Authors · 2022-11-15
> **EMPIRICAL COMPARISON TO PRIOR WORK LAMB ET AL. (2022)**
>
> Please see the generic response to all reviewers.
>
> We included a generic response in section L.1 of updated draft comparing the ACRO objective with
> Lamb et al. (2022), both empirically and theoretically. We included details on how ACRO differs
> from prior works, and why an approach based on ACRO can be generally applicable, compared to
> the limitations in Lamb et al. (2022).

---

> ### Author Response · Authors · 2022-11-15
> **EXPERIMENTS WITHOUT ANY EXOGENOUS NOISE; ACRO WORKS BETTER EVEN WITHOUT ANY EXOGENOUS INFORMATION**
>
> Question : "It would be more comprehensive to include a noise/distractor-free setting to get an idea
> of how much each method suffers from visual distractors."
>
> Response : We actually included results in table 2 where we do not consider any exogenous noise
> or distractors at all. All the results provided in 2 are based on the no-distractor setting from the Lu
> et al. (2022a) pixel-based offline benchmark. We label this as the EASY-EXO setting, with other
> more difficult exogenous noise datasets as in MEDIUM-EXO and HARD-EXO based results, that are
> extended, and provided as new benchmarks in this work.
>
> We would like to emphasize here that even though the main focus of the work is on exogenous
> information in offline datasets, our results as in table 2 also highlights that ACRO can be generally
> applicable for representation learning. While we focus mostly on exogenous information in the
> description of our setting, we should have emphasized the scope of the ACRO technique is more
> broadly applicable in the representation learning literature for RL.

---

> ### Author Response · Authors · 2022-11-15
> **NORMALIZED PERFORMANCE FIGURES**
>
> We apologize that some of the normalized or averaged results that were presented in
> the paper were not clear. We originally included details of the averaging in the caption of figures.
> For further clarification, and since it would be helpful for other reviewers as well, we provided an
> explanation of the normalized or averaged results figures in section L.4 in updated draft.

---

> ### Author Response · Authors · 2022-11-15
> **EXPLANATION OF RECONSTRUCTION FIGURES**
>
> Feedback : "Although I appreciate the analysis through reconstruction (Figure 7), it’s unclear
> whether the proposed method is clearly better than DRIML."
>
>
> Response : The reconstruction for ACRO removes exogenous information more than DRIML.
> Note that blurring of the controllable part does not necessarily mean that there is less information
> about the controllable part in the encoder. Specifically for such locomotion tasks, only capturing
> certain pixels for each joint is actually sufficient to predict the actions accurately. Therefore, in
> principle the controllable part can be even more noisy while still encoding sufficient information for
> control. This combined with the fact that ACRO achieves better downstream performance on both
> with and without distractors case suggests that it is able to remove exogenous information and retain
> endogenous information better than DRIML. Another case in point here is the DrQ reconstructions,
> which blur out most of the agent yet achieve similar performance to DRIML.

---

> ### Author Response · Authors · 2022-11-15
> **Summary of Individual Responses to your questions :**
>
> Thank you for your insightful questions and comments. We hope our generic responses to all reviewers and updated manuscript (which we provide as generic response since it would be helpful for all reviewers) would help you re-evaluate the score of our paper.
>
> If there are any other questions, please let us know and we would be happy to provide clarifications. Below is a brief summary of your individual questions as well :
>
> 1. Comparison to Lamb et al., 2022 : Please see generic response. We empirically and theoretically compared to Lamb et al., 2022. We hope the experimental results showing why ACRO still performs better than Lamb et al., would help clarify your concerns about prior works.
>
> 2. Justification of why we think ACRO still performs better without any exogenous noise in datasets. To repeat, our primary focus in this work is to learn robust representations when different types (difficulty) of exogenous noise may be present in offline datasets. We show ACRO performs significantly better in all cases, and table 2 is a preliminary step showing how ACRO objective performs even without any noise.
>
> 3. Explanation of experiment result figures + reconstruction figures.

---

> ### Author Response · Authors · 2022-12-05
> **Following Up On Our Responses. Any further questions/feedback?**
>
> Dear Reviewer,
>
> We have tried to address all your questions/feedback in our response (please see the general responses, summary and individual responses below along with the additional experimental results that you asked for in the updated appendix of this manuscript). Our additional experiment results comparing to Lamb et al., 2022 that you asked for shows the significance of ACRO compared to prior methods. We also tried to explicitly address how ACRO is different compared to the work of Lamb et al., 2022 (theoretically) and the significance of our work in presence of the different exogenous information based offline datasets that we use (empirically).
>
> We wanted to know if our responses have addressed all your concerns, and whether you would be willing to re-evaluate the score of our work?
>
> If there are any further questions/comments from you, we would like to address them too and improve our work accordingly.
>
> Looking forward to your reply.
>
>
> Thanks.

---

### Official Review · Reviewer_yx3d · 2022-10-24

**Confidence:** 4
**Correctness:** 2
**Technical Novelty And Significance:** 2
**Empirical Novelty And Significance:** 2
**Recommendation:** 3

**Clarity, Quality, Novelty And Reproducibility:**

The usage of exogenous does not match the typical usages this reader is familiar with. In particular, in control/RL literature exogenous is typically used to describe elements that cannot be controlled (but might still affect control), and in causal literature, it describes elements that have no incoming causal links from the variables of interest. It seems like "irrelevant" or "distractor" information would be a more appropriate term.

This work repeatedly cites Guaranteed Discovery of Controllable Latent States with Multi-Step Inverse Models (Lamb et al 2022), which appears to approach the same problem with the same solution. If this work is by the authors, then this is disingenuous since the authors' claims are being supported by citations of the author's own work. Furthermore, the claims being supported, such as the resolving of single-step issues by using a multi-step predictor, appear to be made without strong support.

The authors contrast this work from others using inverse dynamics (ID) models by saying that this is the first which uses ID for representation learning instead of exploration. Not only is this not the case after a short search (Integrating State Representation Learning Into Deep Reinforcement Learning), but just because the ID representations are used for exploration, and often with more complex and refined techniques. If the authors do not write this piece of prior work, it would appear that this work introduces nothing new to the community.

The proposed offline RL datasets are the same (at least in conceit) as those described by Learning invariant representations for reinforcement learning without reconstruction (Zhang et. al. 2020), except that the prior work was introduced outside of the context of offline RL.

**Details Of Ethics Concerns:**

I'm not sure the level reaches that for ethical review, but I am concerned by a paper support its claims by an arxiv paper of the same work.

**Strength And Weaknesses:**

Strengths:

This work investigates an important question of how to construct representations based on controllability and introduces a method for doing so.

This work implements a large number of baselines, which would be useful if released as code.
It would have been nice to include in the introduction some indication of how this work innovates over existing representation learning methods since inverse dynamics models have been heavily used in RL and model-based RL.

Weaknesses:
The description of an MDP actually describes a PO-MDP, with an exogenous block context. In addition, the typical formulation of the agent carries \pi(a|x_{1,...t}), or the history of past observations. Policies conditioned on the current observation require the "rich observation" assumption, which should be stated in 2.1, not 2.2.

There is an unstated assumption that the policies from which the data is collected are also invariant to the exogenous information, otherwise, information about the action between two states would result in improved performance. This is actually a significant issue in the offline RL setting since we don't have control over the collected data.

The work claims that the multi-step objective allows for more diverse representations because it forces the representation to encode long-range dependencies. However, only the first action in the sequence is added, so it is entirely unintuitive as to why the representation would have to be different. In the example of the action being stored in the observation, this would still be the case even if t+k is given. Furthermore, this representation introduces its own issues, since the problem itself is now ill-defined: suppose that there are multiple action sequences between s_t and s_{t+1}. In this case, the inverse dynamics model would have to output an ambiguous probability which would be dependent on the frequency one path was taken over another in the dataset. It is not clear what representation this would encode. Figure 2 also shows x_t and x_{t+k} with the same image.


A clear weakness of this work is that even though it equates exogenous with irrelevant, it is actually likely to remove information that could be task-vital. This is because the inverse dynamics model only needs to capture sufficient information to predict the first action between a pair of states, and can lose any information once it can make that prediction. Suppose that we are in an object manipulation domain where the agent uses an arm to grasp a block. The representation has no reason to capture the block because only the arm is necessary to predict the action. None of the experiments appear to capture these kinds of relationships

**Summary Of The Paper:**

By defining exogeneous information as information irrelevant for control, this work works to learn a representation that removes these features in the context of offline RL. These representations are learned by taking a latent space learned through inverse dynamics modeling. It also provides a set of benchmarks for offline RL by adding background videos to Mujoco tasks. The key insight of this method is to predict the first action of the sequence connecting the first state and a state k steps in the future.


**Summary Of The Review:**

I propose to reject this work due to lack of novelty since inverse dynamics models are a common choice for state representation learning. It also has some concerns with the blind format, since it repeatedly cites works that are likely by the same authors. Last, the writing and definitions are not entirely consistent with the motivation.

---

> ### Author Response · Authors · 2022-11-15
> **INVERSE DYNAMICS IN RL**
>
> Feedback : "It would have been nice to include in the introduction some indication of how this work
> innovates over existing representation learning methods since inverse dynamics models have been
> heavily used in RL and model-based RL."
>
> Response : We would like to thank you for bring this question, and highlight that inverse dynamics
> models have appeared in the literature in the past Pathak et al. (2017); Efroni et al. (2021); Lamb et al.
> (2022). Please see how response on this in section L.1 where we compare how the ACRO is different
> compared to past works in terms of novelty, and significance in offline RL. We emphasize that even
> though objective-wise ACRO is a variation of past works, the context to which we study this in
> presence of exogenous noise, and to show the capability of learning generic robust representations
> that can be used in any RL algorithm, can be of practical importance. In addition to discussions
> around this in generic responses and in section L of updated appendix, we would be happy to answer further questions on novelty
> and significance of ACRO.

---

> ### Author Response · Authors · 2022-11-15
> **POMDP AND EXOGENOUS BLOCK MDP**
>
> The decision process we consider here is an MDP in the sense that the observations are Markovian;
> the observation at time step $x\_t$ does not depend on the history conditioning on $x\_{t−1}$ and $a\_{t−1}$. We
> will highlight that in section 2.2 to avoid confusions and miss-understandings in the future. Indeed,
> in this work we do not study the problem of representation learning in POMDPs (in the presence of
> high-dimensional observation data). This problem is of great interest and challenge.

---

> ### Author Response · Authors · 2022-11-15
> **ABOUT MULTI-STEP INVERSE DYNAMICS OBJECTIVE**
>
> Question : "The work claims that the multi-step objective allows for more diverse representations
> because it forces the representation to encode long-range dependencies. However, only the first
> action in the sequence is added, so it is entirely unintuitive as to why the representation would have
> to be different."
>
> Response : What is the specific reference on the claim that ACRO allows for “more diverse representations”?
> We talk about our newly proposed datasets having diverse exogenous distractors, but we do not
> claim that the representations are more “diverse”.
>
> Feedback : "In the example of the action being stored in the observation, this would still be the case
> even if t+k is given".
>
> Suppose that $\tilde{x}\_t = (x_t, a\_{t-1})$, so the observation stores an original observation along with the most recent action.  Then consider the 1-step inverse model: $p(a\_t |  \tilde{x}\_t, \tilde{x}\_{t+1}) = p(a\_t | x\_t, a\_{t-1}, x\_{t+1}, a\_t)$.  The optimal solution can thus completely ignore $x$.  Note that you can't use $\tilde{x}\_t = (x\_t, a\_t)$ as the observation for a counter-example because it violates causality.  You are right that recording many recent actions will cause issues for multi-step inverse models, so we updated the text to reflect this.
>
> Feedback : “Furthermore, this representation introduces its own issues, since the problem itself
> is now ill-defined: suppose that there are multiple action sequences between $s\_t$ and $s\_{t+1}$. In this
> case, the inverse dynamics model would have to output an ambiguous probability which would be
> dependent on the frequency one path was taken over another in the dataset. It is not clear what
> representation this would encode.”
>
> You are right that there are multiple action sequences for travelling from one state to another. Even in
> the case of a simple empty gridworld, there are two equally valid actions for going from the top-left
> to the bottom-right in as few steps as possible. We can either go down, or go right on the first step.
> As another example, if the agent starts in the top-left corner, going left or going up will both keep
> the agent in the same position. Thus the negative log likelihood of the optimal multi-step inverse model will indeed
> not be zero for most environments. Can you explain further why you believe this is a problem?

---

> ### Author Response · Authors · 2022-11-15
> **ACRO and TASK RELEVANT INFORMATION**
>
> We would like to thank you for such insightful comments and feedback, on whether
> ACRO objective can ignore task relevant information. If this is true, then we agree this would
> definitely be a key weakness of our work. We provided a generic response to this to all reviewers and in section L.3 of updated draft
> since this would also provide better insights to other reviews.
>
> Please see the generic response to all reviewers.

---

> ### Author Response · Authors · 2022-11-15
> **COMPARISONS TO PRIOR WORK LAMB ET AL. (2022)**
>
> The question about comparisons to prior work Lamb et al. (2022) have been brought up
> by other reviewers too. We provided a generic response, along with additional experimental results,
> and detailed comparison between ACRO and Lamb et al. (2022) in section L.1. We emphasize that
> there are few major differences between the two works, and how Lamb et al. (2022) can be practically limited as it only recovers a small underlying tabular MDP for planning and does not show any
> significance in a practical deep RL setting. In contrast, ACRO pre-trains robust representations that
> can then be used in any existing deep RL algorithm, which makes the approach more generic, and
> not requiring any specific discrete bottlenecks as in Lamb et al. (2022).

---

> ### Author Response · Authors · 2022-11-15
> **PREVIOUS WORKS WITH DISTRACTORS IN RL ZHANG ET AL. (2020)**
>
> Feedback : "The proposed offline RL datasets are the same (at least in conceit) as those described
> by Learning invariant representations for reinforcement learning without reconstruction (Zhang et.
> al. 2020)"
>
>
> Response : We would like to emphasize how the exogenous distractors considered in this work is
> quite different than what appeared in past literature. In section L.5 we provided a generic response
> to all reviewers about the significance and impact this work can have, towards practical scalability
> of RL in real world. Prior works in the online setting, as in Zhang et al. (2020) have only considered
> time uncorrelatred distractors whereas we build primarily on the theoretical studies of exogenous
> RL Efroni et al. (2022b; 2021) to further provide an offline dataset benchmark with exogenous
> information. We would like to note that we will be releasing these datasets so they can be used as
> benchmarks, for future research and reproducibility, beyond the novelty of specific exogenous noise
> that we considered.

---

> ### Author Response · Authors · 2022-11-15
> **ASSUMPTIONS ON DATASETS**
>
> We assume that the reward function is free of any exogenous noise; ie, even if observations consist of exogenous noise, the rewards depend on the raw endogenous states, r(s,a).
>
> We clarify that a purely random policy can be free of any exogenous noise in observations, since the action selection of a random policy does not depend on the observations. Any other non-random polciy would depend on the exgeonous noises π(a|x) where x consists of both exogenous and endogenous part of the state space. However, when we pre-train representations, and then use it in an RL algorithm, we are indeed learning a policy π(a|z) and depending on the representation being learnt, the resulting optimal policy would depend on the extent to which z depends on the exogenous noise. We show that the latents recovered from ACRO are robust and filters out exogenous noise compared to other baselines considered.
>
> There are no assumptions being made on the requirement of the dataset. The offline datasets are collected with SAC policy in presence of exogenous noise, and can have different data distributions depending on the data collecting policy (expert, medium-expert or medium datasets). We emphasize and repeat here that the offline datasets consist of all pixel-based observations.
>
> We assume we are in an exogenous block MDP setting Efroni et al. (2021); Du et al. (2019) that has been studied in the theoretical RL community before. In this work, we provide a practical algorithm that can be scaled easily to existing deep RL algorithm pipeline, where learning representations play a key role. The most important contribution of the work is that ACRO can be generally applicable in settings with different types of exogenous information, under block MDP assumptions Du et al. (2019)
>
> On the issue of assumptions in datasets : There are no assumptions being made on the requirement
> of the dataset. The offline datasets are collected with SAC policy in presence of exogenous noise,
> and can have different data distributions depending on the data collecting policy (expert, mediumexpert or medium datasets). We emphasize and repeat here that the offline datasets consist of all
> pixel-based observations.

---

> ### Author Response · Authors · 2022-11-15
> **Summary of Individual Responses to your questions**
>
> Thank you for your insightful questions and comments. We hope our generic responses to all reviewers and updated manuscript (which we provide as generic response since it would be helpful for all reviewers) would help you re-evaluate and improve the score of our paper.
>
> If there are any other questions, please let us know and we would be happy to provide clarifications.
>
> Below is a brief summary of your individual questions as well :
>
> 1. Justification of novelty, even if there are prior works with inverse dynamics based objectives in RL. We provide clarification of the significance of our work (in generic response as well), and why we think ACRO can be useful to the offline RL community.
>
> 2. Clarifications on the multi-step inverse dynamics objective, and what ACRO like objective can capture in terms of representations.
>
> 3. Clarity on POMDP and EX-BMDP
>
> 4. Clarification on whether ACRO removes task relevant information (also in generic response)
>
> 5. Comparisons with Lamb et al., 2022 both theoretically and empirically, justifying why ACRO objective may be better suited for offline deep RL compared to Lamb et al., 2022
>
> 6. Clarification on how the exogenous noise datasets in offline RL that we work with here is different compared to what appeared in past literature in online RL.
>
> 7. Clarifications on assumptions and limitations

---

> ### Author Response · Authors · 2022-12-02
> **Feedback on the updated manuscript (Rebuttal Section in Appendix)**
>
> Dear Reviewer,
>
> Thank you for your feedback on our paper. Please let us know if our responses, additional experimental results and justifications have addressed your primary concerns.
>
> If there are any further questions/feedback, please let us know too, so we can address all your concerns. We hope the Summary of Our Responses (please see below) to all your questions are helpful for you to re-evaluate the score for our work.
>
>
> Thanks.

---

### Official Review · Reviewer_M6qw · 2022-10-25

**Confidence:** 3
**Correctness:** 3
**Technical Novelty And Significance:** 2
**Empirical Novelty And Significance:** 3
**Recommendation:** 6

**Clarity, Quality, Novelty And Reproducibility:**

The paper is clearly written with reasonable quality.

The novelty is relatively limited since the proposed algorithm is directly extended from prior work.

The authors have provided code and detailed experiment settings.

**Strength And Weaknesses:**

### Strengths

1. The idea is neat and easy to implement, intuitive and theoretical grounded.
2. The empirical results are promising, covering multiple scenarios.
3. Good presentation and visualization of the ideas and results.
4. Interesting theoretical insights.

### Weaknesses and Questions:

1. The novelty of the proposed method is relatively limited, as a continuous-control extension of prior work (Lamb et al. 2022).
2. Following bullet 1, I am wondering whether the authors can comment more on the theoretical comparison with Lamb et al. 2022. Without the bottleneck constraint, will the learned representation be guaranteed to discard exogenous information? If not, the claims of Section 3 do not hold for the proposed algorithm, right?
3. The algorithm adopts a process of offline representation pretraining + policy finetuning with frozen representation. However, such representation learning may highly depend on coverage of the offline dataset. I hope there could be more discussion on the requiremenet of data/exploration, both in theory and in experiments.
4. The representation is only tested with offline RL finetuning, while it is not clear whether the learned representation is sufficient for learning optimal policies in an online manner. Intuitively, the latter would be harder and thus more interesting (the representation should be not only good for modeling optimal behaviors, but also good for exploration). If the claims hold that the learned representation is controllable, it should be able to learn a policy by interaction. Combined with bullet 3 above, I am worried that if the offline dataset does not have good coverage, downstream online policy learning will be hard.

**Summary Of The Paper:**

This paper proposes a multi-step inverse model to learn the representation for RL problems with offline data. The proposed method ACRO predicts the current action based on the current state and a future state, in a reward-free manner. ACRO can learn controllable states while keep invariant to exogenous information. Experiments in various settings show that ACRO outperforms prior methods. Theoretical analysis shows the benefits of ACRO. The paper also provides several benchmarks to evaluate representation learning in RL.

**Summary Of The Review:**

This is an interesting paper with good empirical results. But I am hesitating to recommend it for acceptance, because 1) the method is a direction extension from prior work (and in the meanwhile it may lose the theoretical guarantees of prior work), and 2) there is a lack of in-depth discussion on the assumption and limitation of the current method, including requirement for offline dataset. I may missed some information in the paper and appendix. I would consider increase the ratings if the authors can provide convincing analysis/results for the above points.

---

> ### Author Response · Authors · 2022-11-15
> **KEY ASSUMPTIONS AND CLARIFICATION**
>
> Thank you for the feedback, and apologies if the key assumptions made in the work were not clear
> from the main text. We repeat here some of the key yet minor assumptions :
>
> 1. We assume that the reward function is free of any exogenous noise; ie, even if observations
> consist of exogenous noise, the rewards depend on the raw endogenous states, r(s,a).
>
> 2. We clarify that a purely random policy can be free of any exogenous noise in observations,
> since the action selection of a random policy does not depend on the observations. Any
> other non-random polciy would depend on the exgeonous noises π(a|x) where x consists
> of both exogenous and endogenous part of the state space. However, when we pre-train representations, and then use it in an RL algorithm, we are indeed learning a policy π(a|z) and
> depending on the representation being learnt, the resulting optimal policy would depend on
> the extent to which z depends on the exogenous noise. We show that the latents recovered from ACRO are robust and filters out exogenous noise compared to other baselines
> considered.
>
> 3. There are no assumptions being made on the requirement of the dataset. The offline datasets
> are collected with SAC policy in presence of exogenous noise, and can have different data
> distributions depending on the data collecting policy (expert, medium-expert or medium
> datasets). We emphasize and repeat here that the offline datasets consist of all pixel-based
> observations.
>
> 4. We assume we are in an exogenous block MDP setting Efroni et al. (2021); Du et al. (2019)
> that has been studied in the theoretical RL community before. In this work, we provide a
> practical algorithm that can be scaled easily to existing deep RL algorithm pipeline, where
> learning representations play a key role. The most important contribution of the work
> is that ACRO can be generally applicable in settings with different types of exogenous
> information, under block MDP assumptions Du et al. (2019)

---

> ### Author Response · Authors · 2022-11-15
> **COMPARISONS WITH PRIOR WORK LAMB ET AL. (2022)**
>
> Thank you for the question about comparisons with prior work Lamb et al. (2022). We empirically
> compare ACRO with the AC-State objective Lamb et al. (2022) along with details on how ACRO
> is different compared to previous works (also mentioned in related works).
>
> Please see the generic response to all reviewers about this in section L.1 in updated draft, and generic response to all reviews ("Comparisons between ACRO and AC-State, empirical and theoretical comparisons"), since we believe clarifying these differences
> would be helpful for all reviewers.

---

> ### Author Response · Authors · 2022-11-15
> **DATA COVERAGE AND ROLE OF EXPLORATION IN OFFLINE RL**
>
> Response : Role of data coverage in offline representation learning: we agree that lack of coverage
> would degrade the algorithm. This, however, is a well known fact in the offline RL setting. The fact
> that the concentrability coefficient is finite implies there is a good dataset coverage. We do agree
> that studying the theoretical aspect of this question is needed. In this work, we mostly focused on
> this problem from its empirical side and established the failure of existing approaches, as well as
> offering a fix (that works under the dataset coverage assumption.
>
> Further, the online RL problem with exogenous noise is also of interest. As of now, there is no
> algorithm with provable guarantees for the general RL problem with exogenous noise (there are
> some solutions under different sets of assumptions as we elaborated on in the related work section).
> Our work does not tackle this challenging problem, but study the offline aspects of its; a prevalent
> problem from a practical perspective.
>
> Experiments : We varied the amount of data coverage in the offline datasets, by taking the of
> times, the data collecting policy takes random actions. We vary the % of random actions from
> 10% to 50% where we assume that more randomness in actions taken by an expert policy means
> higher state space coverage. We follow the same experiment setup as before, and now show how the
> performance of ACRO (and two other baselines) varies as we have lower to higher coverage in the
> datasets, as in figure 30, in updated appendix including author rebuttal.
>
> Figure 30 in updated draft : Experimental results comparing different representation learning methods as we have
> varying amounts of coverage on the datasets. We make the assumption that adding more random
> actions to the datasets means the dataset has a higher coverage of state and action space. We show
> that as the % of random actions on datasets increases, the performance of each method degrates,
> especially the ones like DRIML that are independent of action prediction. In contrast, all other
> methods that rely on action prediction, suffer less depending on the amount of random actions
> that may exist in the datasets. This shows a potential for representation objectives based on action
> prediction that the performance of these methods degrades least even if the dataset quality is poor
> (where in our case, higher coverage due to more random actions means the dataset is degrading from
> an expert dataset to more of a random dataset).

---

> ### Author Response · Authors · 2022-11-15
> **ACRO IN ONLINE RL FOR REPRESENTATION LEARNING AND EXPLORATION**
>
> Thank you for the comment and providing suggestions on how we can extend the paper. We would
> like to note here that the problem of considering the online setting, and extending beyond our current
> focus on offline RL is that the online setting is dependent on exploration too (since representation
> learning in online setting is inherently coupled with the amount of exploration) and this is something,
> while interesting, is perhaps beyond the scope of our current work. In future, we would definitely
> consider extending ACRO to the online setting, which might additionally require a mechanism for
> exploration while ignoring exogenous noise. In our current work, we are mostly studying robustness to exogenous information given datasets, and we study the exploration question there through
> assumptions on the role of coverage in the datasets (as you noted in the previous question). As
> pointed out rightly, in offline RL this is tested based on data coverage in the data distributions, and
> our response and results above is an attempt towards addressing this with ACRO.

---

> ### Author Response · Authors · 2022-11-15
> **Summary of Individual Responses to your questions**
>
> Thank you for your insightful questions and comments. We hope our generic responses to all reviewers and updated manuscript (which we provide as generic response since it would be helpful for all reviewers) would help you re-evaluate the score of our paper.
>
> Below is a brief summary of our responses :
>
> 1. Experimental results and theoretical justification on the data coverage issue in offline RL
>
> 2. Comparisons with prior work, Lamb et al., 2022 (both theoretically and empirically) - also see generic responses please
>
> 3. Why ACRO in online RL is perhaps beyond scope of our work
>
> 4. Clarifications on the key assumptions being made, which are required theoretically, but not empirically.

---

> ### Comment · Reviewer_M6qw · 2022-12-02
> **Thank you for the response**
>
> Thank you for providing the detailed response and additional results. My concerns are partially addressed. Although the technical novelty of this work still looks limited to me, it is different from prior work and does show interesting empirical results. These results and insights can be useful for the community. I have thus increased my rating.

---

> > ### Author Response · Authors · 2022-12-02
> > **Thank you for improving the score**
> >
> > We greatly appreciate your constructive and detailed feedback on our work. Thank you for acknowledging the changes and additional results, and that you find the work to be interesting and insightful for the community.
> >
> > If there are any further questions/feedback that you would like us to address, please let us know.
> >
> > We would definitely add these discussions with new results in the final version of the manuscript (along with dataset release, which we believe can be used as offline benchmarks in the community).
> >
> >
> > Thanks.

---

### Official Review · Reviewer_GMBs · 2022-10-25

**Confidence:** 4
**Correctness:** 4
**Technical Novelty And Significance:** 3
**Empirical Novelty And Significance:** 3
**Recommendation:** 6

**Clarity, Quality, Novelty And Reproducibility:**

Clarity:
Overall, this paper is written clearly. The tables and figures are easy to read and convey information efficiently. I especially like table 1 which helps to compare different representation learning approaches.
Also, it will be great to explain the importance of the Bellman completeness more clearly and intuitively.

Quality:
The quality is good with the technically solid method and impressive experiment results.
However, the experiments are only conducted on locomotion tasks. It will be great to show results in D4RL kitchen tasks, especially because these tasks require object manipulation, and the proposed representation approach may fail in this scenario. I'm curious whether the proposed method can be modified or extended to the manipulation tasks.

Novelty:
The novelty is okay but not very great, because the multi-step inverse model has been studied for a well in the RL area, and this work is just to adapt it from standard RL to offline RL.

Reproducibility:
The code is provided but the new datasets have not been released yet. Considering there is detailed information about hyper-parameters in the Appendix, the reproducibility is fine as long as the datasets can be released later.

**Strength And Weaknesses:**

Strengths:
The authors introduced the multi-step inverse model in offline RL problems and proved that Bellman completeness can be achieved via the representations without exogenous noise.
The experimental results support the claims well. In the datasets of noisy observations, this paper demonstrates the significant advantage of the proposed representation learning approach.
The newly introduced datasets for offline RL can be useful for the community.

Weaknesses:
The major concern is about the limitation of the representation only containing control-relevant information. How about there are other moving entities in the observations, which are not controllable by the agent but are still very important for the success of the policy? For example, a robot navigates busy streets. Pedestrians are not controllable by the robot, but the robot should consider the status of the pedestrian to avoid collisions. Will the proposed representation learning approaches fully ignore pedestrians and make errors in the decision?

Another example is robotics manipulation tasks. The robot hand should manipulate the objects, so the object position and orientations are important information for the control policy. But the proposed method may tend to ignore the object information. This is problematic and limits the scope of the proposed method.



**Summary Of The Paper:**

This paper considers learning a good representation for offline reinforcement learning algorithms with pixel-based visual observation space. The authors aim to extract the representation, which ignores any control-irrelevant information. The authors choose multi-step inverse models to learn the observation representation because the multi-step action prediction for learning exogenous-invariant representations has been shown useful in  RL theory community.

The proposed method ACRO first trains the image encoders so that the features can be used to predict multiple actions between two observations. With the well-trained encoder, the downstream offline RL algorithm TD3+BC takes in the learned representation as policy input to output actions.

In order to evaluate the robustness of the proposed method for pixel observations, the authors introduce new datasets with temporally-correlated noise and diverse noise in the observations. The dataset is similar to D4RL but with some exogenous images or videos incorporated in the observations as noise.

In comparison with other representation learning methods, the proposed one significantly outperforms the baselines.

**Summary Of The Review:**

This paper presented a well-motivated representation learning method for offline RL, with great experiment performance and some theoretical foundation. But the proposed method may be only suitable for locomotion tasks, due to the limitation of the learned representation. This seems not general and widely useful.

---

> ### Author Response · Authors · 2022-11-15
> **EXOGENOUS INFORMATION AND DISTRACTOR SETTINGS FROM PRIOR WORKS**
>
> Question : Similarity of D4RL with some exogenous images or videos incorporated as noise in
> observations
>
> Response : We would like to emphasize, as briefly mentioned in L.5 that this work studies pixelbased offline RL with exogenous noises in observations. In comparison, Fu et al. (2020) provides
> benchmark datasets for raw state-based offline RL, whereas recently Lu et al. (2022a) provided pixelbased offline RL benchmarks. We extended from Lu et al. (2022a) to additionally add exogenous
> noises in observations, with different levels of difficulty : EASY-EXO is what we call the existing
> Lu et al. (2022a) benchmark; we then extend and re-collect datasets using the same pipeline as Lu
> et al. (2022a); Fu et al. (2020), but now additionally add MEDIUM-EXO or HARD-EXO exogenous
> noise during data collection with SAC policy. For example, during MEDIUM-EXO, we place random
> STL10 images in corner or side, or place fixed video in background. The fixed video background
> distractor has been studied in online RL before (Zhang et al., 2020), and we provide an extension
> of it to offline RL. In the HARD-EXO setting, we consider novel datasets and distractors, where now we have either changing video distractors in background, or have multiple agent observations
> that the agent sees, in addition to the controllable environment observation. Please see figure 8 for
> samples of what the agent exogenous observation looks like. Such datasets and distractors have not
> been studied in the literature before, and we provide a well capable representation learning approach
> to robustly learn representations under such difficult distractors.

---

> ### Author Response · Authors · 2022-11-15
> **CONCERN ON REPRESENTATION CONTAINING ONLY TASK RELEVANT INFORMATION.**
>
> Thank you for the comments and feedback, which are playing a major role in improving
> the impact of the paper. Please see general response (Does ACRO remove task relevant information?) and section L.3 in updated appendix, where we provide a generic response to all reviewers.

---

> ### Author Response · Authors · 2022-11-15
> **IMPORTANCE OF BELLMAN COMPLETENESS**
>
> Feedback : "It will be great to explain the importance of the Bellman completeness more clearly and intuitively"
>
>
> Response and Comment on Bellman Completeness : As we elaborated on in the paper, the importance of the Bellman completeness assumption was recently demonstrated in the theoretical RL
> community. Intuitively, this assumption says the following: for any function f(x, a) in the function
> class it is possible to represent its Bellman backup $r(x, a) + E\[max\_{a'} f(x', a') | x, a\]$. That is, it
> is possible to optimally learn the Bellman backup at each point of the training process. We will
> highlight this fact better in the text. Bellman completeness overall means the ability to represent the
> optimal solution that minimizes the bellman error of any function. For example, if the expected
> bellman error is small then completeness holds, however, vice versa does not necessarily hold. That
> is, the bellman error might be large because of stochasticity and not because completeness doesn’t
> hold.

---

> ### Author Response · Authors · 2022-11-15
> **ADDITIONAL EXPERIMENT RESULTS ON D4RL BENCHMARK**
>
> Thank you for the feedback on the impressive experimental results that ACRO achieves. We agree
> that all our experiments are on the locomotion tasks, based on the available offline benchmark (Lu
> et al., 2022a) and our newly proposed benchmark datasets. We definitely agree that it would be
> interesting to show whether ACRO kind of objectives can also perform well in object manipulation
> based tasks.
>
> For the D4RL kitchen task, we implemented ACRO on top of the existing CQL and TD3 + BC
> baselines for testing. However, unfortunately we find that, as also reported in past works, that none
> of the baselines actually work well on the kitchen tasks. For ACRO to work well, we would still
> require a well performing offline RL algorithm that can solve the kitchen tasks based on the learnt
> representations, and we found that the existing baselines are not significantly well performing on
> this task.
>
> However, we also ran ACRO on the D4RL benchmark (Fu et al., 2020). We would like to first
> mention that D4RL is only a raw state based benchmark and not pixel based observations. As such, even though past works have shown results for learning representations in D4RL, learning in raw
> state based environments does not play a significant role for learning representations. Moreover, exogenous noise in observations are primarily for rich observation MDPs with pixels. Despite that, we
> implemented ACRO on top of the TD3 + Baseline Fujimoto & Gu (2021b) which has been shown
> to work well on raw state based D4RL benchmark.
>
> In figure 29 in updated draft we show results on two different
> domains with two different datasets, where on top of the existing baseline, we learn representations with ACRO. Experimental results show that we can still achieve some marginal improvements
> over baselines by learning with ACRO. We would like to mention here that even though there are
> marginal improvements, this result may not be significant in the context of our work, since there
> is no exogenous noise present here, and ACRO is primarily proposed for pixel based RL tasks, to
> demonstrate ability of learning robust representations by ignoring the irrelevant information.
>
> Figure 29 (updated draft) : Experimental results on the D4RL benchmark comparing ACRO with the baseline TD3
> + BC method. We mention here that previously we do not do experiments with D4RL since the
> environments are primarily raw state based, instead of pixel based observations. In this work, we
> primarily propose ACRO for learning robust representations that can ignore exogenous information
> from pixel observations. We include this result since it is asked by the reviewer on why we do not do
> experiments with D4RL. We include this preliminary result, and demonstrate that ACRO can work
> well in raw state D4RL too, even though that is not the primary contribution of our work.

---

> ### Author Response · Authors · 2022-11-15
> **NOVELTY OF WORK AND RELEASE OF OFFLINE DATASETS**
>
> Thank you for your feedback on this, and on raising the question on the novelty of the
> work. We provided several generic responses in section L clarifying how this work is different from
> what appeared in the past literature. Most importantly, please see section L.5 where we highlight
> how this work can be significant to the community. We understand the issue that inverse dynamics
> models have appeared in the past literature. We emphasize the context of which such objectives have
> been used (mostly for online RL and exploration), whereas we are primarily focused on learning in
> presence of exogenous noise (of varying difficulty) and found an instantiation of multi-step inverse
> models that be very significant in learning from such datasets in offline RL. Furthermore, theoretical
> works have studied variations of the ACRO objective theoretically (Efroni et al., 2021; Lamb et al.,
> 2022), while we provide a practical algorithm based on pre-training representations that can be used
> off the shelf in any existing deep RL based offline RL pipeline. Our primary focus is to provide such
> findings, along with the novel datasets, which we believe would be significant for stepping towards
> practical scalability of RL in the real world.

---

> ### Author Response · Authors · 2022-11-15
> **Summary of Individual responses to your question**
>
> Thank you for your insightful questions and comments. We hope our generic responses to all reviewers and updated manuscript (which we provide as generic response since it would be helpful for all reviewers) would help you re-evaluate the score of our paper.
>
> If there are any other questions, please let us know and we would be happy to provide clarifications. Below is a brief summary of your individual questions as well :
>
> 1. How the exogenous noise datasets we work with in this paper is different compared to what appeared in past literature
>
> 2. Clarification on whether ACRO removes task relevant information or not.
>
> 3. Importance and intuition on Bellman completeness
>
> 4. Additional experimental results on D4RL benchmark, and clarification on why we do not primarily use this benchmark
>
> 5. Novelty and significance of the work, and how the offline datasets that we release in this work can be significant to the community.

---

### Author Response · Authors · 2022-11-15
**Thank you to all reviewers; General Responses and updated draft including Author Rebuttal Section (Section L in Appendix)**

We thank all the reviewers for helpful and detailed feedback, which we have already taken advantage
of to improve the paper. We thank the reviewers for praising the strong experimental results, new
theory that this paper provides, along with an important contribution to the community by providing
new offline RL datasets containing varying levels of exogenous information in the datasets. We plan
to release these datasets, along with our implementation, learning robust representations in offline
RL, which we hope will be significant to the RL community.

Note : After the rebuttal period, we will merge the responses and figures from the sections below into
the main draft. Currently, we include these as separate sections to address the reviewer comments
explicitly.

We provide a common response to all reviewers, since some of the feedbacks are useful for overall
improvement and understanding of the paper for everyone

---

### Author Response · Authors · 2022-11-15
**COMPARISON WITH AC-STATE ( LAMB ET AL. (2022)) AND ACRO - Empirical Comparison**

We would like to thank the reviewers for highlighting the need to produce comparisons between our
proposed method and the AC-State algorithm (Lamb et al., 2022). The reviewers rightly pointed
out, as we already addressed in the paper, the similarity with the AC-State objective from Lamb
et al. (2022) and ACRO. We would like to highlight here the main differences between AC-State
and ACRO and what makes ACRO critical for learning robust representations that can be used for any RL algorithm.

The AC-State objective uses a multi-step inverse model with an additional conditioning on the k-th time-step, unlike ACRO
objective as in equation 1. We would also point out that the theoretical support for multi-step inverse models for representation learning largely comes from Efroni et al. (2021), an ICLR2022 paper which predates the AC-State paper.

The work of Lamb et al. (2022) primarily has a different purpose and scope, where the goal is
to recover latents with perfect accuracy, for then using it for exact planning in a tabular MDP, an
approach which only scales to very small discrete systems. Most importantly, Lamb et al. (2022)
learns 100 discrete latent states using controlled exploration, by the use of a discretization bottleneck
Van Den Oord et al. (2017) and primarily addresses recovering discrete ground truth latent states.
Lamb et al. (2022) considers only environments with well under 100 discrete endogenous states. In
comparison, ACRO considers offline RL with general continuous hidden states, which is much more
general. Lamb et al. (2022) does not address how the learnt representation can be used in any RL
algorithm, and the controlled exploration is mostly required to recover the ground truth states. The
discrete bottleneck along with the requirement for exploration (Lamb et al., 2022) are not generally
applicable for any deep RL algorithm to be used off-the-shelf with learnt representations. Lamb et al.
(2022) considered moderate state space sizes and used planning on top of the latents. In contrast,
we consider larger scale problems and train an RL algorithm on top of the representation. Hence,
unlike prior works, we study larger scale problems in offline deep RL.

In contrast, ACRO addresses how to effectively use the multi-step inverse objective in situations
where there are infinite continuous latent states and uncontrolled exploration. ACRO shows the
capability of learning robust representations in presence of exogenous noise, that can be used for
learning continuous representations, and does not require any use of a discretization bottleneck.
While the objective remains the same, the data and representations here differ in critical ways,
greatly broadening potential applications. Together these changes provide results in regimes the
AC-state paper does not address. In ACRO, the focus shifts from doing exact planning (DP or
Dijkstra) in a small tabular-MDP to extracting a representation that enables learning a policy via
Offline-RL with a learned value function.

In figure 25 and 26 in updated appendix, we empirically compare ACRO and a version of ACRO with a discretized
bottleneck representation, an extension of Lamb et al. (2022) that can be used in an offline RL algorithm. We compare both these methods under two of the hardest HARD-EXO offline datasets of varying data distributions. The primary difference is the use of the small discrete representation required by AC-State. In most cases, ACRO works significantly better in the offline RL task with
HARD-EXO noise.

---

### Author Response · Authors · 2022-11-15
**COMPARISON WITH AC-STATE (LAMB ET AL, 2022) AND ACRO - Theoretical Claims**

In addition to the empirical comparison included in the updated appendix, we provide a discussion on the  theoretical claims comparing ac-state (Lamb et al, 2022) and ACRO :

We see our work as a bridge between earlier theoretical work on multi-step
inverse models and the Offline-RL literature. In terms of the novelty of theory, it is true that the
proof in Appendix A is the same as these papers, which we already stated very clearly in our paper:
“This proof is essentially the same as lemmas found Lamb et al. (2022), Efroni et al. (2021), but is
presented here for clarity.”. The proofs in Appendix B and Appendix C are novel and specifically
related to conditions for Offline-RL to succeed. To better explain the differences, we added a table
with these prior works, and we added clearer explanations of novelty to the main text.

We established new results for the setting of RL with exogenous information. Namely, that including
some exogenous information as part of the representation may violate Bellman completeness, i.e.,
the ability to optimally represent any possible Q function that should be learned during the training
process. Additionally, we showed that the agent controller representation is sufficient to represent
the optimal Q function and induces a Bellman complete class. We first showcase the challenge of
the offline RL problem with exogenous noise, and then show the advantage of the agent controller
representation.

---

### Author Response · Authors · 2022-11-15
**Novelty of Using Inverse Models for Representation Learning**

We highlight here that even though inverse dynamics models have been used previously in the literature, they are primarily used for either exploration solely (Pathak et al., 2017) or for representation learning in an online task that requires exploration (Misra
et al., 2020; Efroni et al., 2021; Lamb et al., 2022).

In contrast, in this work, we first use inverse
dynamics models in exogenous block MDPs to show the capability of learning robust representations that can then be integrated to an offline RL algorithm. As such, ACRO does not require any additional exploration and can work with any data distribution in the offline datasets. We already show results comparing ACRO to one-step inverse models (Pathak et al., 2017).

---

### Author Response · Authors · 2022-11-15
**SIGNIFICANCE OF MULTI-STEP ACTION PREDICTION VS. SINGLE ACTION PREDICTION**

We thank the reviewers for asking insightful questions about how the ACRO algorithm compares
if we use multi-action prediction up to the k-th timestep, in comparison to only predicting a single
next step action. Since ACRO already conditions on $\phi(x_{t+k})$ for k-th step in the future, it is natural
to ask how the performance varies if we predict multiple future actions compared to a single action.

The k-step action sequence prediction model, in our updated appendix including empirical comparisons between ACRO and a variation of ACRO with multiple action prediction (ie, predicting an action sequence), can be implemented using an LSTM that outputs the k actions. One reason to prefer predicting just the first action is that it is a simpler model and is computationally cheaper (as it
requires just a classifier over actions and not an autoregressive model over sequences like the LSTM).
Intuitively, we also felt that predicting the first action to reach a goal would be sufficient, because
ultimately every action along the trajectory is still predicted, but conditioned on the observation prior
to the action being taken. In an environment with stochastic dynamics, we see this as being better in
principle, because the best action to take at a given step is dependent on what has happened in the
environment. In a deterministic environment, both approaches are valid in principle.  We have added theory exploring this in Appendix C.

Nonetheless, we agree that it is important to also answer this question experimentally. Figures 27
and 28 in updated draft show comparison between ACRO and a variation of ACRO predicting multiple actions in the future. We use the same training setup, where all the representations are pre-trained in presence
of HARD-EXO noise in observations.

---

### Author Response · Authors · 2022-11-15
**DOES ACRO REMOVE TASK RELEVANT INFORMATION?**

Various reviewers expressed concerns about the agent-controller representation ignoring information that is vital to the task (pedestrians for reviewer GMBs and objects / blocks in a robot hand
manipulation task for reviewers GMBs and yx3D). However, the notion of what is in the agentcontroller representation is actually more extensive than might be imagined at first glance. Anything
that influences the actions taken by the policy needs to be captured by the representation in order
to predict the first action it took from a pair of representations. Going back to the examples above,
in the case of pedestrians, the policy used to collect the data (hopefully) avoided the pedestrians to
reach its goal. In order to properly predict this avoidance behavior (compared to presumably moving
in a straight line in the absence of pedestrians), the representation needs to encode the presence of
these pedestrians. Similarly, for the example of a robot hand grabbing a block, let us consider a pair
of states (st, st′ ), where st corresponds to the state before the object was grasped and st′ after. Then, the agent-controller representation needs to include information that pertains to the block’s position
and orientation, otherwise, it would be impossible to predict the actions governing the robot’s motion (presumably in the direction of the object), or how its joints are adjusting to grab the object.

Note that this would not be required for a simple one-step inverse model (which justifies the use of
a multi-step one) as the robot hand’s joint positions in successive states are sufficient to infer what
action was taken.

Additionally, in ACRO we also consider expert policies as a source of offline training data. In the
case of an expert policy, the knowledge about what parts of the world matter will be captured even by
a behavioral cloning policy. ACRO inherits this strength of behavioral cloning, while dramatically
outperforming it when the data comes from a random policy.

---

### Author Response · Authors · 2022-11-15
**EXPLANATION OF NORMALIZED RESULT FIGURES**

We apologize that the results presented in our paper, given we had too many results to present, might
have been confusing for reviewers to interpret. In offline RL, since our experiments are across several domains (cheetah, walker, humanoid), and for each domain, we have different dataset distributions (expert, medium-expert, medium) as generally present in offline benchmarks (Lu et al., 2022a; Fu et al., 2020), we averaged or normalized results to present the significance of ACRO across the
experiments. Given the experiments, whether we are considering MEDIUM-EXO or HARD-EXO,
we either average the results across domains or datasets. All individual experiments are run for 5
different seeds across the board. We emphasize that the normalization or averaging is done so we
could present all our experimental results, explaining significance of ACRO in a precise way, across
the board. All individual results showing improvements that ACRO achieves are included in the
appendix.

Figure 4 for example, presents results where averaging is being made across domains. We further
include individual results for each domain in the appendix. We show that ACRO achieves improvements across all domains in general, in addition to improvements across all dataset types, as shown
in figure 4.

Figure 6 shows results where the normalization is being made across the dataset types, instead of
domains, as in figure 4. Figure 6 shows results for the 3 domains (cheetah, humanoid, walker)
for different HARD-EXO types, and averaged across expert, medium-expert and medium datasets.
We show that ACRO can achieve significantly better performance compared to 5 other baselines
considered in this work.

---

### Author Response · Authors · 2022-11-15
**IMPACT IN OFFLINE RL AND SIGNIFICANCE OF THE WORK**

We would like to emphasize and repeat here, to the reviewers, the significance of our work and the
impact it can have in the community. Firstly, we propose an approach for learning robust representations, where we considered many diverse set of exogenous noise that can be present in the observations. We emphasize that we are the first to study, both empirically and theoretically, presenting a practically feasible algorithm, capable of working under different exogenous noises. We propose
an algorithm based on the multi-step inverse dynamics objective, a variation of which has already
been studied theoretically (Efroni et al., 2021; Lamb et al., 2022), and show that when pre-trained
under exogenous noises in observations, it can lead to significant performance during fine-tuning.
Additionally, we are releasing all exogenous noise based offline datasets such that it can be used
later in the community. We believe, for practical scalability and significance of RL in the real world,
offline RL can play a significant role, and demonstrating ability to learn from such datasets, that are
likely to appear in practice, is an important step in the future.

---

### Author Response · Authors · 2022-11-15
**Summary of Responses, Discussion and Updated Manuscript with New Experiment Results**

We would like to thank all the reviewers for their insightful comments, questions, and acknowledgement of our work, which helped us to provide clarification of our approach and improve the paper. Below, we provide a summary of our responses :

A. We have added new results to the appendix (please see here and in section L for generic responses to all reviewers, including new results)

B. We responded to individual reviewer questions and comments (please see here and in section M of appendix). We have responded to each reviewer questions in responses here too.

Key points for discussion/responses :

1. Comparison to Lamb et al., 2022 : Few reviewers asked for comparison, either empirically or theoretically, to Lamb et al., (2022) (AC-State algorithm). We provided empirical comparison to ac-state, along with clarifications of theoretical claims, which justifies why ACRO is better suited for deep offline RL in general.

2. Inverse Dynamics models and Action Sequence Prediction : Our work already compares to 1-step inverse dynamics models experimentally. We added new results on a variation of ACRO objective, predicting a sequence of actions instead of single actions. We provide clarification on the choice of ACRO objective, and why our approach still has novelty, especially when learning representations in presence of exogenous noise.

3. Removal of Task Relevant Information : Few reviewers asked whether ACRO also removes task relevant information. We provided a justification for why this is not the case.

4. Explanation of Results : We provided clarification on why we averaged results, to provide better insights, since otherwise there would have been too many experimental results to present. Our normalized result figures summarizes the significant improvements that ACRO achieves, experimentally.

5. In individual responses to reviewers : We provided clarification on how the exogenous distractors we work with are different compared to what appeared in the past literature in online RL. We also provide open-source exogenous noise based offline RL datasets which can later be used as benchmarks by the community. We had few minor assumptions made in the work, which have also been clarified. In our responses, we also provided justification to why ACRO performs significantly better even without any additional exogenous noise being present (table 2 of paper). Finally, through our responses, we tried to highlight the significance of the work in terms of novelty, and why we believe ACRO can be a significant contribution in the representation learning and offline RL literature.

We hope our responses would address all concerns of reviewers and help them re-evaluate the score of the paper. The discussion here is helpful to provide better insights of ACRO, and we hope this will clarify a lot of doubts about our method in general. We hope our responses would help reviewers re-evaluate and improve scores of the paper.

---

### Author Response · Authors · 2022-11-18
**Following up on our responses and updated manuscript; can address any further comments/questions**

Dear Reviewers,

We have tried to provide responses to all your questions. We have accordingly updated the manuscript with new experiments (please see section L, Author Rebuttal section).

Please let us know if you have any further questions/comments. Since tomorrow (18th Nov) is the last date for author responses, we would like to address any further questions, if any. We hope our responses and updated manuscript with new results would help you further re-evaluate and improve the score of the paper.


Thanks

---

### Author Response · Authors · 2022-11-22
**Feedback from our Responses and Updated Appendix in Manuscript to address All Reviewer Questions**

Dear Reviewers,

We would like to thank you again for your insightful comments and feedback on our paper.

We have provided responses to all your questions, and also updated the appendix of our paper with new additional experiment results. We would appreciate if the reviewers could take a look at our changes, and additional results, and let us know if they would like to either revise their rating of the paper, or request additional changes that would alleviate their concerns.

Thank you for your time. Looking forward to your feedback.

---

### Author Response · Authors · 2022-11-28
**Any Further Comments/Feedback Based On Our Revised Manuscript?**

Dear Reviewers,

Thank you for your value feedback and comments on our manuscript. We have updated and revised the manuscript based on your feedback, and also posted detailed replies to your comments. We hope that the updated appendix would clarify a lot of your concerns about the paper.

We hope you had the opportunity to read our responses. Please let us know if there are any further things you would want us to address. We look forward to your feedback on the updated manuscript, and hope our responses would help you re-evaluate the score of the paper.

Please let us know if there are any further comments/feedback.


Thanks.

---

### Decision · Program_Chairs · 2023-01-20

**Decision:**

Reject

**Justification For Why Not Higher Score:**

There was a major concern repeated by several reviewers.

**Justification For Why Not Lower Score:**

N/A

**Metareview: Summary, Strengths And Weaknesses:**

This papers aims to find representations for reinforcement learning that are not influenced by irrelevant superfluous information. While the paper and its results were deemed interesting by some reviewers, there was strong concern repeated by several reviewers: the novelty compared to the prior work of Lamb et al., 2022 seems to be quite limited. As such, I recommend this paper to be rejected.